# Symbioses shape feeding niches and diversification across insects

Charlie K. Cornwallis [1] ✉, Anouk van 't Padje [2,3], Jacintha Ellers [2], Malin Klein [2], Raphaella Jackson[4], E. Toby Kiers[2], Stuart A. West [5] & Lee M. Henry [4] ✉

For over 300 million years, insects have relied on symbiotic microbes for nutrition and defence. However, it is unclear whether specific ecological conditions have repeatedly favoured the evolution of symbioses, and how this has influenced insect diversification. Here, using data on 1,850 microbe–insect symbioses across 402 insect families, we found that symbionts have allowed insects to specialize on a range of nutrient-imbalanced diets, including phloem, blood and wood. Across diets, the only limiting nutrient consistently associated with the evolution of obligate symbiosis was B vitamins. The shift to new diets, facilitated by symbionts, had mixed consequences for insect diversification. In some cases, such as herbivory, it resulted in spectacular species proliferation. In other niches, such as strict blood feeding, diversification has been severely constrained. Symbioses therefore appear to solve widespread nutrient deficiencies for insects, but the consequences for insect diversification depend on the feeding niche that is invaded.

Across the tree of life, microbial symbionts have enabled organisms to harness new forms of energy, access unobtainable nutrients and outsource critical functions such as defence[1–4]. So valuable are symbiotic partnerships that they have repeatedly led to organisms becoming obligately dependent on each other for survival[5]. Such interdependence between hosts and symbionts has led to the evolution of new levels of organismal complexity that have ultimately shaped the diversity of life on Earth[3,6].

The essential metabolic services provided by symbionts have enabled hosts to expand into previously uninhabitable environments[1,4,7]. For example, sulfur-oxidizing bacteria enable giant marine tubeworms to live in deep-sea vents, root-associated fungi helped plants colonize land and nutrient-supplementing symbionts have allowed insects to live solely on the imbalanced diets of plant sap and vertebrate blood[2,8,9]. However, it is unclear whether there are unifying factors that guide how and why symbiotic relationships evolve.

Insects are an excellent system to study the evolution of obligate symbiosis. Multiple insect families have acquired microbes to perform a range of functions, including defence and nutrition[10]. Defensive symbionts protect their hosts from attack by natural enemies[11], whereas nutritional symbioses allow insects to feed on specialized resources that lack essential nutrients, such as plant sap, blood (haematophagy) and wood (xylophagy)[2]. It is therefore widely accepted that symbiotic partnerships have opened new ecological niches and helped the incredible diversification of insects[7]. However, previous work has primarily focused on the functional role and impact of obligate symbiosis within single groups of insects. Consequently, whether we can generalize about the ecological causes and consequences of obligate symbiosis across different groups of insects is unknown. Are there consistent nutrient limitations that have repeatedly selected for the evolution of symbioses across different feeding niches? Do symbioses influence diversification in a consistent or niche-dependent way?

[1]Department of Biology, Lund University, Lund, Sweden. [2]Amsterdam Institute for Life and Environment, section Ecology and Evolution, Vrije Universiteit, Amsterdam, the Netherlands. [3]Laboratory of Genetics, Wageningen University and Research, Wageningen, the Netherlands. [4]School of Biological and Behavioural Sciences, Queen Mary University of London, London, UK. [5]Department of Biology, University of Oxford, Oxford, UK. ✉e-mail: charlie.cornwallis@biol.lu.se; l.henry@qmul.ac.uk

In this article, we address these questions by examining the macro-evolutionary patterns of obligate symbiosis across 1,850 microbe-insect combinations from 402 insect families. Data were collated across bacteria, fungi and protist symbionts with nutritional and defensive functions (Supplementary Tables 1–4). First, we estimated how often insect lineages within different feeding niches have evolved obligate symbiosis, where the host cannot survive without symbionts. We are interested in cases where hosts are obligately or highly dependent (effectively obligately) on their symbionts. Obligate dependence is ideally proven experimentally, but only a limited number of such studies exist[5]. To allow comparison across a wider range of species, we used two criteria to establish putative obligate dependence, hereafter referred to as obligate dependence, both of which had to be fulfilled: (1) the symbiont is universally present in reproductive females; and (2) the insect possesses morphological structures that are predominantly associated with symbionts being required for survival (for example, bacteriocytes[10]), or where information on symbiont housing organs was lacking, data on the impact of symbiont removal and patterns of host–symbiont co-speciation were used to determine obligate dependence ('Insect and symbiont data' in Methods). Known parasitic symbionts, such as reproductive manipulators (for example, *Spiroplasma*, *Cardinium* and *Wolbachia*), that have not evolved beneficial functions were excluded from our dataset.

Second, we examined the composition of insect diets to determine whether specific nutrient deficiencies have consistently led to the evolution of obligate symbiosis across different feeding niches. The nutritional composition of diets was determined by collating literature on the food sources used by adults and juveniles ($n_{food\,sources}$ = 362) and extracting information on carbohydrates, fats, proteins, essential amino acids, non-essential amino acids and vitamins A, B, C and E from as many example foods as possible (range 1–24) from nutritional databases ('Nutrient data' in Methods, Supplementary Table 4 and Extended Data Fig. 1). Data on other vitamins were collected but had >30% missing data and so were excluded from analyses ('Nutrient data' in Methods). We differentiate between insect families that specialize on single plant-based resources (phloem, xylem or wood) from families that exploit various plant parts (phytophagy, referred to here as herbivores), as there were large differences in the nutrients of these diets (Supplementary Tables 1 and 4). Third, we tested if the acquisition of obligate symbionts has increased or decreased host species richness after radiating into different feeding niches. We circumvent the problem of poorly resolved species level phylogenies by reconstructing the evolutionary history of obligate symbioses at the family level.

## Evolutionary origins of obligate symbiosis

We found at least 16 independent origins of obligate symbiosis spread across 89 insect families (Bayesian phylogenetic mixed model (BPMM): Fig. 1 and Supplementary Table 5). These origins were estimated on the time-calibrated phylogeny[12] to date back as far as 336 million years. Within insect families, there were also several more recent transitions to obligate symbiosis. For example, 15 families were found to contain species with and without obligate symbionts (Fig. 1 and Supplementary Tables 1 and 2) but without species phylogenies the exact number of origins cannot be resolved. Our analyses therefore focus on the deeper, family-level origins of symbiosis, while accounting for variation within families by modelling the percentage of species within families with obligate symbionts ('Specific analyses' in Methods).

Reconstructing the ancestral feeding niches of insect families showed that obligate symbioses evolved from omnivorous, herbivorous and predatory ancestors (respective percentage of origins estimated using stochastic character mapping (SCM): 75%, 8% and 17%; Fig. 1 and Supplementary Table 5). Following the acquisition of obligate symbionts, 60% of lineages switched to a single food source (phloem 42%, blood 12%, xylem 6%; Fig. 1 and Supplementary Table 5). This pattern of food utilization explains the current distribution of obligate

symbiosis remarkably well, where over 90% of insect species feeding on blood, phloem, xylem and wood have obligate symbionts (Fig. 1 and Supplementary Tables 1, 2 and 6). Conversely, there are no known cases of obligate symbioses in insect families that are predominantly predators or fungivores (Fig. 1 and Supplementary Tables 1, 2 and 6).

Only five insect families are known to have endosymbionts with defensive functions. This is probably influenced by sampling effort, as defensive symbionts residing within insect hosts have only been discovered relatively recently[11,13], in taxa such as aphids, drosophilids, psyllids, crabronids (beewolves) and beetles (reviewed in ref. 14). There are well-known examples of defensive mutualisms that reside outside of the host (ectosymbionts), such as the antimicrobial producing actinobacteria of fungus-farming ants[15], but these were excluded as our analyses focus on endosymbiosis. Out of the 13 endosymbiont species shown to provide insects with protective services, nearly all maintain facultative relationships with their hosts. There is only one exception in our database, the Asian citrus psyllid, *Diaphorina citri*, which has evolved obligate dependence on a defensive symbiont, which is housed in bacteriocytes alongside a putative nutrient provisioning symbiont[16].

Several defensive symbioses show evidence of strong vertical inheritance and are associated with hosts at high frequencies, such as those found in lagriid beetles[17], beewolves[18] and fungus-growing ants[19], suggesting they may be near obligate in nature. However, the absence of the symbiont in some individuals[18,20], existence of multiple symbiont strains within the same host individual[20] and evidence of frequent acquisitions from environmental sources[19,21] demonstrates that most defensive symbioses have not reached the high degree of mutual dependence observed in obligate nutritional associations. While more work is clearly needed, these data support the hypothesis that selection for protection against natural enemies is too inconsistent across generations to favour the evolution of obligate dependence[5,11].

## Nutrient deficiencies and obligate symbiosis

Our results show that the evolution of obligate symbioses in insects is associated with transitions to specialized feeding niches (Fig. 1). Studies have shown that symbionts have enabled these transitions by synthesizing a range of essential nutrients missing in their hosts' diet including vitamins, carotenoids and amino acids, as well as digestive enzymes that aid in nutrient recycling[10,22–24]. However, it is not clear if certain key nutrients are consistently deficient in the diets of insects with obligate symbionts across different feeding niches.

We found that only one dietary component was significantly correlated with the presence of obligate symbionts across all feeding niches: low levels of B vitamins (Fig. 2; BPMM: phylogenetic correlation −0.32, credible interval (CI) −0.54 to −0.09, proportion of iterations above or below a test value correcting for the finite sample size of posterior samples (pMCMC) = 0.006; Supplementary Table 7). This pattern held across hosts with diets that are highly variable in carbohydrates, proteins, fats, vitamins and amino acids (Fig. 2 and Supplementary Table 7). Examining types of B vitamins further showed that specifically B5 and B9 are phylogenetically negatively correlated with the evolution of obligate symbiosis (Fig. 2 and Supplementary Table 8). Different types of B vitamins were, however, highly correlated, indicating that sets of B vitamins are often concurrently absent from the diets of some insects (Extended Data Fig. 2). For example, vitamins B1 and B6 were correlated with B9 (Pearson's correlation coefficients $r$ = 0.89 and 0.96) and vitamins B2 and B3 were correlated with B5 ($r$ = 0.92 and 0.64; data on vitamins B7 and B12 had >30% missing data and so were not analysed).

No other macro- or micronutrients were significantly correlated with obligate symbiosis across insect families (Fig. 2 and Supplementary Table 7). This is not to say that obligate symbionts are not important for provisioning other nutrients, but rather these nutrients are restricted to specific niches. For example, essential amino acids are deficient in certain feeding niches associated with obligate symbiosis,

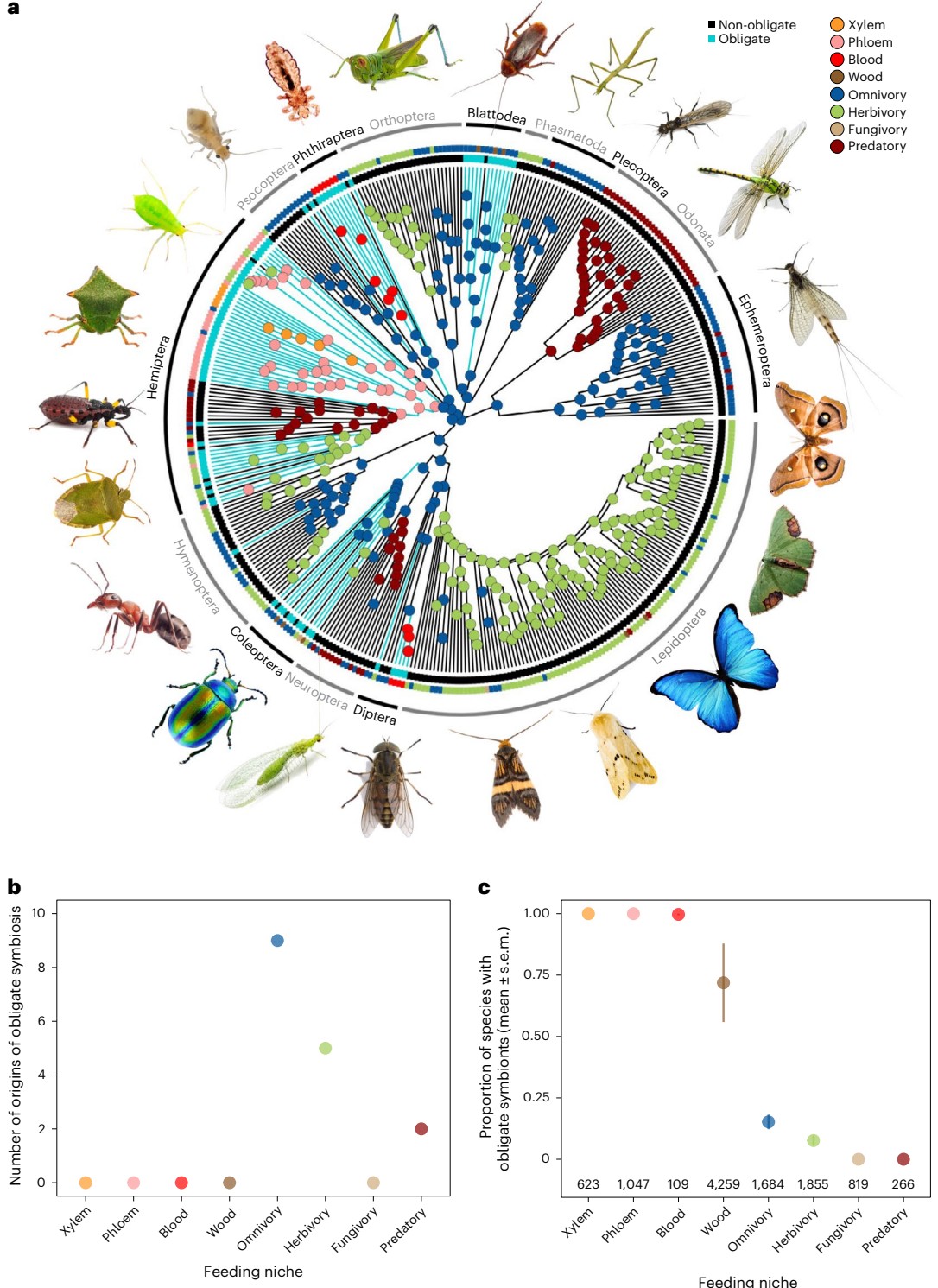

**Fig. 1 | The evolutionary origins of obligate symbionts and their association with different feeding niches. a**, The phylogenetic distribution of obligate symbionts across insect families investigated for symbiosis and their feeding niches. Turquiose tips and branches represent obligate symbiosis and different coloured dots represent different feeding niches. Ancestral feeding niches were estimated using SCM, and obligate symbiosis states were estimated using a BPMM (Supplementary Table 5; for tree with tip labels, see Extended Data Fig. 4). **b**, The number of times obligate symbiosis evolved in different ancestral feeding niches of insects estimated using a BPMM. **c**, Proportion of species within families with obligate symbionts (mean ± standard error of the mean (s.e.m.)) in relation to the feeding niches of insects. The average number of species within families is given along the *x* axis.

such as phloem (BPMM: phloem versus background levels $\beta = -1.11$, CI = −1.53 to −0.67, pMCMC = 0.001), but are enriched in other niches with symbionts, such as blood (BPMM: blood versus background levels $\beta = 3.25$, CI = 2.77 to 3.86, pMCMC = 0.001; Supplementary Table 9). Note that all amino acids concentrations were highly correlated across insect diets (Extended Data Fig. 3).

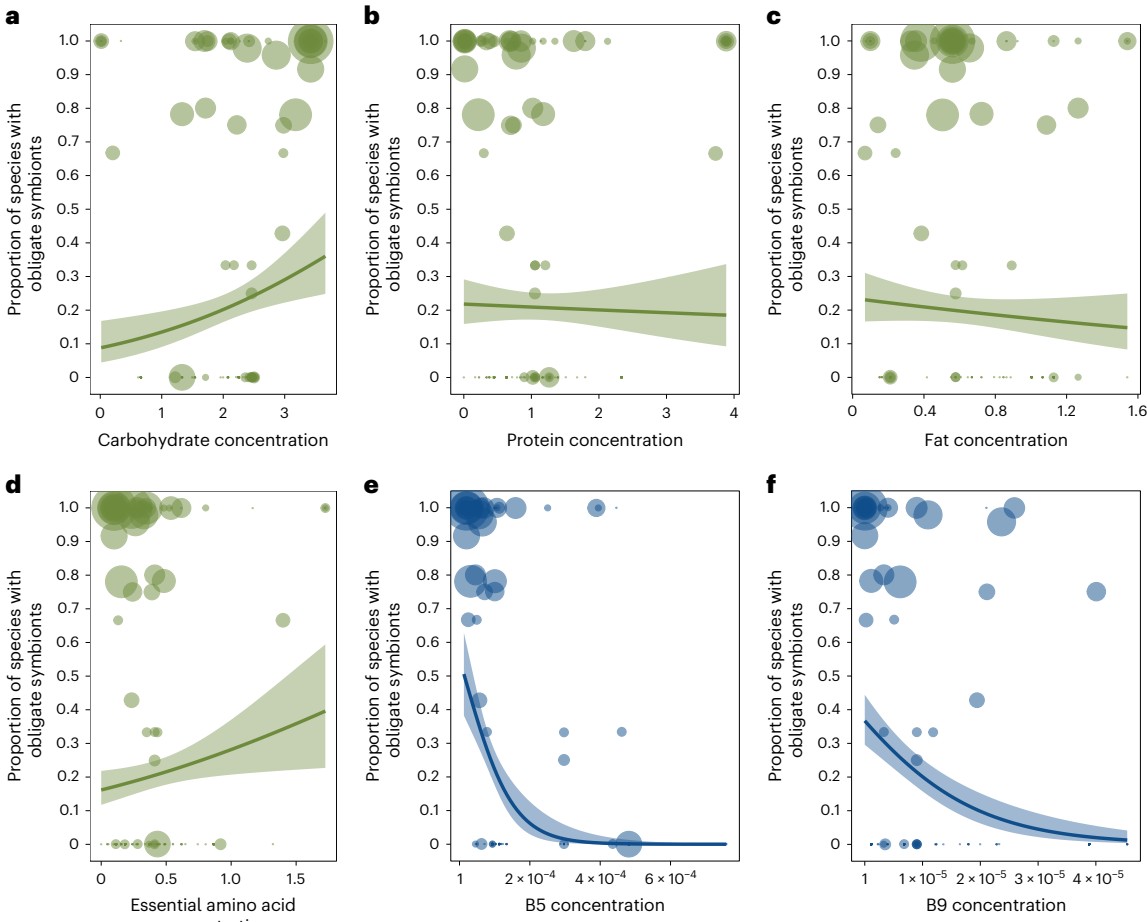

**Fig. 2 | Nutrient deficiencies and the evolution of obligate symbioses.**
**a**–**d**, Across all diets, macronutrients (**a**–**c**) and essential amino acids (**d**) were not consistently associated with the proportion of species within families that had obligate symbionts. **e**,**f**, Insect families with diets deficient in B5 (**e**) and B9 (**f**) vitamins had a significantly higher proportions of species with obligate symbionts than families feeding on diets with high levels of B vitamins (B5 phylogenetic correlation (CI) −0.45 (−0.61 to −0.22), pMCMC = 0.001; B9 phylogenetic correlation (CI) −0.25 (−0.48 to −0.05), pMCMC = 0.03). Nutrient values on x axes are standardized amounts per gram ('Nutrient data' in Methods). The size of points represents the number of host species (log-transformed) examined for obligate symbionts per family. Lines represent logistic regressions with 95% confidence intervals (shaded bands) plotted for illustrative purposes. Analyses of individual amino acids confirmed that obligate symbiosis was not consistently associated with any amino acid deficiencies ('Obligate symbiosis and types of amino acids' in Methods and Supplementary Table 29).

Our results are consistent with detailed studies that have demonstrated the fitness consequences of providing B vitamins to specific insect species. For example, the fitness of tsetse flies depends on B9 and B6 vitamins provided by *Wigglesworthia* bacteria[25,26], and *Buchnera* supplements aphids with B5 and B2 vitamins, with B5 having a particularly strong effect on host survival[27]. Similarly, the *Baumania* symbiont of xylem-feeding sharpshooters has retained the capacity to synthesize six B vitamins (all but B3 and B12), and the coordinated host–symbiont biosynthesis of B7 and B5 in whiteflies has been shown to be critical for host fitness[28,29]. Dietary studies have also confirmed that mutualistic *Wolbachia* provide essential B vitamins for *Cimex* bed bugs[30]; and metabolic homeostasis is restored in symbiont-free *Dysdercus* cotton stainers when B vitamins are supplemented, or hosts are re-infected with their actinobacterial symbionts[31]. In addition, genome studies have shown that the metabolic pathways to biosynthesize B vitamins have been retained in the genomes of symbionts from insects that occupy diverse feeding niches (for example, blood[32], plant sap[33,34] and seeds[35]), suggesting their widespread importance in maintaining symbioses. A useful future step would be to assess whether B-vitamin pathways are more highly conserved in symbiont genomes, compared with pathways that encode other host-beneficial factors.

## Evolutionary transitions to nutrient-deficient diets

Our results suggest that B-vitamin deficiencies are widespread and important for the evolution of obligate symbiosis in insects. There are, however, two evolutionary scenarios for why such transitions occur. One possibility is that insects feeding on diets low in vitamin B recruited symbionts to supply B vitamins. The alternative is that insects first acquired obligate symbionts that could synthesize B vitamins, possibly for some other benefit, which then enabled them to invade feeding niches where B vitamins were scarce. The question is therefore whether the evolution of obligate symbioses were triggered by low B vitamins in diets or whether obligate symbioses facilitated specialization on these diets.

We tested these competing hypotheses by estimating the amount of B5 and B9 vitamins in ancestral diets before, and following, transitions to obligate symbiosis. A high phylogenetic signal of B vitamins across insects allowed us to relatively accurately estimate ancestral values (Supplementary Table 8 and Extended Data Fig. 4). We found little evidence that levels of B5 and B9 vitamins were reduced in the diets of insects before they acquired obligate symbionts (Fig. 3 and Supplementary Tables 10 and 11). Instead, we found that hosts that

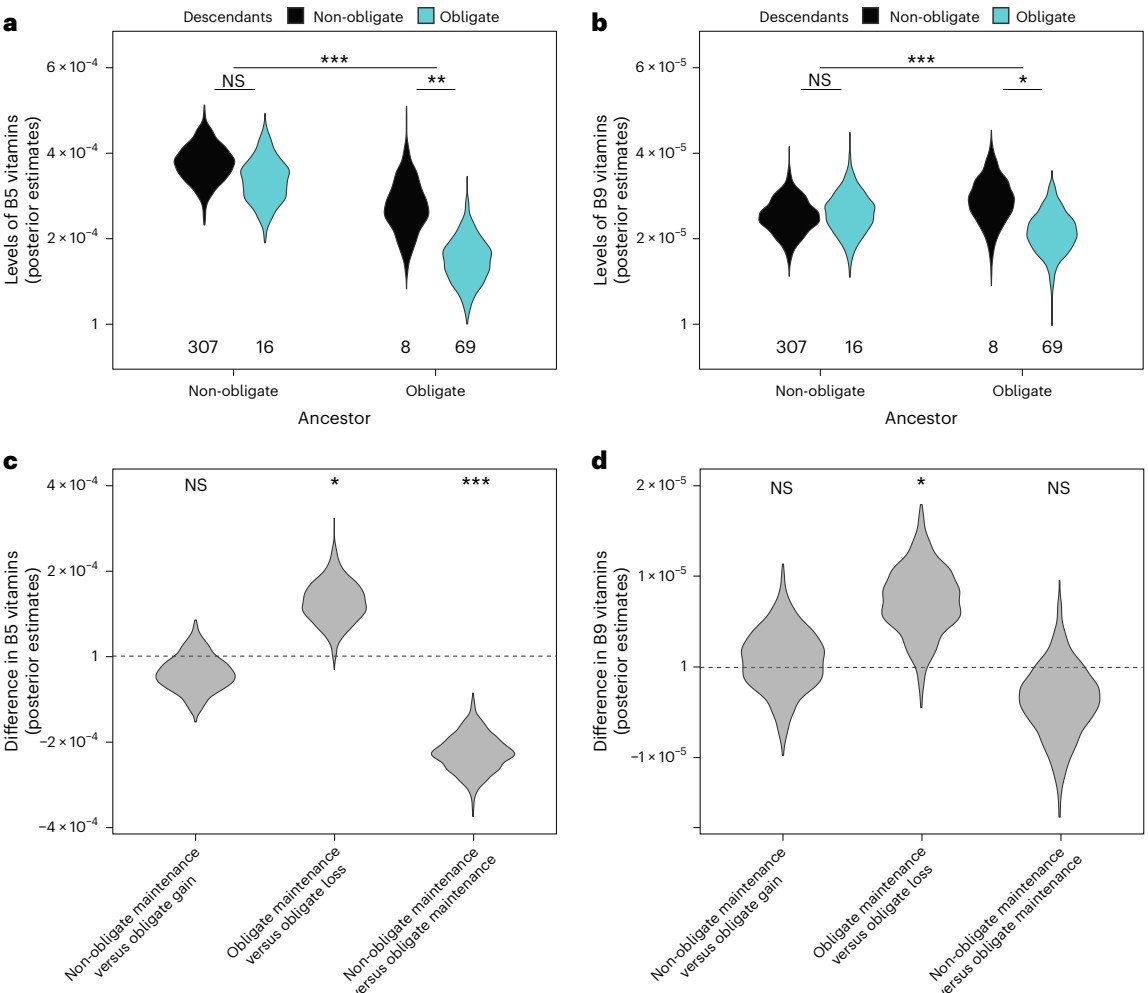

**Fig. 3 | Obligate symbiosis enables evolutionary shifts to diets deficient in B vitamins. a,b,** The ancestors of lineages that evolved obligate symbioses had similar levels of B5 (**a**) and B9 vitamins (**b**) in their diets compared with lineages that did not evolve obligate symbioses. **c,d,** After acquiring obligate symbionts, lineages evolved diets with significantly lower levels of B5 (**c**) and B9 vitamins (**d**) ('non-obligate maintenance versus obligate maintenance' comparison). The evolutionary loss of obligate symbiosis was also associated with increases in dietary levels of vitamin B5 and B9 compared with where obligate symbiosis was maintained (**c,d**: 'obligate maintenance versus obligate loss' comparison). Vitamin B concentrations are standardized amounts per gram for each ancestral node, estimated from the diets of extant insect families ('Nutrient data' in Methods; for reconstructed levels of B vitamins plotted on the

tree and robustness of estimates to rate shifts in B vitamins, see Extended Data Fig. 4). Numbers along the $x$ axis in **a** and **b** indicate the numbers of transitions. Violin density curves represent the posterior distribution of estimated ancestral levels of B5 and B9 vitamins (1000 samples) estimated using a BPMM. The violin width corresponds approximately to the most likely estimate of B vitamins. A BPMM was used to test for significant differences between transitions (*pMCMC <0.05, **pMCMC <0.01, ***pMCMC <0.0001; exact pMCMC values are given in Supplementary Table 10): In **a** and **b** we tested if the posterior distribution of B-vitamin estimates for a given transition was above or below the comparison transition, and in **c** and **d** we tested if the posterior distribution of the difference in B-vitamin estimates for transition comparisons was above or below 0. NS, not significant.

recruited obligate symbionts subsequently evolved to specialize on diets with low levels of B5 and B9 vitamins (Fig. 3 and Supplementary Tables 10 and 11). Once obligate symbioses evolved, shifts to diets deficient in B vitamins were much more frequent, particularly for B5, where transition rates to low levels of B5 were 30 times higher than for lineages without obligate symbionts (Supplementary Table 11).

The importance of obligate symbionts in supplying B vitamins was further supported by the loss of obligate symbioses when insects switched to diets with elevated B-vitamin levels (Fig. 3 and Supplementary Tables 10 and 11). Insect lineages with above-average levels of B5 and B9 vitamins in their diets were more likely to lose their obligate symbionts (Fig. 3a,b and Supplementary Table 11). Our results match with observations from specific taxa, where obligate symbiont losses have been associated with dietary changes in their insect hosts. In the mealybug genus, *Hippeococcus*, symbiont losses are thought to be associated with nutrient provisioning by *Dolichoderus* ants, and *Typhlocybides*

plant hoppers lost their ancestrally acquired obligate symbionts when switching from plant sap to more nutrient-rich parenchyma[36].

## Symbiont specialization in nutrient provisioning

Given the importance of B-vitamin provisioning by symbionts in insects, we examined whether specific lineages of symbiotic bacteria specialize in providing B vitamins to hosts. Have hosts relied on a restricted set of symbiotic partners, or have a variety of symbionts converged to provide B vitamins? To address this question, we constructed a phylogeny for symbionts to quantify the amount of variation in dietary B vitamins explained by symbiont ancestry and their co-evolutionary relationships with hosts.

We found that hosts have evolved dependence on a broad range of microbes (Supplementary Tables 12 and 13). Less than 1% of variation in B5 and B9 vitamins in host diets was explained by symbiont phylogenetic history and the co-evolutionary relationships between symbionts

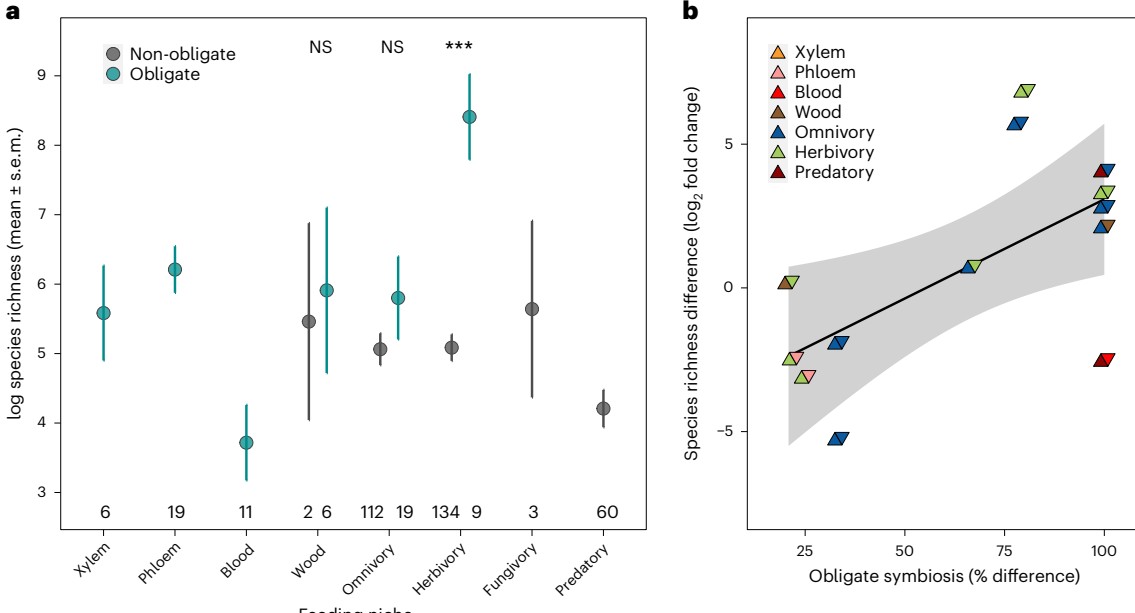

**Fig. 4 | Obligate symbioses and the evolutionary potential for diversification.**
**a**, Diversification was measured as the number of species within families presented on a natural logarithmic scale (mean ± s.e.m.). BPMMs were used to test if the species richness of each feeding niche was significantly higher or lower than the average of all other niches (*pMCMC <0.05, **pMCMC <0.01, ***pMCMC <0.0001. Exact pMCMC values are given in Supplementary Table 17), and if families with (>50%) and without (<50%) obligate symbionts feeding within the same niche had higher or lower species richness (Supplementary Table 17). BPMMs controlled for differences in family age, holo- and hemi-metabolism and insect phylogenetic history. The number of insect families are given along the

x axis. **b**, Sister lineage comparisons showing families with a higher percentage of species with obligate symbionts are associated with greater species richness (y axis: log₂ fold change: 0, no difference; 1, double the number of species; −1, half the number of species; Extended Data Fig. 6 shows differences in absolute numbers of species). Details of the taxa involved in sister comparisons are presented in Supplementary Table 29. Symbols are coloured by the feeding niches of the sister families, with colours on the left indicating the feeding niche of the family with the lowest rates of obligate symbiosis and the highest on the right. Regression line with 95% confidence intervals (shaded band) plotted for illustrative purposes. NS, not significant.

and hosts (BPMM: symbiont phylogeny (% variance) B5 0.08, CI = 0.02 to 0.15; B9 0.1, CI = 0.03 to 0.2; co-evolutionary interaction (% variance) B5 0.06 CI = 0.02 to 0.12; B9 0.09 CI = 0.03 to 0.16; Supplementary Tables 12 and 13). Instead, divergent symbiotic lineages appear to have become convergently associated with insects feeding on low-vitamin-B diets (Extended Data Fig. 5). Following the establishment of obligate symbioses, hosts and symbionts tend to co-evolve, as related insect families were significantly more likely to be partnered with phylogenetically similar symbionts (BPMM: co-evolutionary interaction (% variance) 22.65, CI = 10.47 to 37.35; Parafit: *P* = 0.05; Supplementary Tables 14 and 15). These results match with research showing that diverse symbiotic bacteria have retained the genes for synthesizing B vitamins[37], and those insects whose bacteria lose the capacity to provide B vitamins recruit new symbiont lineages to compensate for the loss[38].

## Obligate symbiosis and insect diversification

Finally, we tested if obligate symbioses have influenced patterns of diversification across insects. The current paradigm, based on observations from specific lineages, such as sap-feeding Hemipterans, is that the acquisition of symbionts opens up new niches and increases host species proliferation[39,40]. Host–symbiont co-evolution can also generate incompatibilities between populations that may increase speciation rates[4]. Dependence on symbionts may, however, 'trap' hosts in specific niches, leading to the opposite prediction that symbionts reduce diversification[4]. For example, hosts can be restricted to feeding on specific resources because of symbiont-assisted specialization[41], or limited by the sensitivity of their obligate symbionts to environmental conditions, such as temperature[42,43]. Mutation accumulation can also degrade symbiont functioning, resulting in hosts being stranded with maladapted symbionts that may increase extinction risk[44].

Although symbioses are thought to have facilitated adaptive radiations in specific insect lineages[7,45], these competing hypotheses have not been systematically tested across insects, generating debate over the general role of symbiosis in insect diversification. We therefore analysed the relationship between obligate symbiosis and species richness in three ways: across all families, across families with and without obligate symbionts that have the same feeding niche, and between sister lineages.

Across all insects, we found that obligate symbiosis was associated with extreme highs and lows of species richness compared with background rates (Fig. 4a and Supplementary Tables 16 and 17). At the extreme high, herbivorous insect families with obligate symbionts had 12 times as many species compared with the average across families (Fig. 4a; BPMM: herbivores with obligate symbionts versus background species richness 2.75, CI = 0.97 to 4.01, pMCMC = 0.001). At the other extreme, extraordinarily low species richness was associated with insect families feeding on blood, which had eight times fewer species than the average (Fig. 4a; BPMM: blood feeders with obligate symbionts versus background species richness −1.95, CI = −3.64 to −0.26, pMCMC = 0.01). This resulted in a 51-fold difference in the number of species in herbivorous insect families with obligate symbionts versus those in blood-feeding niches. These estimates of species richness were after accounting for differences between holo- and hemi-metabolism, the age of insect families and insect phylogenetic history, all of which can affect the number of species in families[12,46].

## Obligate symbiosis promotes diversification within niches

Patterns of species richness across insects appear to be niche specific (Supplementary Table 17). However, within feeding niches, symbionts

may directly promote diversification if they allow species to exploit different resources. For example, in insect families feeding on more varied diets, such as generalist herbivores and omnivores, symbionts may enable resource partitioning between species, fuelling the speciation process. There were three feeding niches where families with and without obligate symbionts could be compared: herbivores, omnivores and wood feeders. If true, then families within these niches with obligate symbionts should have higher species richness than families without them.

We found that herbivorous insect families with obligate symbionts had 15 times as many species as families without symbionts (Fig. 4a; BPMM: families with versus without obligate symbionts 3.47, CI = 1.57 to 4.79, pMCMC = 0.001; Supplementary Table 17). Omnivorous and wood feeding families of insects with obligate symbionts also had two to three times as many species as families that lacked symbionts, but these differences were not statistically significant (Fig. 4a; BPMM: omnivorous families with versus without obligate symbionts 0.17, CI = −1.08 to 1.34, pMCMC = 0.78; wood families with versus without obligate symbionts 0.13, CI = −2.78 to 3.71, pMCMC = 0.71; Supplementary Table 17).

In support of obligate symbiosis playing a role in promoting diversification at finer evolutionary scales, we found that among more closely related sister taxa, lineages with higher percentages of obligate symbionts (n = 13) were more specious (Fig. 4b; BPMM: percentage of species with obligate symbionts 7.43, CI = −0.03 to 12.79, pMCMC = 0.05; Supplementary Tables 18 and 19 and Extended Data Fig. 6). Our results are consistent with findings from specific taxonomic groups. For example, symbionts allowed Curculionidae weevils, now one of the most diverse families of insects, to feed and radiate exclusively on plants[47,48]. Similarly, the success of certain highly specious ant lineages has been facilitated by nutrient-provisioning symbionts that have allowed them to thrive on primarily plant-derived diets[49].

## Sensitivity analyses

We examined the robustness of our analyses to different methodological approaches. First, we examined the robustness of our estimates of obligate symbiosis and ancestral feeding niches to different analytical approaches (Supplementary Table 20 and 'The number of origins of obligate symbiosis' and 'Estimating ancestral feeding niches' in Methods). Second, we tested how inserting families (n = 23) that were not included in the published phylogeny[12] influenced our results (Supplementary Table 21 and Extended Data Fig. 7). Third, we examined the robustness of our results to excluding non-bacterial symbionts and families that had multiple co-occurring obligate symbionts (Supplementary Tables 22–24 and Extended Data Fig. 8). Fourth, we repeated our analyses using a second dataset restricted to species where dependence on symbionts had been directly studied, rather than inferred from microscopy studies on the presence of bacteriocytes within certain insect orders and superfamilies (Supplementary Tables 25–27 and Extended Data Fig. 9). Fifth, 15 out of the 402 insect families had species with and without obligate symbionts. This was accounted for by modelling the percentage of species with obligate symbionts within families. However, in the transition rate analyses, families had to be classified as having obligate symbionts (>50% of species with obligate symbionts) or not (<50% of species with obligate symbionts). We tested the sensitivity of these analyses to this binary classification by re-running models including only families where 100% of species had, or did not have, obligate symbionts (Supplementary Table 28). All of these different analyses provided qualitatively and quantitatively similar conclusions ('Sensitivity analyses' in Methods).

## Discussion

Genomic and experimental studies have provided important insights into the metabolic function of obligate symbioses for specific insects. This previous work has focused on well-studied groups—mostly plant-feeding hemipterans containing symbionts with highly reduced genomes[10,24,50,51], but also herbivorous beetles[23,47,52], omnivorous ants[53], cockroaches[54] and blood-feeding insects (flies[55,56], bed bugs[30] and lice[57]). Our approach complements this work by allowing tests of the importance of obligate symbioses in overcoming nutrient limitations at a much broader taxonomic scale, including lineages where symbioses have not been studied in-depth using detailed molecular methods. This enables contrasts to be made between insect lineages with and without symbionts, circumventing potential biases that arise from studying only positive associations.

Our analyses highlight that B-vitamin deficiencies are the primary nutrient limitation associated with obligate symbioses. While B vitamins are consistently important, this does not mean that other nutritional deficiencies are not important for specific clades. For example, certain ant and beetle lineages have evolved tyrosine-supplementing symbioses that are thought to help thicken their heavily sclerotized cuticles[47,53,58]. Specialized herbivores have evolved symbioses to solve a variety of problems, from including the production of digestive enzymes to breakdown of plant cell walls[23] and the synthesis of specific amino acids missing from plant-based diets[35]. Similarly, transitions to feeding on plant phloem and xylem are well known to be associated with essential amino acid provisioning by bacteria in insects[2,24,52]. These feeding niches are, however, also deficient in B vitamins, a common feature of symbiont-associated diets, including those rich in amino acids, such as blood feeding. The widespread need for B-vitamin supplementation is also reflected in the genomes of symbionts that retain metabolic pathways to synthesize B vitamins across a diversity of host feeding niches[32–35]. Further insights into the consistent need for B-vitamin supplementation may be gained by extending such genomic analyses to compare the synthesis pathways of B vitamins with those of other nutrients, such as amino acids, in both symbionts and their insect hosts.

To conclude, it appears possible to make relatively broad inferences about the causes and consequences of obligate symbioses in insects. Vitamin B supplementation by microbial partners is widespread in insects and has helped insect hosts to exploit novel food resources. In some cases, such as herbivorous insects, the shift to feeding on new resources appears to have facilitated adaptive radiations, analogous to textbook examples such as Darwin's finches[59]. In other cases, such as strict blood feeding, new niches seem to have severely constrained diversification. The intricate relationships between hosts and their nutritional symbionts therefore appear key to shaping patterns of global insect diversity.

## Methods

### Data collection

**Insect and symbiont data. Literature searches.** We compiled a database on insect–microbe symbioses by: (1) searching published literature using the following key words [order name] OR [family name] AND 'symbio'* using the search engines Web of Science and Google scholar during 2015–2017 and again in 2020, (2) searching prominent reviews (for example, Ries[60], Schneider[61], Müller[62], Buchner[36], Douglas[63], Abe et al.[64], Bourtzis and Miller[65–67], Baumann[68] and Baumann et al.[69]) and (3) forward and backward searches from the resulting papers. A full list of the papers screened can be found in Supplementary Table 2.

The insect families included in the literature search were those listed in Bouchard et al.[70], Davis et al.[71] and Rainford et al.[12], and those included in published phylogenies investigating insect biodiversity: Hedges et al.[72], Misof et al.[73] and Rainford et al.[12]. For symbiont detection, we considered only studies using methods capable of capturing phylogenetically diverse bacteria species (for example, deep-coverage sequencing, or cloning, using 'universal' 16S ribosomal RNA (rRNA) primers), or microscopy studies investigating whole insects for the presence of symbionts.

Specific clades of insects are known to carry the same obligate symbionts due to strict vertical transmission (Supplementary Table 2 'reference obligate criteria'). We therefore searched Genbank to recover

all insect species that have been associated with specific vertically transmitted symbionts (identified taxonomically by symbiont genus name in most cases) to increase our coverage of host–symbiont associations (Supplementary Table 3). Search results were checked manually to ensure host species belonged to the insect clade known to harbour the symbiont (Supplementary Table 3). In families that have species both with and without obligate symbionts, we considered only species directly studied for obligate symbiosis. Vertically transmitted symbionts were included only in analyses of host evolution ('Specific analyses' in Methods). In analyses of host–symbiont co-evolution we were interested in testing how host and symbiont phylogenetic history influence symbiont recruitment ('Nutrient deficiencies and host–symbiont co-specialisation' in Methods). Vertically transmitted symbionts were excluded as they inflate the signature of co-evolution, not because of symbiont recruitment, but because of inheritance from a common ancestor.

**Data inclusion and exclusion.** The aim of our paper was to investigate the evolution of beneficial obligate symbioses. We therefore excluded studies: (1) on parasitic symbionts, such as those that manipulate host reproduction (for example, *Spiroplasma*, *Cardinium* and *Wolbachia*) that have not evolved beneficial functions; (2) that failed to screen the entire insect (for example, performed only insect gut analyses); and (3) on symbionts with presumed beneficial functions, but that lacked data needed to assess our obligate criteria (see below). Fungal and protist symbionts were included where data on host dependency were available. Analyses of host–symbiont co-evolution were restricted to symbionts for which a phylogeny could be constructed (bacteria with 16S rRNA genetic data: for details, see 'Insect and symbiont phylogenies' in Methods).

For each insect–microbe association we collected data on: the insect species; juvenile and adult insect diets; whether insects were holo- versus hemi-metabolous; the identity of symbionts; symbiont domain; whether symbionts were intra- or extracellular; whether symbionts were housed within specialized structures (for example, bacteriocytes); whether symbionts are thought to have a defensive or nutritional function in nature; and whether insects were obligately dependent on symbionts (for assessment criteria, see below).

**Criteria for assessing obligate symbiosis.** We are interested in cases where hosts are obligately or highly dependent upon their symbionts (effectively obligate). Obligate dependence is usually proven experimentally, but only a limited number of such studies have been carried out[5]. Consequently, to allow comparison across a wider range of species, we used the following criteria as indicators of putative obligate dependence, both of which must be fulfilled: (1) Symbiont is universally present in reproductive females; and (2) symbionts are housed in a bacteriome (or mycetome) within bacteriocytes (specialized symbiont-housing cells), or insect-symbiont phylogenies are concordant, or symbiont removal results in significant reduction in host fitness.

In well-studied systems, insects with bacteriocytes typically cannot survive without their symbionts under natural conditions, as observed in aphids[74], coccids[75], carpenter ants[76], cockroaches[77], anobiid beetles[78], cerambycid beetles[79], tsetse flies[80] and lice[81] (for examples under other conditions, see refs. 75,82,83). In these cases, stable mutual dependence is evident from extreme partner fidelity, genetic uniformity of symbionts within a host species, and concordance of host and symbiont phylogeny as the microbe is faithfully maternally transmitted from a common ancestor (reviewed in refs. 10,84). Consequently, species where a single symbiont is not universally present in all reproductive females, or where specialized symbiont housing organs are lacking were classified as not having obligate symbionts. If symbionts were universally present, but cophylogenetic and/or host fitness data were unavailable the relationship was classified as unresolved and were not included in analyses.

Data on individual species were used to estimate the percentage of species in each family that have evolved dependency on symbionts, which is summarized in Supplementary Table 1. Data on the insect species examined, their associated symbionts and the criteria to assess dependency are provided in Supplementary Table 2.

**Feeding niche classification.** The feeding niches of species were classified using information on their diets. Omnivores were defined as species that feed on both plant and animal matter, or those that scavenged on detritus. Due to large differences in the nutrient contents of different plant tissues, insect species that specialize on phloem, xylem and wood (xylophagy) were considered separately from species that exploit non-vascular/non-woody plant tissues (for example, leaves, flowers, fruits, seeds and/or root tips), which we refer to as generalist herbivores (also known as phytophagous).

The feeding niches of families were classified using the information on species feeding niches (Supplementary Table 1). Families were described as having omnivorous diets if they contained species that were omnivores/detritivores, or if single species within families utilized multiple food sources that combined plant, animal or fungal material. Families containing species that fed on multiple plant tissues were classified as generalist herbivores. Families were assigned to the feeding categories of haematophagy, phloem feeding, xylem feeding and predatory where most species in the family, if not all, feed exclusively on those resources. Families assigned as xylophagous were those where most species fed on wood as their primary food source. In cases where species-specific diets were not available, we based diets on family-level feeding habits published in books and reviews listed in Supplementary Table 2.

**Nutrient data.** To estimate the nutrient composition of each insect family, we performed the following steps:

(1) The food types (for example, fruit and roots) utilized by adults and juveniles of all species in our obligate symbiosis dataset were collected from published literature (Supplementary Table 2). Literature was searched using the terms [species name] and [adult diet] or [juvenile diet] in Web of Science and Google Scholar. Where possible, we cross-validated diet assignments using multiple published studies (Supplementary Table 2).

(2) The nutrient composition of all food types that species were found to feed on ($N_{food\ types}$ = 362) was estimated by searching dietary databases for as many examples of those food types as possible (range per food type 1–24, total number of nutrient estimates 5,446; Supplementary Table 4). From dietary databases, we extracted information on the concentration of carbohydrate, fat, protein, essential amino acids (histidine, isoleucine, leucine, lysine, methionine, phenylalanine, threonine, tryptophan and valine), non-essential amino acids (arginine, cystine, glycine, glutamic acid, proline and tyrosine) and vitamins (A, B, C, D, E, K, choline and betaine). Nutrient contents were from a range of foods and are therefore an approximation of insect diets. Where possible, micronutrients were broken down into their subcomponents, for example, individual B vitamins. There were substantial missing data for vitamins C, D, K, choline and betaine (>30% of insect families missing data), which were excluded from analyses.

(3) For each food type, the median concentration of each nutrient across example foods was calculated. Median nutrient values were combined with information on insect diets to calculate a nutrient profile for adults and juveniles of each species (Extended Data Fig. 1). For omnivorous species, nutrients were calculated by averaging across all foods.

(4) Estimates of the nutrients in the diets of each species were calculated by taking the median across adults and juveniles.

(5) Estimates of the dietary nutrients for each family were calculated by taking the median across all species within families. In cases where species-specific diets were not available, we based diets on family-level feeding habits published in books and reviews listed in Supplementary Table 2.

**Standardization of nutrient data.** Reported data on nutrients (carbohydrates, fats, proteins, amino acids and vitamins) were converted to amount per gram. For some foods, however, this was wet weight and for others it was dry weight. We therefore standardized nutrient values to make them comparable across food types. Values were standardized by dividing each nutrient by the mean weight of each food type (for details, see R script 'DataConstruction.R'). The mean was used rather than the sum to avoid the non-independence of percentages when analysing all nutrients together.

**Diversification.** Diversification is typically modelled using three different approaches when species level phylogenies are not available: (1) diversification rates are calculated using clade age and species richness information (also known as 'age richness rate'); (2) raw estimates of species richness are used; and (3) species richness of sister taxa are compared. It has recently been shown that calculating diversification rates should be avoided when examining patterns of biodiversity[85]. We therefore examined patterns of diversification using species richness, including family age as an explanatory variable ('Species richness and obligate symbiosis' in Methods), and by testing differences in species richness between sister lineages. The ages of families were extracted from the time-calibrated Rainford phylogeny[12]. Data on the number of extant species in insect families (species richness) were taken from ref. [12]. It is likely that species richness for all insect families is underestimated as new species are continually being described. Our analyses do not, however, rely on exact absolute numbers of species, but rather that estimates are representative of relative species richness across families, which has been argued as reasonable[46]. Variation in estimates of species richness, for example, due to differences in research effort, are also unlikely to correlate to the proportion of species with obligate symbionts, likely adding noise to our results rather than systematic bias.

## Insect and symbiont phylogenies

**Insects.** Our analyses were conducted using the dated insect phylogeny published in ref. [12]. The tree was constructed using maximum likelihood (RAxML) and dated with a relaxed molecular clock in MrBayes calibrated with 86 fossils[12]. A single consensus tree was published, and therefore our analyses do not account for uncertainty in tree topology or branch lengths. A useful next step will be to incorporate tree uncertainty into analyses, which can be done, for example, by integrating results over a sample of candidate trees. More in-depth details of topological uncertainties and dating methods are available in ref. [12].

Families that lacked data on obligate symbioses were pruned from the tree. There were 23 families for which there were data on obligate symbioses that were not included in ref. [12]. We therefore added these families to the phylogeny at branches corresponding to published sister taxa (Supplementary Table 1) using the bind.tip function in the R package 'phytools'[86] (for details, see R script 'Rainford_adding_tips.R'). Added families were not included for diversification analyses due to uncertainty of the age of these families.

**Symbionts.** We estimated the phylogenetic relationships for bacterial symbionts for which genetic data were available. A ~1,500 bp region of the bacterial 16S rRNA gene downloaded from the SILVA RNA database was aligned with MUSCLE and edited in the alignment software Geneious 8.1.8 (https://www.geneious.com). We constructed a maximum likelihood phylogeny for the bacterial lineages using the on-line PhyML server[87], and the best-fitting models of evolution were estimated using the Aikake information criterion. The symbiont phylogeny was bootstrapped 100 times and rooted to *Thermus thermophilus*, which is basal to all the bacterial lineages presented in this study.

## General statistical methods

Data were analysed using Bayesian phylogenetic mixed models with single (BPMM) and multiple response variables (MR-BPMM), SCM and transition rate models with Markov chain Monte Carlo (MCMC) estimation. In this section we provide general details of modelling approaches, and in 'Specific analyses' in Methods, we outline the details of the analyses conducted. All analyses were performed in R 4.1.3 (ref. [88]), apart from transition rate models that were conducted in BayesTraits V4 (ref. [89]). Continuous response and explanatory variables were Z-transformed before analyses (mean 0, standard deviation 1).

**BPMMs.** Model construction, parameter estimates and assessing significance. To estimate phylogenetic signature, co-evolutionary relationships and ancestral trait values we used BPMMs and MR-BPMMs with MCMC estimation in the R package MCMCglmm[90]. Obligate symbiosis was analysed as the number of species with and without obligate symbionts (proportion) within each family using a binomial error distribution with a logit link function. Analysing obligate symbiosis in this way enables variation in the number of species examined for obligate symbionts across insect families to be accounted for. Nutrient concentrations were Z-transformed before analysis and modelled as Gaussian response variables and species richness was modelled using a Poisson error distribution with log link function.

The global intercept was removed from MR-BPMMs to allow trait specific intercepts to be estimated. Parameter estimates from models are presented as posterior modes with 95% CIs, together with approximate P values (pMCMC)[90].

The non-independence of data resulting from phylogenetic relatedness between insect hosts and phylogenetic relatedness between symbiont lineages was modelled using random effects. For phylogenetic effects we fitted a variance–covariance matrix constructed from the insect and bacteria phylogenies. Phylogenetic and residual correlations between traits were calculated using the variance and covariance estimates from unstructured phylogenetic and residual variance–covariance matrices. We estimated the amount of variation in response variables explained by random effects (RE), including phylogenetic effects, as the intraclass correlation coefficient on the latent scale estimated as

$$V_i / V_{RE} + V_e$$

where $V_i$ is the focal random effect, $V_{RE}$ is the sum of all random effects and $V_e$ is the residual variance on the latent scale. For binomial error distributions $V_e$ was calculated as the observed residual variance plus the variance associated with the link function (logit = $pi^{2/3}$; for discussion, see refs. [91,92]).

**Prior settings.** For random effects we began prior selection by assessing model convergence using inverse-gamma priors ($V = diag(n)$, nu $= n - 1 + 0.002$, where $n$ was equivalent to the number of response traits). If the mixing properties of the MCMC chain were poor, which was often the case for binomial response variables, we examined parameter expanded priors ($V = diag(n)$, nu $= n - 1$, alpha.mu $= rep(0, n)$, alpha.V $= diag(n) \times 25^2$) (ref. [92]). For fixed effects the default priors in MCMCglmm (independent normal priors with zero mean and large variance ($10^{10}$)) were used apart from in models with binomial response variables where a prior of mu $= 0$, $V = \sigma^2$ units $+ \pi^{2/3}$ was specified. This is approximately flat on the probability scale when a logit link function is defined[90,93], and in all cases improved the mixing of chains. The final prior settings used for each analysis are specified in the supplementary R code (R script 'Analyses.R').

**Model convergence.** Models were run for 2 million iterations with a burn-in of 1 million iterations, and chains were sampled every 1,000 iterations. We examined the convergence of models by repeating each analysis three times and examining the correspondence between chains using the R package 'coda'[94] in the following ways: (1) visually inspecting the traces of the MCMC posterior estimates and their overlap; (2) calculating the autocorrelation and effective sample size of the posterior distribution of each chain; and (3) using Gelman and Rubin's convergence diagnostic test that compares within- and between-chain variance using a potential scale reduction factor. Potential scale reduction factor values substantially higher than 1.1 indicate chains with poor convergence properties. For convergence checking, see R script 'ModelCheckingCombining.R'.

### Transition rate models

**SCM.** SCM was used to estimate ancestral states of obligate symbiosis and feeding niches across the insect phylogeny in the R package 'phytools'[86]. In brief, this approach calculates the conditional likelihood that each ancestral node is in a given state depending on the estimated transition rate matrix ($Q$) between states and the length of the branch associated with that node. On the basis of these conditional likelihoods, ancestral states at each node are stochastically simulated and used in combination with observations at the tips to reconstruct a character history along each branch. Each character history is simulated using a continuous-time Markov chain where changes between states and the time spent in each state are modelled as a Poisson process (for more details, see ref. 95).

**BayesTraits.** BayesTraits V4 was used to reconstruct ancestral values of feeding niches and examine co-evolutionary relationships between B vitamins and obligate symbiosis (for details, see 'Specific analyses' in Methods). We used hyper priors where values are drawn from a uniform distribution with a range 0 to 10 to seed the mean and variance of an exponential prior to reduce uncertainty over prior selection[89]. We ran each model three times for a total of 11,000,000 iterations with a burn-in of 1,000,000 iterations and sampled every 1000 iterations. We examined the convergence of models in the same way as 'Transition rate models' in Methods.

Bayes factors (2(log marginal likelihood of complex model − log marginal likelihood of simple model)) were used to test if models where traits were allowed to co-evolve provided a better fit to the data than models that assumed independent evolution. To calculate the log marginal likelihood, we used the stepping-stones procedure as described in the BayesTraits V4 manual where 100 stones were run for 1,000 iterations each. Bayes factors over 2 are considered to offer positive evidence, over 5 strong evidence and over 10 very strong evidence[89]. To test whether transition rates were significantly different from each other, we calculated the posterior mode, 95% CIs and pMCMC value of the posterior distribution of differences between transition rates.

### Specific analyses

**Evolutionary history of obligate symbioses.** The number of origins of obligate symbiosis. The probability of each node in the insect phylogeny having an obligate symbiont was estimated using a BPMM with the proportion of species in families with obligate symbionts as a response variable. The posterior probability of each node having obligate symbionts was estimated using the 'predict' function in MCMCglmm and nodes with a posterior probability greater than 0.5 were classified as 'obligate'. We found support for 16 origins and 8 losses of obligate symbiosis.

We examined the robustness of the estimated number of origins and losses of obligate symbiosis using SCM. As SCM requires categorical states (obligate versus non-obligate), insect families ($n$ = 402) were classified as having an obligate symbiosis where over 50% species in families had symbionts. Data on obligate symbiosis were used to build

1,000 stochastic character maps across the insect phylogeny using an all-rates different $Q$ matrix estimated with MCMC. Nodes were classified as having obligate symbionts if the proportion of the 1,000 stochastic character maps was above 50%. We found high correspondence between the SCM and BPMM analyses: 98% of ancestral states had the same predicted state of obligate symbiosis across analyses, indicating that our results are robust to different statistical approaches (Supplementary Table 20).

**Estimating ancestral feeding niches.** Ancestral feeding niches were estimated using SCM. Data on the feeding niches of insect families ($n$ = 402) were used to build 1,000 stochastic character maps across the insect phylogeny using an equal-rates $Q$ matrix estimated with MCMC. Ancestral estimates of nodes were assigned according to the feeding niche with the highest proportion of the 1,000 stochastic character maps. Transitions between feeding niches were identified where ancestral and descendant nodes were in different states (Supplementary Table 5).

We examined the robustness of ancestral feeding niches estimated using SCM by performing a second set of ancestral reconstructions. We used the MULTISTATE module in BayesTraits V4 with reversible-jump MCMC estimation and compared them with the estimates gained by SCM. There was very good correspondence between SCM and MULTISTATE, with 94% of ancestral nodes predicted to have the same feeding niche (Supplementary Table 20).

**Rates of obligate symbiosis across different feeding niches.** The probability that insects with different feeding niches had obligate symbionts was modelled using a BPMM with the proportion of species in families with obligate symbionts as a response variable. The feeding niche of each family was fitted as an eight-level fixed effect (Supplementary Table 6). To determine if rates of obligate symbiosis were significantly different across niches, we calculated the pairwise differences between niches and examined if the 95% CIs spanned 0 (Supplementary Table 6).

**Nutritional deficiencies and the evolution of obligate symbiosis.** Phylogenetic correlations between obligate symbiosis and nutrients. The phylogenetic correlations between obligate symbiosis and nutrients within diets was estimated using a MR-BPMM with the proportion of species in families with obligate symbionts and concentrations of carbohydrate, fat, protein, essential amino acids (sum of histidine, isoleucine, leucine, lysine, methionine, phenylalanine, threonine, tryptophan and valine), non-essential amino acids (sum of arginine, cystine, glycine, glutamic acid, proline and tyrosine), vitamin A, vitamin B (sum of individual B vitamins) and vitamin E as response variables (Supplementary Table 7). For analyses of individual amino acids, see 'Obligate symbiosis and types of amino acids' in Methods and Supplementary Table 29. There were missing values for some nutrients for some families in our dataset (Supplementary Table 4). In BPMMs, missing data are permitted in response variables and are predicted with an accuracy relative to the phylogenetic signature in traits and magnitude of trait correlations. This can enable missing values to be predicted with high accuracy[93,96]. All nutrients had high phylogenetic signature (phylogenetic heritability (phylo H$^2$) 0.71–0.96; Supplementary Table 7), and therefore nutrients with up to 30% missing values were included in analyses (vitamin A = 21%, vitamin E = 29%; all other nutrients had less than 5% missing data).

**Differences in the nutrient contents of feeding niches.** Differences in the nutritional composition of feeding niches were estimated using a MR-BPMM with carbohydrate, fat, protein, essential amino acids, non-essential amino acids, vitamin A, vitamin B and vitamin E as response variables. To test if nutrient levels of each feeding niche differed from background rates, we sequentially fitted two-level factors

of focal feeding niche versus all other niches as a fixed effect (Supplementary Table 9).

**Phylogenetic correlations between obligate symbiosis and B vitamins.** The phylogenetic correlations between obligate symbiosis and individual B vitamins were analysed using a MR-BPMM with the proportion of species in families with obligate symbionts and concentrations of vitamins B5 and B9 as response variables (Supplementary Table 8). Data on vitamins B7 and B12 were not analysed as there were missing values for over 30% of insect families. Data on B1, B2 and B3 were highly correlated with vitamin B5 levels ($r > 0.9$), but there were more data on vitamin B5. As a result, only vitamin B5 was analysed, but it is worth noting that B1, B2 and B3 may contribute to any association with obligate symbiosis.

**Nutrients and the gains and losses of obligate symbiosis.** Ancestral B vitamins in relation to the gains and losses of symbionts. We examined how levels of B5 and B9 vitamins differed between the ancestors of families with and without obligate symbionts using a two-step approach: first, we used the output of the model in 'The number of origins of obligate symbiosis' in Methods to classify nodes as: (1) non-obligate node with non-obligate descendants (Non to Non); (2) non-obligate node with at least one obligate descendant (Non to Ob); (3) obligate node with obligate descendants (Ob to Ob); and (4) obligate node with at least one non-obligate descendant (Ob to Non). Second, node classifications were entered as a four-level fixed factor in an MR-BPMM with B5 and B9 vitamin concentrations as response variables (Supplementary Table 10). B5 and B9 vitamin levels were compared across nodes that preceded the origin (1 versus 2), maintenance (1 versus 3) and loss of obligate symbiosis (3 versus 4). Unstructured phylogenetic and residual variance–covariance matrices were fitted as random effects with the phylogenetic covariance matrix being linked to node labels.

To account for uncertainty in our node classifications, we repeated the analysis 100 times, each time reclassifying nodes as 'obligate' or 'non-obligate' by resampling from the posterior distribution of model 4.1.1. Posterior samples from across the 100 models were combined. Each model was run for 1,100,000 iterations with a burn-in of 1,000,000 iterations and a thinning interval of 10,000 samples, which across the re-samplings resulted in 1,000 posterior samples (100 re-samplings × 10 samples per resampling). To verify that our estimates of B vitamins were robust to rate shifts in B vitamins across the tree, we compared our BPMM estimates with phylogenetic ridge regression models that allowed for rate shifts[97]. We found high correspondence between estimates from BPMM and phylogenetic ridge regression models and between BPMM estimates and actual B vitamin values for insect families (Extended Data Fig. 4).

**Transition rates between obligate symbiosis and B vitamins.** We tested if models that allowed for co-evolution between obligate symbiosis and B5 and B9 vitamins better explained our data than models that assumed independent evolution of each trait using transition rate models in BayesTraits. Co-evolution was modelled using an all-rates different $Q$ matrix with separate models run for B5 and B9 vitamins. For transition rate models, only binary classifications can be modelled. Insect families were therefore classified as obligate (>50% of species within families have obligate symbionts) and non-obligate (<50% of species within families have obligate symbionts), and having high and low B5 and B9 vitamins. For B-vitamin classifications, two different cut-offs were analysed to establish the sensitivity of our results to different thresholds: above and below the 25% and 50% quantile for high and low B vitamins, respectively (Supplementary Table 11). It was not necessary to examine the sensitivity of our results to the classification of obligate symbiosis as 96% of 402 insect families had 100% of species with or without obligate symbionts. Reversible-jump MCMC models were also run to further test if transitions occurred (percentage of models where transition rate was above 0) and if transition were different from each other (percentage of models where transition rates were assigned to different rate categories).

**Nutrient deficiencies and host–symbiont co-specialization.** Host–symbiont interactions and the evolution of obligate symbiosis. To examine how obligate symbiosis has been influenced by the co-evolutionary history between insects and bacteria, we constructed a dataset of pairwise combinations between all insect families and all symbionts (excluding vertically transmitted symbionts). For each combination, the number of insect species within a family with a particular obligate symbiont versus the number of species sampled without that symbiont was calculated and analysed using a BPMM with a binomial error distribution. This enabled differences in the sampling effort across different insect–bacteria associations to be accounted for. Whether symbionts were intra- and extracellular was included as a two-level fixed effect. Three different variance–covariance matrices were fitted as random effects to quantify the amount of variation in obligate symbiosis explained by: (1) insect hosts independent of their phylogenetic history ('h'), for example, certain hosts are more likely to form obligate relationships than others; (2) insect hosts phylogenetic history ('[h]'), for example, certain host lineages are more likely to form obligate relationships than others; and (3) phylogenetic interactions between hosts and symbionts ('[hs]'), for example, particular host phylogenetic lineages are more likely to form obligate symbioses with particular bacterial phylogenetic lineages (Supplementary Table 14). For details of model fitting, see ref. [98] (the variance–covariance matrices outlined in ref. [98] that relate to the number of hosts symbionts associate with were not fitted in models as each symbiont lineage was found in only one insect family).

To further examine whether phylogenetically related lineages of bacteria are more likely to form obligate symbioses with phylogenetically related lineages of insects, we used parafit in the R package 'ape' (Supplementary Table 15). This tests the correlation between host- and symbiont-shared branch lengths against a randomized distribution generated from 1,000 permutations of the data[99].

**Host–symbiont interactions and levels of B vitamins in insect diets.** To test if specific lineages of symbiotic bacteria specialize in providing B5 and B9 vitamins to hosts, we used the same BPMM approach described in 'Host–symbiont interactions and the evolution of obligate symbiosis' in Methods. We estimated variation in levels of B vitamins (Gaussian responses) explained by h, [h], [s] and [hs]. Separate models were run for B5 and B9 vitamins, and data were restricted to combinations of hosts and bacteria that formed obligate symbioses (Supplementary Tables 12 and 13).

**Obligate symbiosis and diversification.** Species richness and obligate symbiosis. The relationship between obligate symbioses and species richness was estimated using an MR-BPMM with the proportion of species in families with obligate symbionts and species richness as response variables. Family age and whether insect families were holo- or hemimetabolous (two-level factor previously shown to influence diversification rates[12]) were fitted as fixed effects (Supplementary Table 16).

**Species richness across symbiont-associated feeding niches.** To test if species richness differed between insect families with and without obligate symbionts occupying different feeding niches, a BPMM with species richness as a response variable was used. Each insect family was classified according to whether it had obligate symbionts (>50% of species with symbionts) and its feeding niche (11-level factor), which was fitted as a fixed effect along with family age and holo–hemi metabolism (Supplementary Table 17). To test if species richness of each obligate symbiosis-feeding niche combination differed from background levels, we re-ran models fitting two-level fixed factor of

the focal obligate symbiosis-feeding niche combination versus all other data (Supplementary Table 17).

**Species richness and obligate symbiosis among sister taxa.** Sister comparisons were extracted from ref. 12 phylogeny using the 'extract_sisters' function in the R package 'diverge'[100] (Supplementary Table 30). We analysed data on all sister comparisons ($n = 123$) using a BPMM to test if feeding niche and obligate symbiosis was related to species richness at a finer taxonomic scale than in 'Species richness and obligate symbiosis' in Methods. Species richness was the response variable and the percentage of species within families with obligate symbionts (% logit transformed), age of the sister comparison (million years) and feeding niche were included as fixed effects. The non-independence of data from sister taxa was modelled by including sister pair identity as a random effect and the non-independence of sister pairs across the phylogeny was modelled by including node identities of each sister comparison linked to the phylogeny as a random effect (Supplementary Table 18).

**Species richness changes and the evolution of obligate symbiosis.** To further examine if the evolution of obligate symbiosis is associated with increased species richness, we restricted our sister comparison dataset to cases where sister families differed in the percentage of species with obligate symbionts. Differences in species richness was calculated as the $\log_2$ of the ratio of species richness between sister taxa and analysed as a Gaussian response variable using a BPMM. The difference in the percentage of species with obligate symbionts between sister taxa and pair age were fitted as fixed effects. Node identity of sister comparisons, linked to the phylogeny, was included as a random effect (Supplementary Table 19).

**Sensitivity analyses.** We tested the robustness of our conclusions to several underlying data assumptions. These sensitivity analyses provided quantitatively similar results to our main analysis (Supplementary Tables 21–27).

**Removing families added to the Rainford tree.** There were 23 families within our obligate symbiont dataset that were not represented in the Rainford phylogeny that were added to the phylogeny ('Insect and symbiont phylogenies' in Methods). To examine the robustness of our results to including these families, we re-ran the analyses detailed in 'Phylogenetic correlations between obligate symbiosis and nutrients' in Methods (Supplementary Table 21) with the 23 families excluded.

**Including only bacterial symbionts.** Bacteria made up the vast majority of obligate symbionts (79 out of 84 insect families had bacterial symbionts, 94%). To verify that our results were not explained by a few outlying eukaryotic symbionts, we re-ran the analyses detailed in 'Phylogenetic correlations between obligate symbiosis and nutrients' in Methods including only insect families with bacterial symbionts ($n_{families} = 395$; Supplementary Table 22).

**Removing co-occurring obligate symbionts.** There were 112 unique host–bacterial symbiont combinations. Of these 49% ($n = 55$) had multiple co-occurring symbionts. It is possible that any signature of bacteria specializing in B5 and B9 vitamin production is obscured by the presence of co-residing obligate symbionts that may change nutrient provisioning roles. We therefore repeated the analyses in 'Host symbiont interactions and levels of B vitamins in insect diets' in Methods section after removing hosts that had multiple co-occurring symbionts (Supplementary Table 23).

**Excluding data from microscope studies.** Out of the 402 insect families included in our analyses, 260 were inferred to lack obligate symbionts from microscopy by Buchner and colleagues that showed an absence of specialized symbiont organs. Insects in the orders Ephemeroptera, Plecoptera, Odonata, Neuroptera, Orthoptera and Lepidoptera, superfamily Tenthredinoidea and subclade Aculeata (excluding Formicidae) all lacked bacteriocytes and in general do not depend on endosymbionts for survival[23]. We tested the sensitivity of our results to including this inferred data by re-running analyses outlined in 4.2.1, 4.2.3 and 4.5.2 on data where obligate symbiosis had only been directly studied (for more details, see 'Criteria for assessing obligate symbiosis' in Methods). This also tested the robustness of our results to removing Lepidoptera that contribute many tips to the phylogenetic tree, lack obligate symbionts and are predominately herbivorous.

**Obligate symbiosis and types of amino acids.** In 4.2.1 we analysed the sum of essential and non-essential amino acids. All amino acids were highly correlated ($r > 0.8$. Extended Data Fig. 3), but to further check if minor differences in the concentration of each amino acid influenced the relationship with obligate symbiosis we ran a second set of analyses. The phylogenetic correlation between obligate symbiosis and each amino acid was examined using a series of MR-BPMMs. The proportion of species with obligate symbionts in families and the concentration of each amino acid were response variables in each analysis (nine different bivariate models for essential amino acids and six different bivariate models of non-essential amino acids; Supplementary Table 29).

**Removing families with and without obligate symbionts.** Out of the 402 insect families, 15 families contained species with and without obligate symbionts (3.7%). For transition rate analyses, these mixed families had to be classified as having obligate symbionts or not. We tested the sensitivity of our results to this classification by removing these families and re-running the analyses described in 'Transition rates between obligate symbiosis and B vitamins' in Methods.

The conclusions from the sensitivity analyses were quantitatively similar to our main analyses (Supplementary Tables 21–29).

### Reporting summary

Further information on research design is available in the Nature Portfolio Reporting Summary linked to this article.

## Data availability

All data are provided in Supplementary Tables 1–4. Full citations of references in supplementary tables are given in the method references[7,12,16,101–404]. All supplementary tables are available at the Open Science Framework (osf.io project number TYK7C; https://doi.org/10.17605/OSF.IO/TYK7C)[405]. Source data are provided with this paper.

## Code availability

R code, BayesTraits code and analysis results are available at the Open Science Framework (osf.io project number TYK7C; https://doi.org/10.17605/OSF.IO/TYK7C)[405].

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

## Acknowledgements

We are very grateful to funding from the Knut and Alice Wallenberg Foundation (Wallenberg Academy fellowship 2018.0138 to C.K.C.), the Swedish Research Council (grant number 2017-03880 to C.K.C.), the European Research Council (grant numbers 335542 to E.T.K. and 834164 to S.A.W.), the Ammodo Foundation (funding to E.T.K.) and Natural Environmental Research Council (grant number NE/M018016/1 to L.H.).

## Author contributions

Conceptualization: C.K.C., A.V.P., J.E., E.T.K., S.A.W and L.M.H. Methodology: C.K.C., A.V.P., J.E., M.K., R.J., E.T.K., S.A.W. and L.M.H. Investigation: C.K.C., A.V.P., J.E., M.K., R.J., E.T.K., S.A.W. and L.M.H. Visualization: C.K.C., A.V.P., E.T.K., S.A.W. and L.M.H. Funding acquisition: C.K.C., J.E., E.T.K., S.A.W. and L.M.H. Project administration: J.E., E.T.K. and L.M.H. Supervision: J.E., E.T.K. and L.M.H. Writing—original draft: C.K.C., E.T.K., S.A.W. and L.M.H. Writing—review and editing: C.K.C., A.V.P., J.E., M.K., R.J., E.T.K., S.A.W. and L.M.H.

## Funding

## Competing interests

The authors declare no competing interests.

## Additional information

**Extended data** is available for this paper at https://doi.org/10.1038/s41559-023-02058-0.

**Correspondence and requests for materials** should be addressed to Charlie K. Cornwallis or Lee M. Henry.

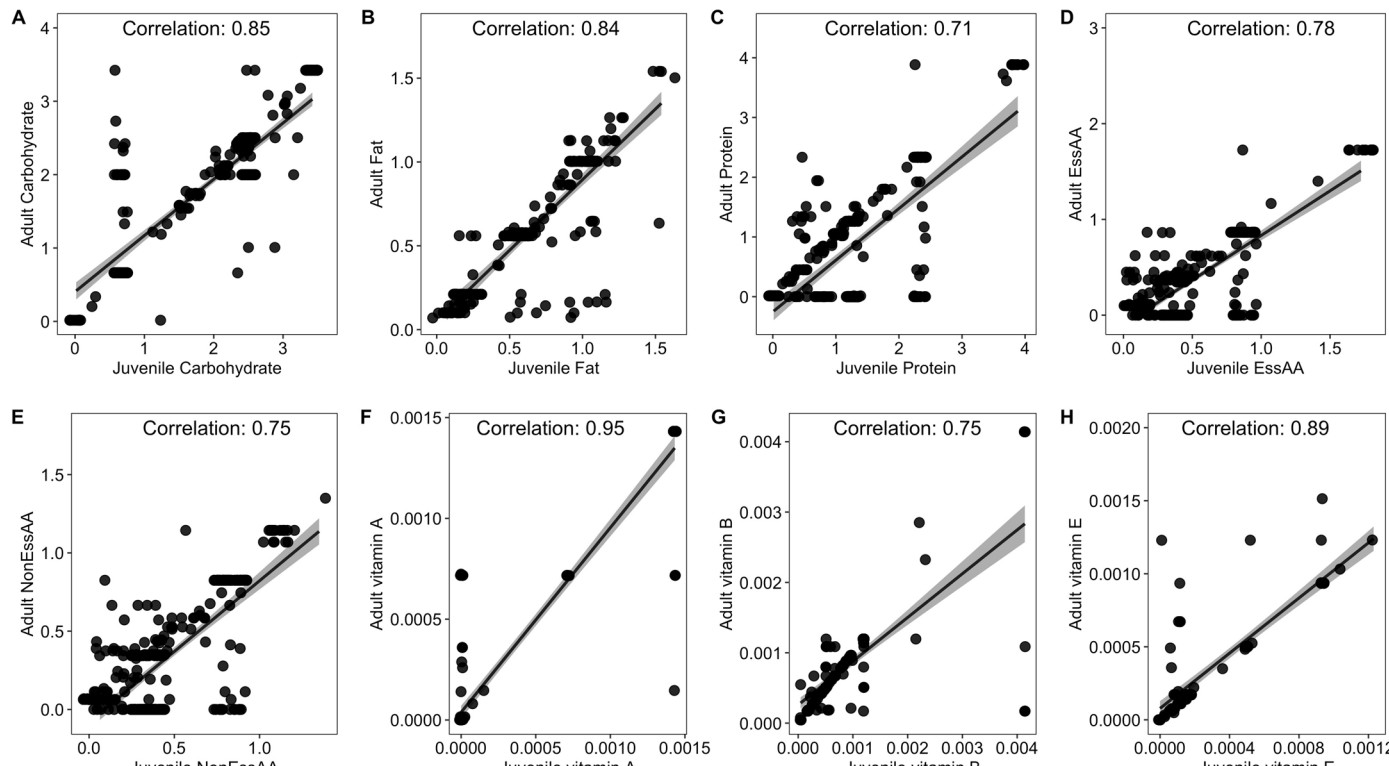

**Extended Data Fig. 1 | The correspondence between the composition of adult and juvenile diets across insect families.** The relationship between adult and juvenile dietary concentrations of (A) Carbohydrate, (B) Fat, (C) Protein, (D) Essential Amino Acids (EssAA), (E) Non-Essential Amino Acids (NonEssAA), (F) Vitamin A, (G) Vitamin B and (H) vitamin E are presented. Each point is one insect family and the lines are linear regressions with 95% confidence intervals (shaded bands). Pearson correlation coefficients ('Correlation') are presented for each nutrient.

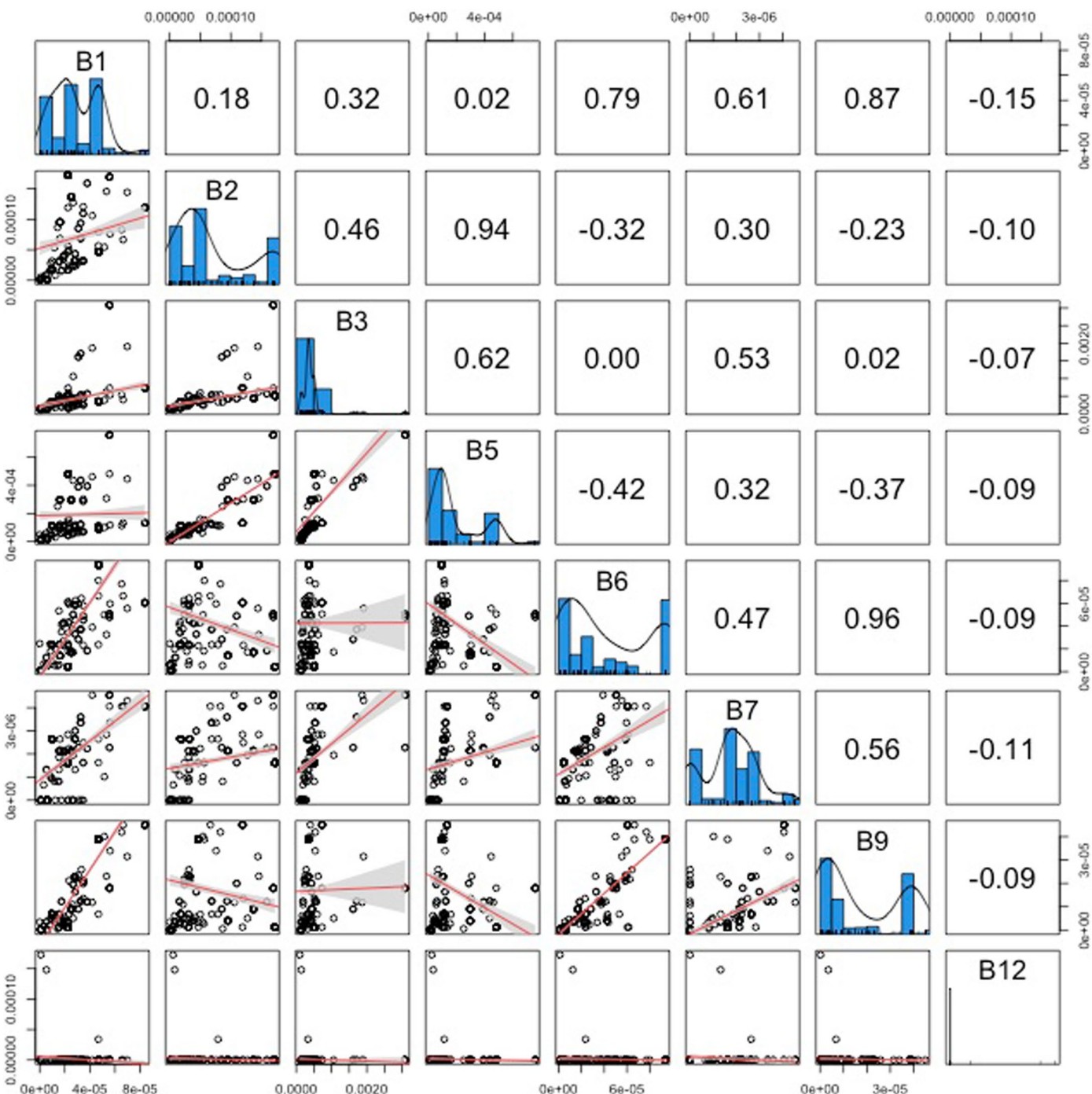

**Extended Data Fig. 2 | The relationship between different B vitamins across diets.** The frequency distributions of the different B vitamins are plotted in the diagonal panels. Scatter plots of the correlation between different B vitamins are plotted in the panels below the diagonal, where each point represents one insect family and lines represent linear regressions with 95% confidence intervals (shaded bands). Pearson correlation coefficients are given in the panels above the diagonal.

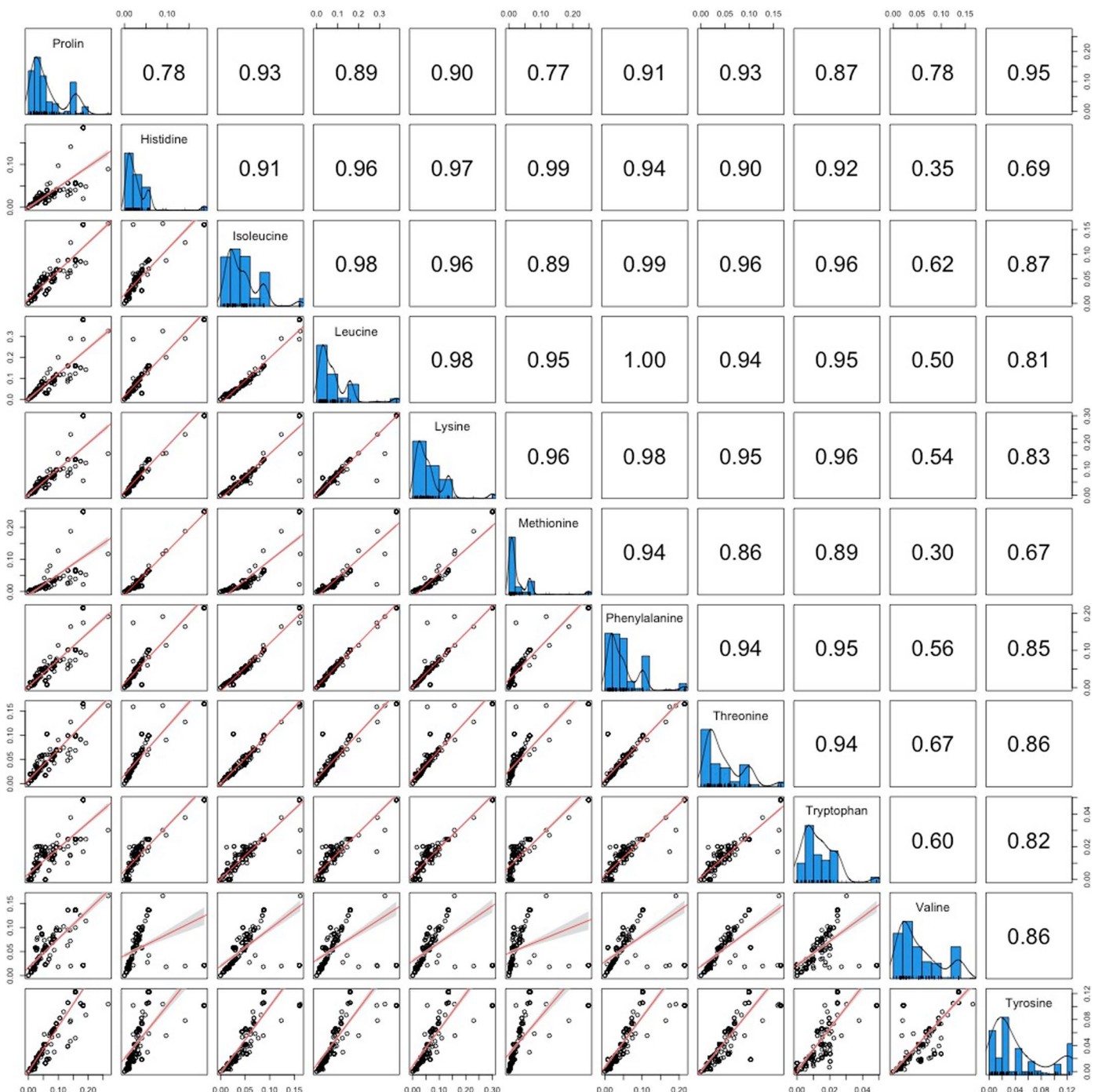

**Extended Data Fig. 3 | The relationship between different amino acids across diets.** The frequency distributions of the different amino acids are plotted in the diagonal panels. Scatter plots of the correlation between different amino acids are plotted in the panels below the diagonal, where each point represents one insect family and lines represent linear regressions with 95% confidence intervals (shaded bands). Pearson correlation coefficients are plotted above the diagonal.

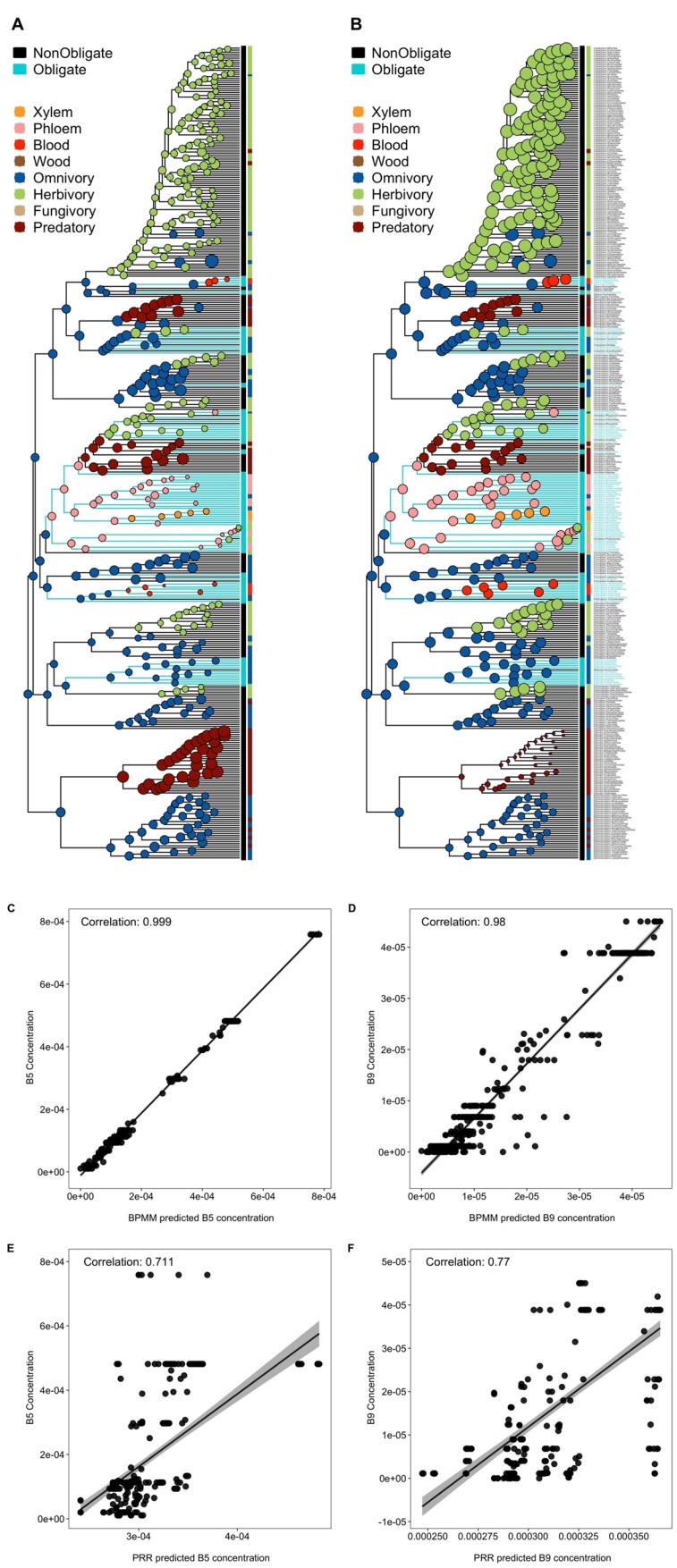

**Extended Data Fig. 4 | See next page for caption.**

**Extended Data Fig. 4 | Reconstruction of ancestral levels of B vitamins, obligate symbiosis and feeding niches across 402 insect families.** Ancestral concentrations of (A) B5 and (B) B9 vitamins were estimated using BPMMs and are shown by the size of circles at each node. Turquoise tips and branches indicate obligate symbionts and different coloured dots represent different feeding niches. Ancestral feeding niches were estimated using SCM and states of obligate symbiosis were estimated using a BPMM (Supplementary Table 5). There was greater correspondence between predictions from BPMMs and raw concentrations of (C) B5 and (D) B9 vitamins than there was for predictions from phylogenetic ridge regressions (PRR, E-F), which allowed for rate shifts in B vitamins across the phylogeny (E-F). In C-F lines represent linear regressions with 95% confidence intervals (shaded bands).

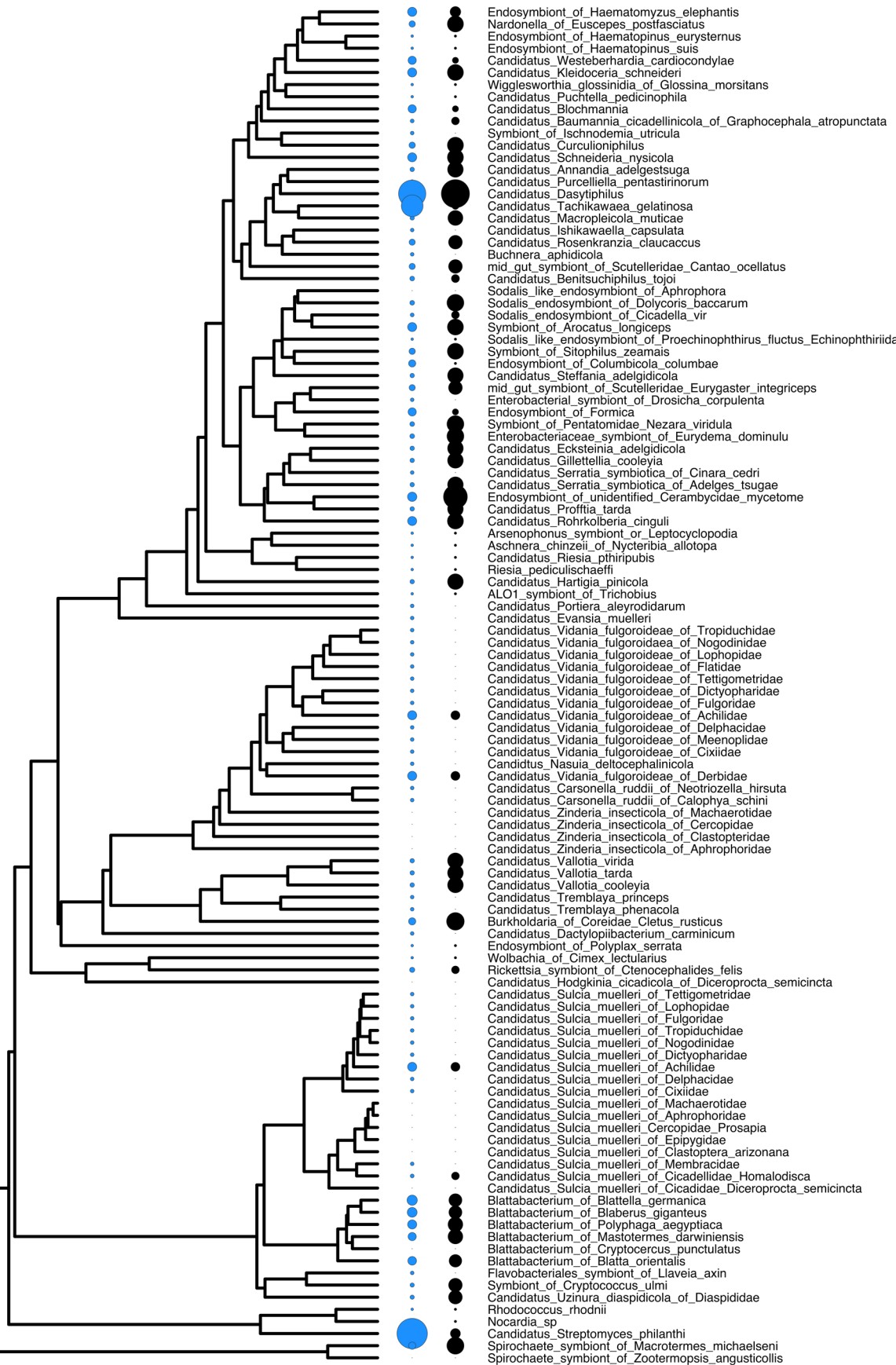

**Extended Data Fig. 5 | Levels of B vitamins in host diets in relation to symbiont phylogenetic history.** B5 vitamins are presented in blue and B9 vitamins are in black with the size of the circles indicating the amount of B5 and B9 vitamins in the diets of insect hosts.

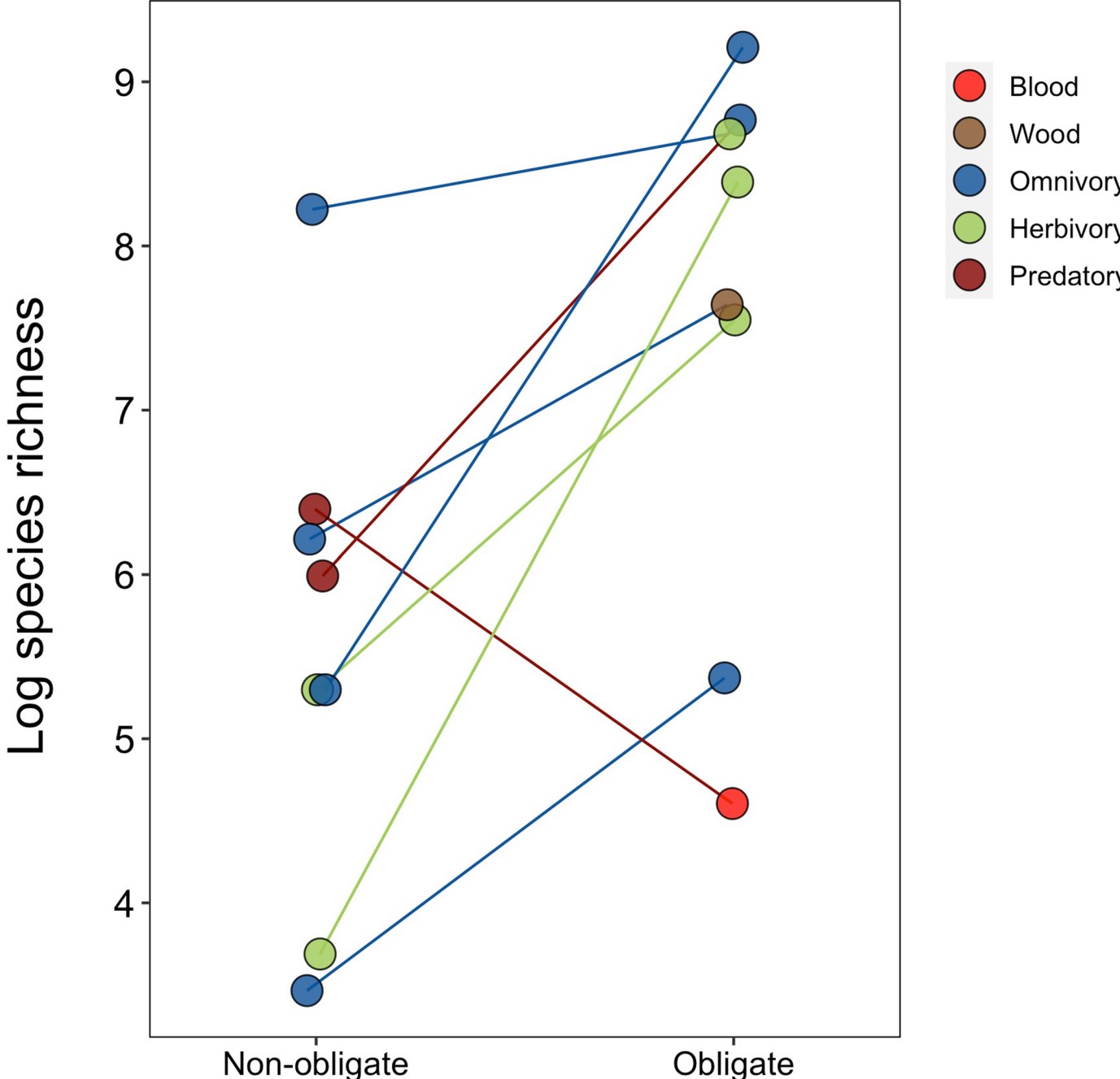

**Extended Data Fig. 6 | Sister comparisons of families with and without obligate symbionts.** Obligate symbiosis was associated with increased species richness in all comparisons apart from one, where there has been a switch to blood feeding from predation. Species richness was measured as the number of species within families and is presented on the natural logarithmic scale.

There were 13 sister families with different percentages of species with obligate symbionts. Of these, 5 comparisons had small differences in rates of obligate symbiosis: <35% difference in percentage of species with obligate symbionts (not plotted, see Fig. 4 for all 13 comparisons).

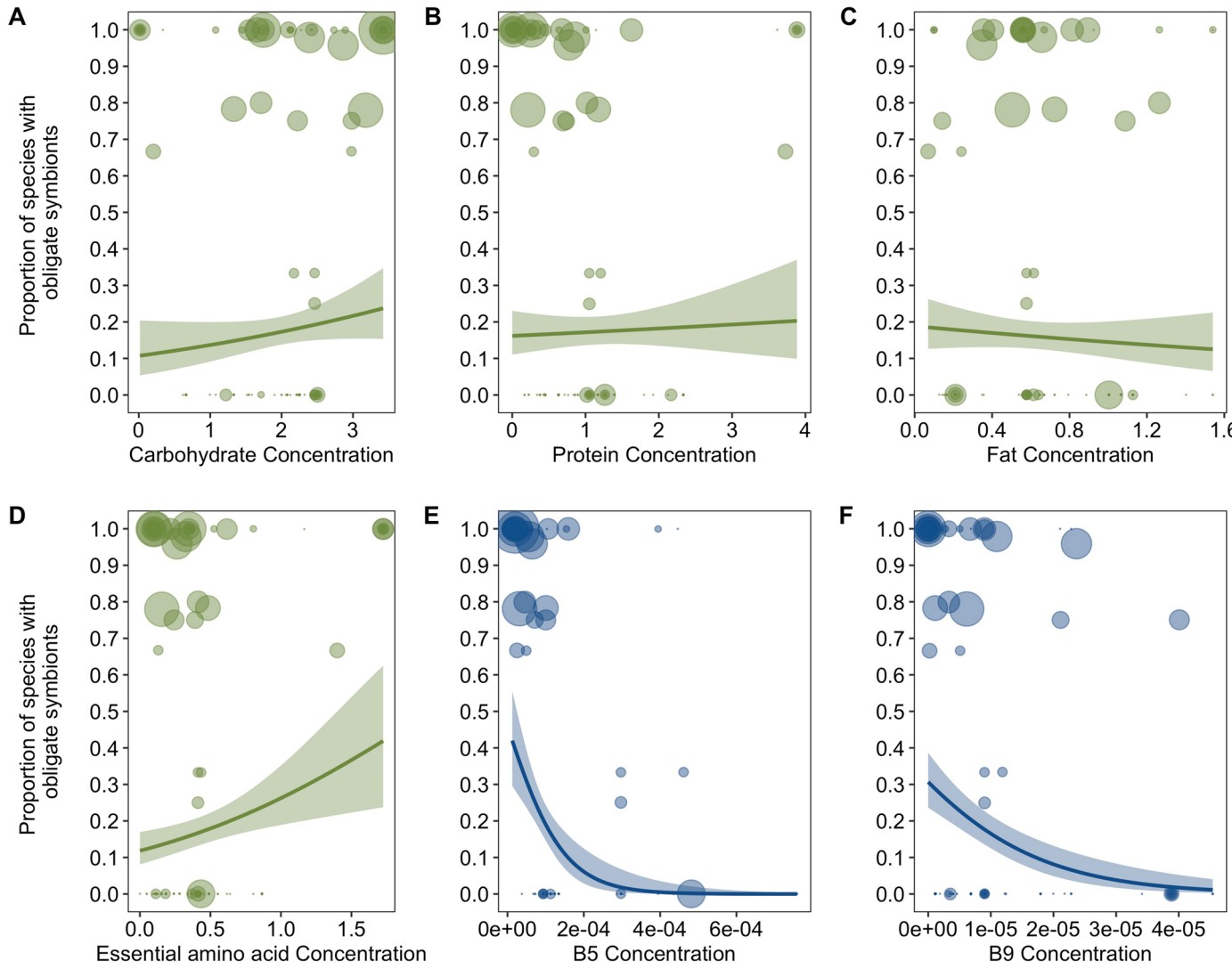

**Extended Data Fig. 7 | The evolution of obligate symbioses in relation to nutrient deficiencies after removing families that were added to the Rainford tree.** The proportion of species with obligate symbionts in families is plotted in relation to dietary concentrations of (A) Carbohydrate, (B) Protein, (C) Fat (D) Essential Amino Acids, (E) Vitamin B5 and (F) vitamin B9. Obligate symbioses was negatively phylogenetically correlated to concentrations of B vitamins (B phylo *r* (CI) = −0.36 (−0.53, −0.11), pMCMC = 0.008. Supplementary Table 21). Values of macro- and micro-nutrients are standardized amounts per gram ('Nutrient data' in Methods). The size of points represents the mean number of host species (log transformed) examined for obligate symbionts per family. Lines represent logistic regressions with 95% confidence intervals (shaded bands) plotted for illustrative purposes.

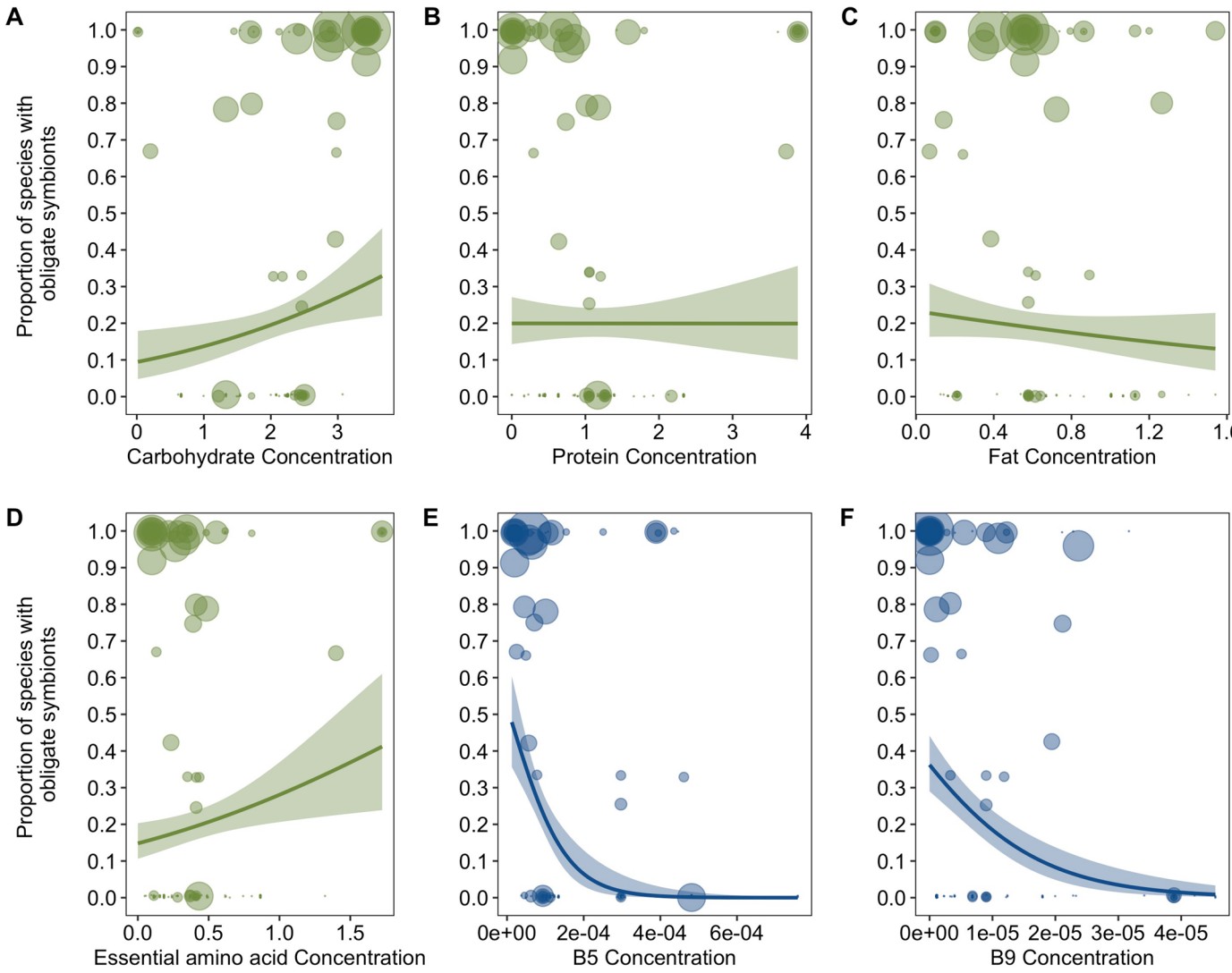

**Extended Data Fig. 8 | The evolution of obligate symbioses in relation to nutrient deficiencies including only bacterial symbionts.** The proportion of species with obligate symbionts in families is plotted in relation to dietary concentrations of (A) Carbohydrate, (B) Protein, (C) Fat (D) Essential Amino Acids, (E) Vitamin B5 and (F) vitamin B9. Obligate symbioses was negatively phylogenetically correlated to concentrations of B vitamins (B phylo *r*

(CI) = −0.35 (−0.49, −0.06), pMCMC = 0.008. Supplementary Table 22). Values of macro- and micro-nutrients are standardized amounts per gram ('Nutrient data' in Methods). The size of points represents the mean number of host species (log transformed) examined for obligate symbionts per family. Lines represent logistic regressions with 95% confidence intervals (shaded bands) plotted for illustrative purposes.

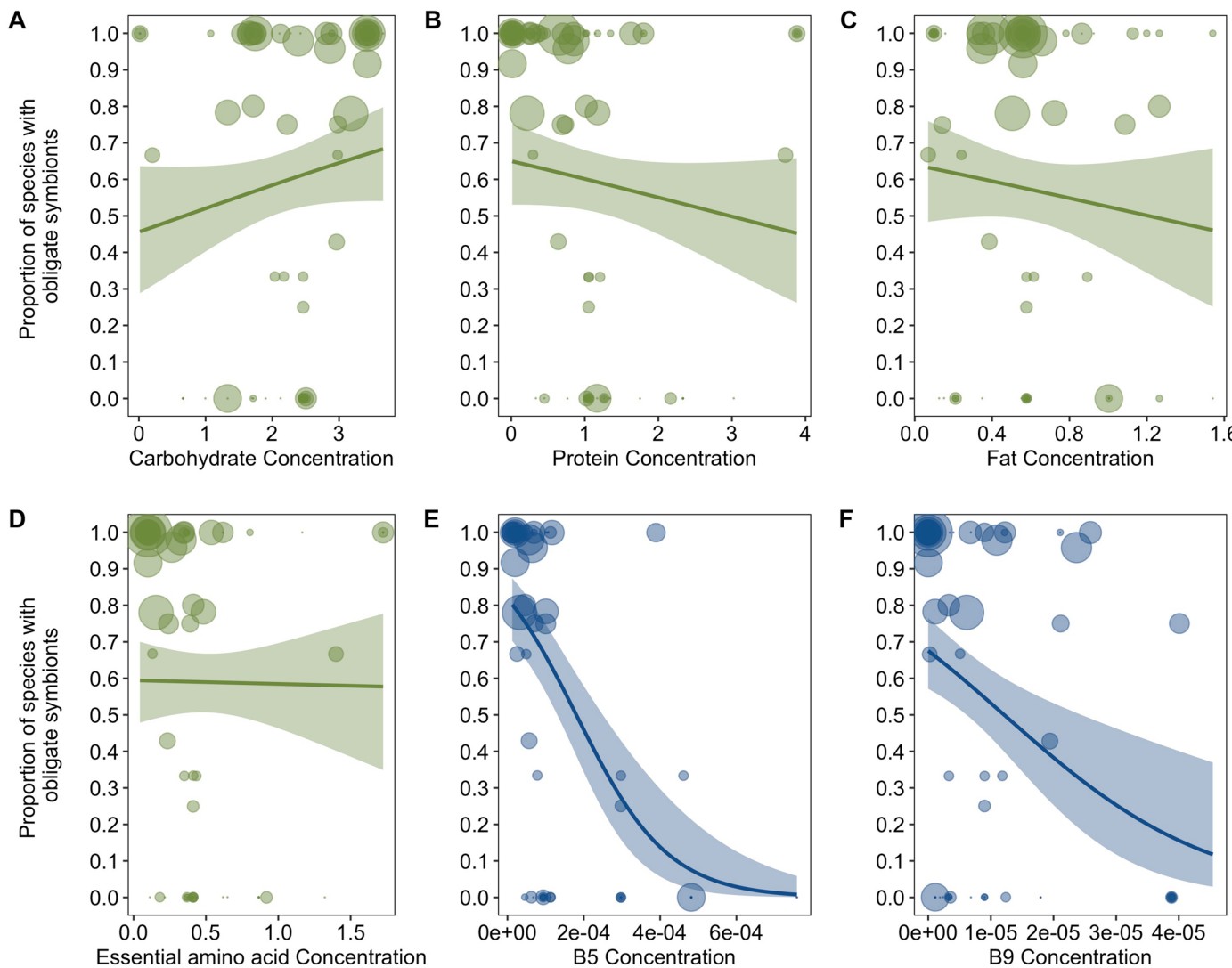

**Extended Data Fig. 9 | The evolution of obligate symbioses in relation to nutrient deficiencies after restricting data to families where obligate symbioses had been directly studied.** The proportion of species with obligate symbionts in families is plotted in relation to dietary concentrations of (A) Carbohydrate, (B) Protein, (C) Fat (D) Essential Amino Acids, (E) Vitamin B5 and (F) vitamin B9. Obligate symbioses was negatively phylogenetically correlated to concentrations of B vitamins (B phylo *r* (CI) = −0.58 (−0.79, −0.04), pMCMC = 0.05. Supplementary Table 25). Values of macro- and micro-nutrients are standardized amounts per gram ('Nutrient data' in Methods). The size of points represents the mean number of host species (log transformed) examined for obligate symbionts per family. Lines represent logistic regressions with 95% confidence intervals (shaded bands) plotted for illustrative purposes.

# Reporting Summary

## Statistics

For all statistical analyses, confirm that the following items are present in the figure legend, table legend, main text, or Methods section.

| n/a | Confirmed | |
|---|---|---|
| ☐ | ☒ | The exact sample size (*n*) for each experimental group/condition, given as a discrete number and unit of measurement |
| ☐ | ☒ | A statement on whether measurements were taken from distinct samples or whether the same sample was measured repeatedly |
| ☐ | ☒ | The statistical test(s) used AND whether they are one- or two-sided <br> *Only common tests should be described solely by name; describe more complex techniques in the Methods section.* |
| ☐ | ☒ | A description of all covariates tested |
| ☐ | ☒ | A description of any assumptions or corrections, such as tests of normality and adjustment for multiple comparisons |
| ☐ | ☒ | A full description of the statistical parameters including central tendency (e.g. means) or other basic estimates (e.g. regression coefficient) AND variation (e.g. standard deviation) or associated estimates of uncertainty (e.g. confidence intervals) |
| ☐ | ☒ | For null hypothesis testing, the test statistic (e.g. $F$, $t$, $r$) with confidence intervals, effect sizes, degrees of freedom and $P$ value noted <br> *Give P values as exact values whenever suitable.* |
| ☐ | ☒ | For Bayesian analysis, information on the choice of priors and Markov chain Monte Carlo settings |
| ☐ | ☒ | For hierarchical and complex designs, identification of the appropriate level for tests and full reporting of outcomes |
| ☐ | ☒ | Estimates of effect sizes (e.g. Cohen's *d*, Pearson's *r*), indicating how they were calculated |

*Our web collection on statistics for biologists contains articles on many of the points above.*

## Software and code

Policy information about availability of computer code

| Data collection | All data was complied using the open source software R. All versions of the R packages used together with the code and data are available at the open science framework: DOI 10.17605/OSF.IO/TYK7C. |
|---|---|
| Data analysis | All data analysis was performed using the open source software R and BayesTraits V4. All versions of the R packages used together with the code and analysis outputs are available at the open science framework: DOI 10.17605/OSF.IO/TYK7C. |

For manuscripts utilizing custom algorithms or software that are central to the research but not yet described in published literature, software must be made available to editors and reviewers. We strongly encourage code deposition in a community repository (e.g. GitHub). See the Nature Portfolio guidelines for submitting code & software for further information.

## Data

Policy information about availability of data

All manuscripts must include a data availability statement. This statement should provide the following information, where applicable:
- Accession codes, unique identifiers, or web links for publicly available datasets
- A description of any restrictions on data availability
- For clinical datasets or third party data, please ensure that the statement adheres to our policy

R code, BayesTraits code, data and analysis results are available at the open science framework: DOI 10.17605/OSF.IO/TYK7C. Full citations of references in supplementary tables are given in the method references48-382.

# Human research participants

Policy information about studies involving human research participants and Sex and Gender in Research.

| Reporting on sex and gender | NA |
| Population characteristics | NA |
| Recruitment | NA |
| Ethics oversight | NA |

Note that full information on the approval of the study protocol must also be provided in the manuscript.

# Field-specific reporting

Please select the one below that is the best fit for your research. If you are not sure, read the appropriate sections before making your selection.

☐ Life sciences ☐ Behavioural & social sciences ☒ Ecological, evolutionary & environmental sciences

For a reference copy of the document with all sections, see nature.com/documents/nr-reporting-summary-flat.pdf

# Ecological, evolutionary & environmental sciences study design

All studies must disclose on these points even when the disclosure is negative.

| Study description | We show that the evolution of obligate symbiosis with microbes has played a key role in explaining the adaptive radiation of insects. Using phylogenetic comparative analyses of data on 1850 microbe-insect symbioses across 402 insect families we show that:<br>• Obligate symbiosis has allowed insects feeding on generalist diets to unlock specialized resources such as phloem, xylem, blood and wood.<br>• Across diverse niches, insect feeding specialization is explained by symbiotic microbes supplying a single key nutrient - B vitamins.<br>• By allowing the exploitation of new feeding niches, symbionts have shaped insect diversification. In some cases, such as herbivorous insects, symbionts have facilitated spectacular radiations. In other cases, such as blood feeding, feeding specialization has been severely limited diversification. |
| Research sample | The study analyses published accounts of 1850 microbe-insect symbioses. Full details of the species studied and the methods used are given in the supplementary data tables 1-4 and in the method references 101-409. |
| Sampling strategy | Published literature was searched to retrieve all studies examining microbe-insect symbiosis. Full details of how literature was found are given in the Material and Methods. |
| Data collection | We complied a database on insect-microbe symbioses by: (1) searching published literature using the following key words [order name] OR [family name] AND "symbio"* using the search engines Web of Science and Google scholar during 2015-2017 and again in 2020, (2) searching several prominent reviews (e.g. Ries 1931, Schneider 1939, Müller 196243, Buchner 1965, Douglas 1989, Abe et. al. 2000, Bourtzis and Miller 2003, 2006 and 2009, Baumann 2005, Baumann et. al. 2013), and (3) forward and backward searches from the resulting papers. A full list of the papers screened can be found in Supplementary Table 2. |
| Timing and spatial scale | The time periods of data collection are given in the Material and Methods. Published literature was searched during 2015-2017 and updated in 2020 due to the time taken for data analyses to be completed. |
| Data exclusions | The aim of our paper was to investigate the evolution of beneficial obligate symbioses. We therefore excluded studies: (1) on parasitic symbionts, such as those that manipulate host reproduction (e.g. Spiroplasma, Cardinium, Wolbachia) that have not evolved beneficial functions; (2) that failed to screen the entire insect (e.g. only performed insect gut analyses); and (3) on symbionts with presumed beneficial functions, but that lacked data needed for our obligate criteria (see below). Fungal and protist symbionts were included where data on host dependency was available. Analyses of host-symbiont coevolution were restricted to symbionts for which a phylogeny could be constructed (bacteria with 16S rRNA genetic data: see section 2 for details). |
| Reproducibility | All data analysis was performed using the open source software R and BayesTraits V4. All versions of the R packages used together with fully reproducible R project scripts are available at the open science framework: DOI 10.17605/OSF.IO/TYK7C. |
| Randomization | NA. The study was a comparative analysis of all published data. |
| Blinding | NA. The study was a comparative analysis of all published data. |

Did the study involve field work? ☐ Yes ☒ No

# Reporting for specific materials, systems and methods

We require information from authors about some types of materials, experimental systems and methods used in many studies. Here, indicate whether each material, system or method listed is relevant to your study. If you are not sure if a list item applies to your research, read the appropriate section before selecting a response.

## Materials & experimental systems

| n/a | Involved in the study |
|---|---|
| ☒ | Antibodies |
| ☒ | Eukaryotic cell lines |
| ☒ | Palaeontology and archaeology |
| ☐ | ☒ Animals and other organisms |
| ☒ | Clinical data |
| ☒ | Dual use research of concern |

## Methods

| n/a | Involved in the study |
|---|---|
| ☒ | ChIP-seq |
| ☒ | Flow cytometry |
| ☒ | MRI-based neuroimaging |

## Animals and other research organisms

Policy information about studies involving animals; ARRIVE guidelines recommended for reporting animal research, and Sex and Gender in Research

| | |
|---|---|
| Laboratory animals | The study did not involve laboratory organisms |
| Wild animals | The study analyses published accounts of 1850 microbe-insect symbioses. Full details of the species studied and the methods used are given in the supplementary data tables 1-4 and in the method references 101-409. |
| Reporting on sex | NA |
| Field-collected samples | NA |
| Ethics oversight | NA |

Note that full information on the approval of the study protocol must also be provided in the manuscript.

