## [Peer Review File · Nature Ecology & Evolution]

Peer Review Information

Journal: Nature Ecology & Evolution

Manuscript Title: Symbioses shape feeding niches and diversification across insects

Corresponding author name(s): Charlie K. Cornwallis, Lee M. Henry

Editorial Notes:

Reviewer Comments & Decisions:

Decision Letter, initial version:

6th September 2022

Dear Dr Cornwallis,

Your Article, "Symbiont-driven niche expansion shapes the adaptive radiation of insects" has now been seen by five reviewers. You will see from their comments copied below that while they find your work of considerable potential interest, they have raised quite substantial concerns that must be addressed. In light of these comments, we cannot accept the manuscript for publication, but would be very interested in considering a revised version that addresses these serious concerns.

We hope you will find the reviewers' comments useful as you decide how to proceed. If you wish to submit a substantially revised manuscript, please bear in mind that we will be reluctant to approach the reviewers again in the absence of major revisions.

We are particularly concerned with the criticisms of Reviewer #4 and we won't be able to accept a revision unless all issues are addressed to the point of making the various conclusions robust.

If you choose to revise your manuscript taking into account all reviewer and editor comments, please highlight all changes in the manuscript text file in Microsoft Word format.

* Include a "Response to reviewers" document detailing, point-by-point, how you addressed each referee comment. If no action was taken to address a point, you must provide a compelling argument. This response will be sent back to the referees along with the revised manuscript.

* If you have not done so already we suggest that you begin to revise your manuscript so that it conforms to our Article format instructions at <http://www.nature.com/natecolevol/info/final-submission>. Refer also to any guidelines provided in this letter.

2* Include a revised version of any required reporting checklist. It will be available to referees (and, potentially, statisticians) to aid in their evaluation if the manuscript goes back for peer review. A revised checklist is essential for re-review of the paper.

[REDACTED]

If you wish to submit a suitably revised manuscript we would hope to receive it within 6 months. If you cannot send it within this time, please let us know. We will be happy to consider your revision so long as nothing similar has been accepted for publication at Nature Ecology & Evolution or published elsewhere.

Nature Ecology & Evolution is committed to improving transparency in authorship. As part of our efforts in this direction, we are now requesting that all authors identified as 'corresponding author' on published papers create and link their Open Researcher and Contributor Identifier (ORCID) with their account on the Manuscript Tracking System (MTS), prior to acceptance. This applies to primary research papers only. ORCID helps the scientific community achieve unambiguous attribution of all scholarly contributions. You can create and link your ORCID from the home page of the MTS by clicking on 'Modify my Springer Nature account'. For more information please visit www.springernature.com/orcid.

Thank you for the opportunity to review your work.

[REDACTED]

Reviewer expertise:

Reviewer #1: insect-bacteria symbioses

Reviewer #2: biological interactions and macroevolution

Reviewer #3: ecological interactions

Reviewer #4: insect-bacteria symbioses

Reviewer #5: phylogenetic comparative methods

2Reviewers' comments:

Reviewer #1 (Remarks to the Author):

In this communication by Cornwallis and colleagues, the authors apply an elegant phylogenetic approach to test various hypotheses that, over the years, have come to define the field of insect symbiosis. Chiefly, why do obligate symbioses evolve? And how do they influence diversification in insects? The manuscript is extremely well-written, the approach is novel, and the results appear to be exciting and of interest to a broad audience. Nonetheless, I do have a few comments and questions.

First, the definition of 'obligate symbiosis' is unusual. Obligate dependence is typically and primarily defined experimentally, where symbiont loss corresponds to a net decrease in survivorship, development, and fitness. Here, obligate dependence was determined by the presence of morphological structures exclusively associated with symbiosis (e.g., bacteriocytes) (Line 64; and 533). But where information on symbiont localization is lacking, data on the impact of symbiont loss was then used to determine the obligate status of a mutualism. This is problematic since a number of insect-microbe symbioses feature bacteriomes, or symbiotic organs, but the benefit to the insect host is either untested, or only context-dependent, most evidently in weevils and beetles relying on tyrosine supplementation for cuticle hardening. Applying a widely accepted definition of obligate symbiosis is thus critical for redefining the initial working dataset, for which much of the downstream analyses, and exciting findings, depends on.

The authors did an exceptional job curating the dataset at the center of this study. This was not a trivial effort, and I am certain that the field will benefit immensely from this work. Nonetheless, a few statements suggest a greater familiarity is still necessary with the literature and the range of obligate symbioses insects engage in. This is most evident in the claim that only four insect families harbor defensive symbionts (line 104). Classical examples from fungus-growing ants (Currie et al. 1999, *Nature*), beetles (Kellner et al. 1996, *Oecologia*; Florez et al. 2017, *Nature Communications*), aphids (Oliver et al. 2009, *Science*), and beewolves (Kaltenpoth et al. 2005, *Current Biology*) indicate that this is an underestimate. Psyllids are certainly not the only insect group that obligately depend on the defensive functions of their microbes (see examples above), so a more thorough examination is necessary.

One of the most exciting findings reported in this study, and for which there are many, is the connection between B vitamin deficiency in an insect's diet and its propensity to engage in an obligate symbiotic partnership. Given a wave of recent studies, the authors find ample experimental evidence to support their claim. This was supplemented with an elegant analysis to delineate between the functional convergence of these symbionts from the reliance on a restricted set of symbiotic partners. The implication here is that hosts evolved dependence on a broad range of microbes that in turn associate with insects feeding on low vitamin diets. Missing, however, is the metabolic conservation of the biosynthetic pathways involved. Specifically, are B vitamin pathways, especially B5 and B9, enriched in the reduced (and not-so-reduced) genomes of bacterial insect symbionts? And given the central role B vitamin deficiency appears to play in the evolution of obligate symbioses, are B vitamin biosynthesis pathways more conserved than those that encode other host-beneficial factors (e.g., amino acid)? Given the publicly available genomic resources for many of the cited study systems, such

3an analysis would complement the rigorous phylogenetic approaches at the center of this study.

Reviewer #2 (Remarks to the Author):

I have read and reviewed the manuscript "Symbiont-driven niche expansion shapes the adaptive radiation of insects", which is a very interesting and well written manuscript. The authors explore the roles that obligate symbionts had for niche evolution and diversification of insects. Obligate symbionts seem to be key to insects that have diets with low levels of vitamin B, allowing them to expand their niche preferences. However, diversification following feeding niche evolution depended on the niche invaded. While symbionts might have helped herbivorous insects to diversify, the opposite happened to blood-feeding insects.

I have enjoyed reading the manuscript and commend the authors for synthesising so many analyses into a fairly condensed manuscript (except for the methods section). My comments are in the interest of making the main text clearer for the readers. Some of my doubts got answered when reading the methods, but I kept them here because the authors might find it worth to try and clarify these sentences in the main text.

- So many Lepidoptera tips in the tree! Seems like there is a taxonomic bias. How does that affect the analyses?
- Which insect life stage is considered?
- If families can have both states (obligate/non obligate), then how did you classify them into either one of the states?
- "This pattern of food utilisation explains the current distribution of obligate symbiosis remarkably well, where over 90% of insect species feeding on blood, phloem, xylem and wood have obligate symbionts" - what about the other way around? The percentage of species with symbionts that feed on blood, phloem, xylem or wood?
- Please include color legend in both panels of fig 4.
- Lines 273-274 - "lineages with higher rates of obligate symbiosis were also more specious", but obligatory symbiosis is binary in the figure.
- 287-289 - Isn't this the main dataset? Maybe it should say "excluding"?
- Lines 510-512. Please explain why vertically transmitted symbionts were not included in coevolution analyses.
- Line 569 - Typo in "hermatophagy".
- Lines 568-569 - "if species fed in more than one of the following niches". Do you mean single species feeding in more than one niche, right? It could be clearer, so that there's no conflict with the next sentence.
- Line 614 - I suggest "diversification can be modelled using three..." because when I first read this, I thought all three approaches had been used in this study.
- Line 727 - Which 2 binary traits? Low or high B vitamin levels and presence/absence of obligate symbionts? I understand section 3 has more general explanations whereas section 4 presents the details, but it would be much easier to follow if those sections were connected by referring to each other in the text.

4- Line 757 - remove 'of' before 'node'.
- Methods section 4.3.1 - Is there a reason for comparing the focal node to both descendant nodes instead of comparing the focal node to its parent node, which would mean that only events along one branch are considered? I find the classification based on at least one descendant being obligate/non-obligate unnecessarily more complicated than simply comparing nodes at the beginning and end of a branch.
- Supplementary tables are hard to follow because there's not much explanation. The connection to the text is there, but there's a lot of information in the tables that is hard to figure out what it means.
- Couldn't find the code repository at osf.io.

Reviewer #3 (Remarks to the Author):

In this paper, the authors address the question whether the engaging in a symbiotic relationship is linked to the tendency of insect taxa to diversify. To address this issue, the authors perform a meta-analysis, in which they analyze more than 1,800 different symbiont-host interactions across different insect families. Using the existence of morphological structures that house the insects' symbionts or diagnostic experiments as a proxy for the interaction being obligate, the authors test whether i) insects with obligate symbionts are more likely to diversify taxonomically and ii) if this is linked to the insects' diet. The latter question has been analyzed by correlating the abundance of the nutrient in the focal food source to the presence of obligate symbionts (Fig. 2). In addition, the most likely ancestral composition of food sources in terms of symbiont-provided nutrients (i.e. vitamins, amino acids) has been calculated to correlate the change to the derived state with the association with obligate symbionts (Fig. 3).

This paper addresses an important question and is the first one to systematically verify whether obligate symbiosis is correlated with an increased rate of diversification. Previous studies have tested this idea using a much smaller set of host-symbiont interactions (e.g.: doi.org/10.1098/rspb.2012.2820). Thus, the main advance of this paper is that a large dataset has been analyzed using the same statistical approach, thus making the results directly comparable.

My main concerns are that the data is presented in a way that is difficult to understand. Moreover, the paper does not present novel findings that was not known before. Finally, I think that besides species richness also the rates of speciation of symbiont-associated and symbiont-free clades should be compared.

Below, I will elaborate on these points in more detail.

Main points:

(1) The data is consistently presented in a way that is extremely difficult to understand. This applies in particular to the figures that I find hard to grasp since neither the main text nor the figure legend provides sufficient information to understand it. Thus, it remains frequently unclear what exactly has been done and what is shown. While this information is presented in detail in the materials and methods, I think that also the main text (and the figures in particular) should be sufficiently clear to understand the main points without having to refer to the methods section.

5For example:

a) Page 8, lines 140-148: It is unclear what has been correlated with what. Please clarify.

b) Figure 2:

- Please explain the x-axis

c) Figure 3:

- It is unclear to me what is shown and compared in this graph. The naming of violins is counterintuitive and it is not clear to me what the set of interactions is that is considered in each case.
- What does the difference between "non to Ob" versus "Ob to Ob" show? How are these two cases linked with each other. I do not see this to be a meaningful comparison.
- Wouldn't it be possible to show the difference in the estimated ancestral and currently used level of vitamin (y-axis) depending on whether an obligate interaction evolved during this time or not (x-axis)? I think this might help to clarify this graph.

- It would be helpful to mention how many cases were considered in each of the different groups.
- Also: Did the authors correct for multiple testing in this graph?

d) Figure 4:

- In panel A please explain what the two different colours mean. Also, the number of cases below should be mentioned in the same order as the data is shown.
- Panel B: Please explain the colour code. Also: why is "NonObligate" written in capital letters and in one word without hyphen?

(2) I find the main conclusions not very surprising. It was known beforehand that obligate symbioses were particularly common in insects that feed on plant sap, xylem, and blood, because essential nutrients are low in these food sources. In these cases, insects can only use these food sources when they have symbionts to complement them with the missing essential metabolites. Thus, it is a circular argument to then conclude that these taxa are more species-rich. This is self-evident, because these insects can only use these food sources when they have the corresponding symbionts. Sister clades, from which these lineages derive, should obviously be less diverse.

(3) The key argument is based on data of species richness of insect clades that are or are not associated with obligate bacterial symbionts (Fig. 4). However, I think that also the rate of speciation should be considered, which could be correlated to the presence of obligate symbionts. If the pattern also holds in this case, this would strengthen the argument even more.

(4) Page 6, lines 107-111: This part lacks many key references and examples. Please have a look at e.g. Florez et al. 2015 Natural Product Reports for an overview.

(5) Verification analysis: The author say they repeated the analyses to examine the robustness of the

6results. However, the results of these analysis are not shown. To allow the reader to really judge how much the presented results depend on the set of examples studied, the results of these control analyses should be presented graphically and using statistics (in the supplementary information).

(6) The discussion is way too short and should link the main findings to the existing literature.

(7) I think the work that has been done does not warrant to talk about niche expansion, since it remains unclear how the size of the original niche was and whether there was just a shift (i.e. use of a different niche) or a niche expansion (i.e. access that a new niche in addition to the previous niche). Thus I recommend rewording the title.

Minor comment:

- Page 6 line 106: I think the original papers rather than a review should be cited in this context, to give credit to the authors making this discovery.

Reviewer #4 (Remarks to the Author):

In this manuscript, the authors are reporting on the results of a meta-analysis on the correlation between the occurrence of obligate symbionts in insects and the feeding ecology as well as species richness of their host families. Based on an extensive literature survey, they compile an impressive set of data on the presence or absence of obligate symbionts across 400 insect families, as well as the nutritional composition of the insects' food sources based on available data for the general food types. They report on 16 independent origins of obligate symbioses in insects and identify several B vitamins as the only nutritional components whose deficiencies significantly associate with the evolution of symbiosis. Reconstructing ancestral feeding niches and their nutritional composition, the authors test whether symbioses evolved before or after switches to B vitamin-deficient diets and find evidence for the former. Finally, they assess the association between feeding niche, symbiosis, and species richness across the different insect families, finding evidence for a significant impact of feeding niche on species richness, and for symbiosis being associated with increased species richness in herbivorous families.

I have reviewed this paper previously for another journal, and the authors have made important changes and additions to the analyses and the manuscript in response to the reviewers' comments. However, there are in my view still a few major drawbacks in the analyses that cast some doubt on the validity of the main conclusions that the authors present, and there are several overstatements in the manuscript, especially pertaining to the causal link between dietary B vitamin content and the evolution of symbiosis. Nevertheless, this is an impressive meta-analysis that provides an interesting overview of evolutionary patterns in insect symbioses and their possible links to dietary transitions of the hosts.

Major comments:

1. While I understand the rationale for collapsing taxa on the family level, this entails a couple of

7shortcomings for the analyses, which are not or only partly considered in the paper and in my view could have a severe impact on the results:

a. Several insect families contain symbiotic and non-symbiotic taxa (e.g. Lygaeidae, Formicidae, Melyridae, Chrysomelidae, etc), with sometimes multiple evolutionary transitions from non-symbiotic to symbiotic. Considering the families as ancestrally symbiotic pushes the transition to an earlier node than is actually true. This causes problems when trying to infer the evolutionary order of events (diet shift and origin of symbiosis).

b. It can impact the analysis of the impact of symbiosis on diversification (see below).

2. One of the most interesting conclusions of the study is that dietary specialization followed the acquisition of symbionts, rather than the other way around. Even though I intuitively tend to agree that this makes sense, I am not convinced that the data presented here provide compelling evidence for this scenario. The conclusion is based on the data presented in Figure 3, indicating that differences in vitamin B5 and B9 levels were lowest on branches from symbiotic to symbiotic nodes. When looking at the phylogeny presented in Figure 1, however, it appears that the vast majority of the branches connecting sym-sym nodes remain within the same feeding niche. If I understand correctly, then the nutrient composition of the diets of non-extant taxa were estimated by reconstructing this taxon's most likely feeding niche (l. 714-723 and 780-786) and then using the average nutritional values for this feeding niche as a proxy. If this is the case (and the methods are not clear in this regard), then all branches connecting taxa with the same feeding niche should get a difference of 0 in all nutritional values. Looking at Fig. 1 quickly shows that the vast majority of sym-to-sym node connections remain within the same feeding niche, so the plot for this category should show a lot of zeroes, while all values are below zero in Fig. 3. I am not sure how to explain this. Maybe I misunderstand the reconstruction of ancestral diets? The only other way that I can see how the authors may have inferred ancestral diets is by extrapolating the actual nutrient content levels of extant species' feeding niches to vitamin contents of presumed ancestral diets, essentially averaging out when shifts between feeding niches occurred. If vitamin levels of ancestral diets were inferred from the extrapolated nutrient contents of diets of extant species, though, I do not think it is a valid approach, as changes in nutrient levels during transitions within feeding niches cannot be reliably reconstructed this way.

3. Estimating the impact of symbiosis on diversification:

a. The sister taxa approach that the authors are using is commendable. However, I see two problems with this analysis:

i. Given that the origin of symbioses is collapsed on the family level, the analysis may not capture the necessary comparisons of symbiotic versus non-symbiotic sister taxa.

ii. I don't understand which taxa actually were compared. The authors describe 16 origins and 8 losses of symbiosis, but they indicate that 123 sister comparisons were done (l. 960). For this analysis, only the 24 transitions between symbiotic and non-symbiotic status seem to be relevant, so I do not understand where this large number of sister taxa comparison comes from. Again, it would be very helpful to have a clear list of origins and losses of symbiotic relationships presented, as well as the sister taxa comparisons that were used for this analysis. As it stands, it is impossible to evaluate the soundness of the approach.

b. The other two approaches are problematic, since it is not clear how the non-symbiotic families included in the phylogeny were selected, so it remains unclear how much of a sampling bias there is. For example, the most diverse order, the beetles, are only represented by 11 families in total, out of which 9 are considered to be symbiotic, even though there are more than 170 beetle families, most of which are non-symbiotic (and some really large non-symbiotic families are missing, e.g.

Staphylinidae, Tenebrionidae, as well as the phytophagous Buprestidae). Furthermore, the data presented in Figure 4a cannot be correct. There are an estimated 1 million described insect species to date (10^6), but some individual families in the figure are indicated to have 10^9 species.

l. 1: The title seems somewhat overstated, since the manuscript does not demonstrate niche expansions (rather: shifts), nor does it establish a causal link between symbiosis and adaptive radiations.

l. 80: It is important to somewhere explicitly list the 16 reconstructed evolutionary origins of symbioses and the eight losses, as they are difficult to impossible to extract from Fig. 1 and Table S5.

l. 159: Here and elsewhere, the authors put too much weight on the B vitamins as the nutritional components driving the evolution of symbiosis, while it has been shown that other factors can be the ones driving this (e.g. eAAs, digestive enzymes). In particular in ll. 307-308: This is correlational, so causality cannot be inferred! Stating that, based on this analysis of inferred ancestral diets, B vitamins are the only nutrients found to significantly correlate with the presence of symbionts is fine, but claiming that this was the driving force behind the evolution of symbiosis is not.

l. 610-611: This seems like a strange approach. Given that there are missing values for some dietary components in some diets, this can have a strong impact on the mean, especially if a number of micronutrients is missing, pushing the mean towards much lower values. I understand the rationale for avoiding the problem of compositional data (although I am not sure whether using this standardization by dividing by the mean instead of the sum actually solves this, since the results will still depend on the other dietary components), but the approach the authors are using created a substantial problem when comparing nutrient composition across diets.

Reviewer #5 (Remarks to the Author):

This manuscript seeks to study the macroevolution of symbiosis in insects. The paper is well written; however, I felt a little dazzled by results and p-mcmc values. So much so that I became slightly confused about which ones are the most important and which are linked to which hypothesis. I think in any revision I would recommend the author go through the manuscript with the reader in mind a really link the results to the specific hypotheses and expectations.

My expertise is in phylogenetic comparative methods so I'm reading the manuscript from that perspective. I think the analyses on the whole look well done. The tree itself is very difficult and I think that needs to be made more clear in any revision. The dating and topological uncertainty of the tree is not really considered. Whilst I appreciate it would be far outside the scope of this study to produce a better foundation tree – I think this need to be discussed more throughout.

I have a few important concerns that I think the authors should consider:

1. Number of species in a family. The number of species estimated to be in each family is a critical estimate to many of the analyses in this study. However, I think it will be quite likely that the number of species estimated in each family will be highly correlated with research effort, which will in turn is likely to be associated with occurrences of symbiosis and other covariates important here. So given

9this, I think the author need to control for research effort (associated with each family) in their analyses.

2. Stochastic mapping and Discrete. I am slightly confused about why the authors need to use both stochastic mapping and Discrete. I would suggest they use the best model (dependent/independent) and then use that to determine the ancestral stats of the nodes of interest. This part of the methods was very unclear – why do you need Stochastic mapping at all here?

3. The significance and difference of rates in the Discrete mode. The authors say they use the posterior distribution of rates to determine whether the rates are significantly different to each other. However, there is a reversible-jump procedure available in BayesTraits that is specifically designed to reduce dimensionality and determine if there are differences among rates – I think the authors should use that.

Author Rebuttal to Initial comments

Response to reviewers

We are very grateful to the editors and reviewers for their positive and constructive feedback. Their input has been extremely valuable in revising the paper, which we now feel is substantially improved. Below are the details of the specific changes we have made to address their comments. Line numbers in word documents with track changes are sometimes not reproducible across computers. All line numbers therefore refer to the pdf version of the revised manuscript.

Reviewer #1 (Remarks to the Author):

In this communication by Cornwallis and colleagues, the authors apply an elegant phylogenetic approach to test various hypotheses that, over the years, have come to define the field of insect symbiosis. Chiefly, why do obligate symbioses evolve? And how do they influence diversification in insects? The manuscript is extremely well-written, the approach is novel, and the results appear to be exciting and of interest to a broad audience. Nonetheless, I do have a few comments and questions.

Thank you very much for the positive and constructive feedback.

First, the definition of ‘obligate symbiosis’ is unusual. Obligate dependence is typically and primarily defined experimentally, where symbiont loss corresponds to a net decrease in survivorship, development, and fitness. Here, obligate dependence was determined by the presence of morphological structures exclusively associated with symbiosis (e.g., bacteriocytes) (Line 64; and 533). But where information on symbiont localization is lacking, data on the impact of symbiont loss was then used to determine the obligate status of a mutualism. This is problematic since a number of insect-microbe symbioses feature bacteriomes, or symbiotic organs, but the benefit to the insect host is either untested, or only context-dependent, most evidently in weevils and beetles relying on tyrosine supplementation for cuticle hardening. Applying a widely accepted definition of obligate symbiosis is thus critical for redefining the initial working dataset, for which much of the downstream analyses, and exciting findings, depends on.

We agree that obligate dependence on endosymbionts is ideally measured through symbiont removal experiments, as we mentioned in our methods. Unfortunately, these experiments are exceedingly rare, which is why we used proxies to allow for comparison across a wider range of species. Following reviewer 1’s comments, we have now added that these are ‘putatively obligate’ relationships, so it is clear our definition is based on proxy information.

In addition, we have provided extra justification as to why the criteria we use are strong indicators of obligate endosymbiosis. To clarify, obligate dependence was based on two criteria, both of which had to be fulfilled: (i) ubiquitous presence of the symbiont in reproductive females under natural conditions; and (ii) the presence of bacteriocytes, or phylogenetic concordance, or fitness losses when symbiont removed. We realise that in the previous version of the manuscript it was unclear that criteria (i) excludes cases where symbiont dependence is context specific. For example, in some species of *Formica* and *Cardioconyla* ants, queens occasionally occur in nature without symbionts in bacteriocytes and are therefore considered putatively facultative in our database. The criteria for assessing obligate dependence have been clarified in the text (lines 73-80 & 1004-1022).

1

11The authors did an exceptional job curating the dataset at the center of this study. This was not a trivial effort, and I am certain that the field will benefit immensely from this work. Nonetheless, a few statements suggest a greater familiarity is still necessary with the literature and the range of obligate symbioses insects engage in. This is most evident in the claim that only four insect families harbor defensive symbionts (line 104). Classical examples from fungus-growing ants (Currie et al. 1999, Nature), beetles (Kellner et al. 1996, Oecologia; Florez et al. 2017, Nature Communications), aphids (Oliver et al. 2009, Science), and beewolves (Kaltenpoth et al. 2005, Current Biology) indicate that this is an underestimate. Psyllids are certainly not the only insect group that obligately depend on the defensive functions of their microbes (see examples above), so a more thorough examination is necessary.

We thank the reviewer for bringing these examples to our attention. We have now expanded our discussion on defensive mutualisms and explained how they differ from obligate nutritional associations, with the inclusion of additional examples (lines 137-175).

Specifically:

Currie et al. 1999, Nature: This is an excellent example of a defensive mutualism, but it was excluded from our database because it is an ectosymbiosis, and our study only considered endosymbionts. This has been clarified in text, and we have made reference to known defensive ectosymbiotic mutualism, and cited a review on the subject (lines 139-143).

Kellner et al. 1996 (and Kellner 2002), Oecologia: Is included in our database, however, there was insufficient information on the frequency of symbiont carriage to test if the symbiont was universally present in females. The obligate status was therefore included as NA until evidence suggests otherwise.

Florez & Kaltenpoth 2017: Has been added to our database as a facultative symbiosis (Supplementary Table 2).

Oliver et al. 2009: This was included in our database as a facultative symbiosis.

Kaltenpoth et al. 2005: The beewolf symbiosis was included in our database. We cited Kaltenpoth et al 2014 *PNAS* as it has broader sampling across species, but we have also now cited Kaltenpoth et al. 2005 for *Crabronidae* in the database.

One of the most exciting findings reported in this study, and for which there are many, is the connection between B vitamin deficiency in an insect's diet and its propensity to engage in an obligate symbiotic partnership. Given a wave of recent studies, the authors find ample experimental evidence to support their claim. This was supplemented with an elegant analysis to delineate between the functional convergence of these symbionts from the reliance on a restricted set of symbiotic partners. The implication here is that hosts evolved dependence on a broad range of microbes that in turn associate with insects feeding on low vitamin diets. Missing, however, is the metabolic conservation of the biosynthetic pathways involved. Specifically, are B vitamin pathways, especially B5 and B9, enriched in the reduced (and not-so-reduced) genomes of bacterial insect symbionts? And given the central role B vitamin deficiency appears to play in the evolution of obligate symbioses, are B vitamin biosynthesis pathways more conserved

2

12than those that encode other host-beneficial factors (e.g., amino acid)? Given the publicly available genomic resources for many of the cited study systems, such an analysis would complement the rigorous phylogenetic approaches at the center of this study.

This is an excellent point. The genome analysis suggested would definitely be a useful next step. However, to do this properly is a significant undertaking given the complexity of vitamin synthesis pathways putting it beyond the scope of our paper. We have therefore linked our results to published genome studies that demonstrate that the metabolic pathways for synthesizing B vitamins have been retained in the genomes of symbionts across a diversity of host feeding niches (lines 253-256, 261-264). Furthermore, we have added that comparative genomic analyses of vitamin B pathways across different insect symbioses would be an exciting future research direction (line 264-266).

Reviewer #2 (Remarks to the Author):

I have read and reviewed the manuscript "Symbiont-driven niche expansion shapes the adaptive radiation of insects", which is a very interesting and well written manuscript. The authors explore the roles that obligate symbionts had for niche evolution and diversification of insects. Obligate symbionts seem to be key to insects that have diets with low levels of vitamin B, allowing them to expand their niche preferences. However, diversification following feeding niche evolution depended on the niche invaded. While symbionts might have helped herbivorous insects to diversify, the opposite happened to blood-feeding insects.

I have enjoyed reading the manuscript and commend the authors for synthesising so many analyses into a fairly condensed manuscript (except for the methods section). My comments are in the interest of making the main text clearer for the readers. Some of my doubts got answered when reading the methods, but I kept them here because the authors might find it worth to try and clarify these sentences in the main text.

We very much appreciate the supportive feedback on our work and time spent helping us clarify the text.

- So many Lepidoptera tips in the tree! Seems like there is a taxonomic bias. How does that affect the analyses?

Lepidoptera are a specious clade that has attracted much research attention and so are very well represented in the tree. However, they are homogenous in their feeding niches and all lack obligate symbionts. This lack of variation across lepidoptera actually means they have minimal influence on our results: statistical power in comparative analyses comes from lineages differing in their traits rather than numbers of species.

A further verification that our conclusions are not influenced by Lepidoptera is that including them in analyses has little effect on our results. Specifically, in the supplementary tables S24-S27 we present the results of analyses where species, including Lepidoptera, that have not been directly studied for obligate symbiosis are excluded. Our results when excluding these taxa were qualitative and quantitatively similar.

3

13We have clarified in the text that these sensitivity analyses also show our results are robust to the exclusion of large orders without obligate symbionts (lines 1677-1679).

- Which insect life stage is considered?

We collected data on both adult and juvenile life stages. For species where adults and juveniles have different diets, we combined all food items to classify species feeding niches. To estimate nutrient values for each family, we first calculated the median content across adult and juvenile diets for each nutrient for each species. Second, to gain family level estimates of each nutrient, we calculated the median concentration of each nutrient across the species in that family. Where species level data was not available, we estimated concentrations of each nutrient for the feeding niche reported for that family.

We have clarified how we calculated the nutritional composition of diets at the end of the introduction and in Methods section 1.3 (lines 92–98, 1074-1113).

- If families can have both states (obligate/non obligate), then how did you classify them into either one of the states?

For all analyses apart from those examining transition rates, we analysed the percentage of species within families with obligate symbionts. Therefore, families that had species with and without obligate symbionts were not classified as having one state or the other, but rather analysed according to the prevalence of obligate symbionts within families.

The transition rate analyses, however, require that obligate symbiosis be classified into states. Out of 402 families, only 15 had species both with and without obligate symbionts (3.7%). For these 15 families, we classified them as having obligate symbionts if over 50% of species had obligate symbionts. We tested how sensitive our results were to this binary classification, by removing families that had species with and without obligate symbionts and re-running our analyses. We found extremely similar results, which are now reported in the sensitivity analyses section and in Supplementary Table S27.

- “This pattern of food utilisation explains the current distribution of obligate symbiosis remarkably well, where over 90% of insect species feeding on blood, phloem, xylem and wood have obligate symbionts” - what about the other way around? The percentage of species with symbionts that feed on blood, phloem, xylem or wood?

The percentage of species with obligate symbionts feeding on blood, phloem, xylem and wood is 52 out of 89 families (58%).

- Please include color legend in both panels of fig 4.

Now added.

- Lines 273-274 - “lineages with higher rates of obligate symbiosis were also more specious”, but obligatory symbiosis is binary in the figure.

For graphical purposes, we classified families as having obligate symbionts (>50% of species with obligate symbionts) or not (<50% of species with obligate symbionts). However, the

percentage of species with obligate symbionts in each family was analysed. We have clarified this in the figure 4 legend.

- **287-289 - Isn't this the main dataset? Maybe it should say "excluding"?**

Many thanks for spotting this typo. Now reads "excluding".

- **Lines 510-512. Please explain why vertically transmitted symbionts were not included in coevolution analyses.**

We wanted to test if related lineages of bacteria are recruited by related host species. Including vertically transmitted symbionts leads to a strong signal of co-speciation, not because of symbiont recruitment, but because of inheritance from a common ancestor. Therefore, we removed vertically transmitted symbionts to more accurately quantify how host and symbiont phylogenetic history influence symbiont recruitment. We have now clarified our approach (lines 974-977).

- **Line 569 - Typo in "hermatophagy".**

Addressed.

- **Lines 568-569 - "if species fed in more than one of the following niches". Do you mean single species feeding in more than one niche, right? It could be clearer, so that there's no conflict with the next sentence.**

Thanks for the suggestion. We have rephrased this sentence to improve clarity (lines 1063-1064).

- **Line 614 - I suggest "diversification can be modelled using three..." because when I first read this, I thought all three approaches had been used in this study.**

Agreed.

- **Line 727 - Which 2 binary traits? Low or high B vitamin levels and presence/absence of obligate symbionts? I understand section 3 has more general explanations whereas section 4 presents the details, but it would be much easier to follow if those sections were connected by referring to each other in the text.**

We have clarified that this refers to the binary traits of B vitamins and obligate symbiosis and cross-referenced sections (lines 1310-1311).

- **Line 757 - remove 'of' before 'node'.**

Addressed.

- **Methods section 4.3.1 - Is there a reason for comparing the focal node to both descendant nodes instead of comparing the focal node to its parent node, which would mean that only events along one branch are considered? I find the classification based on at least one descendant being obligate/non-obligate unnecessarily more complicated than simply comparing nodes at the beginning and end of a branch.**

Our aim was to test if lineages with different levels of B vitamins precede the evolutionary gain and loss of obligate symbiosis. Therefore, we classified each node according to whether at least one descendent was predicted to have, or not have, acquired an obligate symbiont. Classifying each node by its parental state would mean that each parent node contributes to two different data points and non-independence of data (there would be twice as many comparisons as there should be).

- Supplementary tables are hard to follow because there's not much explanation. The connection to the text is there, but there's a lot of information in the tables that is hard to figure out what it means.

We have now added annotation to the tables to aid interpretation.

- Couldn't find the code repository at osf.io.

For some reason the search engine at osf.io returns many hits for the doi number. However, searching osf.io for the project number (TYK7C), or title, does retrieve the project as does searching the doi number at doi.org. We have now included extra information in the manuscript to resolve this confusion.

Reviewer #3 (Remarks to the Author):

In this paper, the authors address the question whether the engaging in a symbiotic relationship is linked to the tendency of insect taxa to diversify. To address this issue, the authors perform a meta-analysis, in which they analyze more than 1,800 different symbiont-host interaction across different insect families. Using the existence of morphological structures that house the insects' symbionts or diagnostic experiments as a proxy for the interaction being obligate, the authors test whether i) insects with obligate symbionts are more likely to diversify taxonomically and ii) if this is linked to the insects' diet. The latter question has been analyzed by correlating the abundance of the nutrient in the focal food source to the presence of obligate symbionts (Fig. 2). In addition, the most likely ancestral composition of food sources in terms of symbiont-provided nutrients (i.e. vitamins, amino acids) has been calculated to correlate the change to the derived state with the association with obligate symbionts (Fig. 3).

This paper addresses an important question and is the first one to systematically verify whether obligate symbiosis is correlated with an increased rate of diversification. Previous studies have tested this idea using a much smaller set of host-symbiont interactions (e.g.: doi.org/10.1098/rspb.2012.2820). Thus, the main advance of this paper is that a large dataset has been analyzed using the same statistical approach, thus making the results directly comparable.

We are very grateful for the positive feedback.

My main concerns are that the data is presented in a way that is difficult to understand. Moreover, the paper does not present novel findings that was not known before. Finally, I think that besides species richness also the rates of speciation of symbiont-associated and symbiont-free clades should be compared.

6

16Below, I will elaborate on these points in more detail.

Main points:

(1) The data is consistently presented in a way that is extremely difficult to understand. This applies in particular to the figures that I find hard to grasp since neither the main text nor the figure legend provides sufficient information to understand it. Thus, it remains frequently unclear what exactly has been done and what is shown. While this information is presented in detail in the materials and methods, I think that also the main text (and the figures in particular) should be sufficiently clear to understand the main points without having to refer to the methods section.

We appreciate the time taken by the reviewer to help us present our findings as clearly as possible. We have edited the manuscript throughout, following reviewer 3's suggestions to make it clearer what analyses we have performed.

For example:

a) Page 8, lines 140-148: It is unclear what has been correlated with what. Please clarify.

This section has been re-written to clarify that B5 and B9 vitamins are correlated to obligate symbiosis (lines 195 -207). In addition, we found that different types of B vitamins (B1, B2, B3, B5, B6 and B9) were correlated across different food sources, showing that entire suites of B vitamins are missing from some insect diets.

b) Figure 2:

- Please explain the x-axis

The x-axis was the Z transformed concentration of each nutrient (lines 441-442 of previous manuscript). Concentrations were Z-transformed (mean centred with a standard deviation of 1) for analyses to enable comparisons between variables because the units of macro- and micro-nutrients are very different. For justification and rationale of this approach see: Schielzeth, H. 2010. Simple means to improve the interpretability of regression coefficients: Interpretation of regression coefficients. *Methods in Ecology and Evolution* 1:103–113.

However, we realize that it may be more straight forward to visualize raw data values. As suggested by reviewer 3, we have replotted all figures with nutrient values without Z-transforming axes.

c) Figure 3:

- It is unclear to me what is shown and compared in this graph. The naming of violins is counterintuitive and it is not clear to me what the set of interactions is that is considered in each case.

The figure shows the estimated ancestral values of B5 and B9 vitamins of nodes according to their predicted ancestral state of obligate symbiosis and their descendent nodes. We have now revised this figure, and associated text, to make it clearer.

- What does the difference between “non to Ob” versus “Ob to Ob” show? How are these two cases linked with each other. I do not see this to be a meaningful comparison.

7

17

Comparing ancestral nodes where descendants gained obligate symbionts (Non to Ob) versus ancestral nodes where obligate symbioses were maintained (Ob to Ob) tests if B vitamins changed after obligate symbiosis evolved (Ob to Ob lower than Non to Ob). We have now explained this more clearly in lines 859–860, 1473–1476 and Figures 3C and 3D.

- Wouldn't it be possible to show the difference in the estimated ancestral and currently used level of vitamin (y-axis) depending on whether an obligate interaction evolved during this time or not (x-axis)? I think this might help to clarify this graph.

We appreciate this very useful suggestion, and we have revised figure 3 to include reviewer 3's clarification.

- It would be helpful to mention how many cases were considered in each of the different groups.

We have added the estimated number of each transition type to the figure.

- Also: Did the authors correct for multiple testing in this graph?

This is an important point, but there are a number of reasons why multiple testing corrections are not needed here: 1) there are only 3 comparisons per trait, 2) the same data is not used in more than one comparison, and 3) multiple testing problems are ameliorated when using hierarchical Bayesian linear models like the ones we use in the manuscript. See Gelman, A., J. Hill, and M. Yajima. 2012. Why We (Usually) Don't Have to Worry About Multiple Comparisons. *Journal of Research on Educational Effectiveness* 5:189–211.

d) Figure 4:

- In panel A please explain what the two different colours mean. Also, the number of cases below should be mentioned in the same order as the data is shown.

We have added a legend to clarify that the two colours correspond to families with (blue) and without (grey) obligate symbionts.

- Panel B: Please explain the colour code. Also: why is "NonObligate" written in capital letters and in one word without hyphen?

We have added a figure legend to explain that the colours represent the feeding niches of families. The x-axis has also been adjusted to read "Non-obligate", consistent with other figures.

(2) I find the main conclusions not very surprising. It was known beforehand that obligate symbioses were particularly common in insects that feed on plant sap, xylem, and blood, because essential nutrients are low in these food sources. In these cases, insects can only use these food sources when they have symbionts to complement them with the missing essential metabolites. Thus, it is a circular argument to then conclude that these taxa are more species-rich. This is self-evident, because these insects can only use these food sources when they have the corresponding symbionts. Sister clades, from which these lineages derive, should obviously be less diverse.

Studies have shown that symbionts of specific lineages produce diverse essential nutrients for their insect hosts, allowing them to feed on a multitude of nutrient imbalanced food sources. However, there has been no broad synthesis across taxa asking whether there are common nutrient deficiencies across diverse diets that lead to the evolution of obligate symbioses. We show that vitamin B is the only nutrient deficiency significantly associated with obligate symbiosis across diets. This indicates that B vitamins are widely limiting and key to the evolution of dependence on microbes for nutrient provisioning in insects.

The reviewer is correct that symbionts can lead to utilization of new food sources, but this does not necessarily lead to an increase in species richness compared to *background* rates. In fact, there are alternative theories. One idea is that symbionts may trap hosts in niches leading to reduced diversification relative to background rates (see lines 302-308 of previous draft and lines 357-367 of the revised version). Alternatively, symbionts may open up new niches leading to greater species proliferation than expected. We found that the acquisition of symbionts is consistent with both scenarios: shifts to blood-feeding significantly decreased species richness, whereas exploitation of herbivorous food sources promoted species richness compared to background rates (lines 385–386). Our study resolves how the relative importance of these different issues vary across species.

(3) The key argument is based on data of species richness of insect clades that are or are not associated with obligate bacterial symbionts (Fig. 4). However, I think that also the rate of speciation should be considered, which could be correlated to the presence of obligate symbionts. If the pattern also holds in this case, this would strengthen the argument even more.

It would be very interesting to estimate rates of speciation, as well as rates of extinction, and test if they are influenced by obligate symbiosis. However, the techniques available to estimate speciation and extinction rates have been heavily criticised recently and also require species level phylogenies (Louca, S., and M. W. Pennell. 2020. Extant timetrees are consistent with a myriad of diversification histories. *Nature* 580:502–505. Rabosky, D. L., and R. B. J. Benson. 2021. Ecological and biogeographic drivers of biodiversity cannot be resolved using clade age-richness data. *Nature Communications* 12:2945). A major problem is that speciation and extinction are very hard to disentangle. If we see a more species rich group this could be because they had either speciated at a higher rate and/or gone extinct at a lower rate. Consequently, it is necessary to examine their joint effects on species richness. A recent introductory review to these issues is provided by Cornwell, W. & Nakagawa, S. Phylogenetic comparative methods. *Current Biology* 27, R333–R336 (2017).

(4) Page 6, lines 107-111: This part lacks many key references and examples. Please have a look at e.g. Florez et al. 2015 Natural Product Reports for an overview.

We thank the reviewer for bringing these examples to our attention. We have now included a more thorough review of defensive mutualisms, and clarified that our study is only considering endosymbionts. Several additional references and examples have been added (lines 137–143 and lines 166-173).

(5) Verification analysis: The author say they repeated the analyses to examine the robustness of the results. However, the results of these analysis are not shown. To allow the reader to really judge how much the presented results depend on the set of examples

studied, the results of these control analyses should be presented graphically and using statistics (in the supplementary information).

The results of verification analyses (now referred to as “Sensitivity analyses”) were all included in the Supplementary Tables 20-27 (SupplementaryTables.html). However, we appreciate that this is not immediately accessible to all readers and so have included extra figures of the restricted data in the extended data document with legends providing the most relevant statistical results (Extended Data Figures 6-8).

(6) The discussion is way too short and should link the main findings to the existing literature.

Our aim was to follow a structure whereby we discuss results as the paper unfolds. This is followed by a short summary discussion as per the journal guidelines. However, as suggested by reviewer 3, we have now: (1) provided additional links between our main findings and the existing literature in the discussion (Lines 481–482, 490-498, 511-514); and (2) added additional discussion and citations in the results section (e.g. Lines 137-143, 166-173, 253-256, 261-266).

(7) I think the work that has been done does not warrant to talk about niche expansion, since it remains unclear how the size of the original niche was and whether there was just a shift (i.e. use of a different niche) or a niche expansion (i.e. access that a new niche in addition to the previous niche). Thus I recommend rewording the title.

We have reworded the title to ‘Symbioses shape feeding niches and diversification across insects’.

Minor comment:

- Page 6 line 106: I think the original papers rather than a review should be cited in this context, to give credit to the authors making this discovery.

We have now revised the referencing in this section. When demonstrating that defensive symbionts have been discovered relatively recently, we refer to the original paper of the first example we are aware of (lines 139). We also clarify that since this discovery more defensive symbionts have been documented and here we reference a review that gives an overview of these findings (lines 139-140).

Reviewer #4 (Remarks to the Author):

In this manuscript, the authors are reporting on the results of a meta-analysis on the correlation between the occurrence of obligate symbionts in insects and the feeding ecology as well as species richness of their host families. Based on an extensive literature survey, they compile an impressive set of data on the presence or absence of obligate symbionts across 400 insect families, as well as the nutritional composition of the insects’ food sources based on available data for the general food types. They report on 16 independent origins of obligate symbioses in insects and identify several B vitamins as the only nutritional components whose deficiencies significantly associate with the evolution of symbiosis. Reconstructing ancestral feeding niches and their nutritional

10

20composition, the authors test whether symbioses evolved before or after switches to B vitamin-deficient diets and find evidence for the former. Finally, they assess the association between feeding niche, symbiosis, and species richness across the different insect families, finding evidence for a significant impact of feeding niche on species richness, and for symbiosis being associated with increased species richness in herbivorous families.

I have reviewed this paper previously for another journal, and the authors have made important changes and additions to the analyses and the manuscript in response to the reviewers' comments. However, there are in my view still a few major drawbacks in the analyses that cast some doubt on the validity of the main conclusions that the authors present, and there are several overstatements in the manuscript, especially pertaining to the causal link between dietary B vitamin content and the evolution of symbiosis. Nevertheless, this is an impressive meta-analysis that provides an interesting overview of evolutionary patterns in insect symbioses and their possible links to dietary transitions of the hosts.

We are grateful for the reviewer's current and previous comments. These suggestions have greatly helped us revise and further improve the paper.

Major comments:

1. While I understand the rationale for collapsing taxa on the family level, this entails a couple of shortcomings for the analyses, which are not or only partly considered in the paper and in my view could have a severe impact on the results:

a. Several insect families contain symbiotic and non-symbiotic taxa (e.g. Lygaeidae, Formicidae, Melyridae, Chrysomelidae, etc), with sometimes multiple evolutionary transitions from non-symbiotic to symbiotic. Considering the families as ancestrally symbiotic pushes the transition to an earlier node than is actually true. This causes problems when trying to infer the evolutionary order of events (diet shift and origin of symbiosis).

b. It can impact the analysis of the impact of symbiosis on diversification (see below).

We agree this is a potential problem so we carried out our analyses to eliminate this possibility. Our analyses do not assume that the ancestors of families containing symbiotic and non-symbiotic taxa had symbionts. This would be the case if we had categorised these 'mixed' families as having symbionts, but we did not. Instead, we analysed the percentage of species that had symbionts within families. In doing so, our models estimate the probability that the ancestral node of each family was symbiotic based on this percentage and that of related lineages. Consequently, families with a low percentage of symbiotic taxa, that are related to families with few symbiotic taxa, are predicted to have ancestral nodes that are non-symbiotic. We therefore did not consider families with and with symbionts as ancestrally symbiotic, and symbiotic states were not pushed to earlier nodes.

We have now further clarified in the main text (as opposed to just the methods) that we analyse percentage of symbiotic taxa within families not just binary classifications (lines 122-123).

There is one exception to this rule, which was the Discrete analysis of transition rates in BayesTraits (see response to reviewer 2). Discrete only allows binary variables to be

analysed and so we needed to classify families as symbiotic (>50% species with symbionts) or non-symbiotic (<50% species with symbionts). To examine whether our results were influenced by families being categorised in this way, we have added a sensitivity analysis removing families (3.7% of families) with and without symbiotic taxa (Supplementary Table 27). These analyses showed our results remain unchanged by removing these taxa.

2. One of the most interesting conclusions of the study is that dietary specialization followed the acquisition of symbionts, rather than the other way around. Even though I intuitively tend to agree that this makes sense, I am not convinced that the data presented here provide compelling evidence for this scenario. The conclusion is based on the data presented in Figure 3, indicating that differences in vitamin B5 and B9 levels were lowest on branches from symbiotic to symbiotic nodes. When looking at the phylogeny presented in Figure 1, however, it appears that the vast majority of the branches connecting sym-sym nodes remain within the same feeding niche. If I understand correctly, then the nutrient composition of the diets of non-extant taxa were estimated by reconstructing this taxon's most likely feeding niche (l. 714-723 and 780-786) and then using the average nutritional values for this feeding niche as a proxy. If this is the case (and the methods are not clear in this regard), then all branches connecting taxa with the same feeding niche should get a difference of 0 in all nutritional values. Looking at Fig. 1 quickly shows that the vast majority of sym-to-sym node connections remain within the same feeding niche, so the plot for this category should show a lot of zeroes, while all values are below zero in Fig. 3. I am not sure how to explain this.

We agree this could have been clearer, and are grateful to the reviewer for pointing out this point of confusion. The nutrient composition of the diets of non-extant taxa were *not* estimated by reconstructing each taxon's most likely feeding niche. Figure 1 and lines 714-723 & 780-786 of the previous version are solely concerned with estimating ancestral feeding niches and not the nutritional composition of diets. The average nutritional value for feeding niches was therefore not used as a proxy for the nutrient content of insect diets.

Estimating nutritional niches was in fact much more difficult, consisting of the following steps, which we outline in much greater detail in section 1.3:

1. Literature searches were performed to ascertain the food types (e.g. fruit, roots ect..) adults and juveniles of the 1850 species fed on (all information is listed in Supplementary Table 2).
2. The nutrient composition of all the food types that species were found to feed on (food types = 365), was estimated by searching dietary databases for as many examples of those food types as possible (range per food type = 1 to 24. Total number of nutrient estimates = 5504). From dietary databases we extracted values for carbohydrate, fat, protein, essential amino acids and non-essential amino acids (arginine, cystine, glycine, glutamic acid, prolin, tyrosine) and vitamins (A, B, C, D, E, K, choline, betaine).
3. For each food type a median concentration of each nutrient across all example food was calculated. Median nutrient values were combined with information on the food types species utilized to calculate a nutrient profile for adults and juveniles of each species.
4. Estimates of the nutrients in the diets of each species were calculated by taking a median across adults and juveniles.

12

225. Estimates of the dietary nutrients for each family were calculated by taking a median across all species within families.
6. In cases where species-specific diets were not available, we based diets on family-level feeding habits published in books and reviews listed in Supplementary Table 2.

As the nutrient composition of families are based on the food items that each species feeds on, nutrient values can differ across families within the same feeding niche.

With respect to figure 3, the y-axis does not represent differences between ancestral and descendent nodes, which we believe is the reviewers interpretation. Instead, the y axis of figure 3 presents estimates of B vitamin levels (Z-transformed) for ancestral nodes that fall into four different categories: ancestors with and without symbionts that have decedents with and without symbionts. Because data are Z-transformed, 0 on the y-axis represents the average level of B vitamins (mean of Z-transformed data = 0), rather than showing no change between ancestral and descendent nodes as implied by the reviewer. However, we realize that it may be more straight forward to visualize raw data values and so have replotted all figures with nutrient values without Z-transforming axes.

We have also heavily revised the description of how the nutrient compositions of families were calculated to make this clearer (lines 1074-1113). To increase the readability of Figure 3, we have changed the scale of the y-axis and two panels so that changes between ancestral and descendent nodes can be visualised.

Maybe I misunderstand the reconstruction of ancestral diets? The only other way that I can see how the authors may have inferred ancestral diets is by extrapolating the actual nutrient content levels of extant species' feeding niches to vitamin contents of presumed ancestral diets, essentially averaging out when shifts between feeding niches occurred. If vitamin levels of ancestral diets were inferred from the extrapolated nutrient contents of diets of extant species, though, I do not think it is a valid approach, as changes in nutrient levels during transitions within feeding niches cannot be reliably reconstructed this way.

Estimating the ancestral trait values using phylogenetic techniques is typically done using data on extant taxa, together with their phylogenetic relationships and a model of how traits change over evolutionary time (e.g Brownian motion, Rate-shifts, Ornstein-Uhlenbeck, Early-burst, slow-down). For a recent overview see: Revell, L. J., and L. J. Harmon. 2022. *Phylogenetic Comparative Methods in R*. Princeton University Press.

We followed standard methods by using the levels of B vitamins in the diets of extant insect families to infer ancestral values of B vitamins. The referee appears concerned with this approach because they believe nutrients levels are simply averaged across changes in feeding niches, rather than allowing rapid shifts in B vitamins that may occur with changes in feeding niches. This is not a concern for the following reasons:

First, we tested if low levels of B vitamins preceded or followed the evolution of obligate symbiosis using two different techniques that differ in the way ancestral values are estimated. In particular, the BayesTrait analyses quantified transition rates between high and low levels of B vitamins from states of obligate and non-obligate symbiosis. These analyses allow instantaneous switches from high to low and low to high B vitamins. We also used different

classifications of high and low B vitamins (above or below the 50% and 25% quantile) to ensure we captured changes in B vitamins of different magnitude.

Second, we examined the robustness of the estimates of ancestral values obtained from Bayesian Phylogenetic Mixed Models (BPMM) to marked shifts in B vitamins using rate shift models (Phylogenetic ridge regression (PRR): Castiglione, S., G. Tesone, M. Piccolo, M. Melchionna, A. Mondanaro, C. Serio, M. D. Febbraro, and P. Raia. 2018. A new method for testing evolutionary rate variation and shifts in phenotypic evolution. *Methods in Ecology and Evolution* 9:974–983). Overall, there was little evidence for rapid rate shifts in B vitamins indicating this does not adversely influence the estimation of ancestral values of B vitamins. Specifically, we found evidence that 3 out of the 401 nodes (<1%) showed evidence of rate shifts and 2 nodes for B9 (<1%).

In addition, we checked how accurately BPMM and PRR models could predict raw values of B vitamins for tips in the tree, as a measure of how well we could model variation in B vitamins across families. Our BPMMs were extremely good at predicting raw values of B vitamins (Pearson correlations > 0.98. Extended Data Figure 3), and more accurate than models that allowed rate shifts in B vitamins (PRRs).

We have now included details of the comparison of rate shift models to the BPMM analyses in the methods (lines 864-865, 1482-1489) and added Extended Data figure 3 that shows the accuracy of our models in predicting B5 and B9 vitamins. Extended Data figure 3 also includes a figure of the ancestral values of B vitamins on the phylogenetic tree to enable readers to visualise the changes in B vitamins predicted by models in relation to feeding niche changes.

3. Estimating the impact of symbiosis on diversification:

a. The sister taxa approach that the authors are using is commendable. However, I see two problems with this analysis: i. Given that the origin of symbioses is collapsed on the family level, the analysis may not capture the necessary comparisons of symbiotic versus non-symbiotic sister taxa.

We are grateful to the reviewer for pointing out that the description of our analyses was not clear. As stated above, the analyses of sister lineages did not use binary classifications of obligate symbioses, but rather the percentage of species within families that had obligate symbionts (section 4.5.3, lines 961-962 of the original manuscript, lines 1598-1608 of new version).

We have now revised figure 4 and the legend to show more clearly that we analyse changes in the percentages of species with symbionts in families.

ii. I don't understand which taxa actually were compared. The authors describe 16 origins and 8 losses of symbiosis, but they indicate that 123 sister comparisons were done (l. 960). For this analysis, only the 24 transitions between symbiotic and non-symbiotic status seem to be relevant, so I do not understand where this large number of sister taxa comparison comes from. Again, it would be very helpful to have a clear list of origins and losses of symbiotic relationships presented, as well as the sister taxa comparisons that were used for this analysis. As it stands, it is impossible to evaluate the soundness of the approach.

14

24The reviewer is correct that there were 16 predicted origins and 8 predicted losses, providing 24 potential comparisons. However, not all of these 24 transitions were among tip sister taxa comparisons.

Across the whole tree, there were a total of 123 possible tip sister comparisons, of which 13 had differences in the percentages of species with obligate symbionts. Using this data, we analysed sister comparisons in two ways (sections 4.53 and 4.54 results presented in Supplementary Tables 18 and 19). First, we analysed species richness across all 123 sister taxa pairs than included comparisons where there were differences in obligate symbiosis (n=13) and comparisons where there were no differences (n=110). Both types of comparison were included as this enables the variance in species richness among sister taxa to modelled more accurately. Specifically, variation in species richness when rates of obligate symbiosis change versus when they do not change can be compared. Including all 123 comparisons also allowed us to verify the effects of feeding niche on species richness at a finer taxonomic scale than our main analysis.

Second, we restricted our sister lineage dataset to the 13 comparisons where there were different rates of obligate symbiosis. This allowed us to quantify the relationship between changes in the percentage of species with obligate symbionts and changes in species richness.

We realise the rationale for each analysis and the reporting of our results was insufficient. We have therefore revised the results (lines 424-427) and methods sections (lines 1600-1604) accordingly, and provided a table of the sister lineages analysed (Supplementary Table 29).

b. The other two approaches are problematic, since it is not clear how the non-symbiotic families included in the phylogeny were selected, so it remains unclear how much of a sampling bias there is. For example, the most diverse order, the beetles, are only represented by 11 families in total, out of which 9 are considered to be symbiotic, even though there are more than 170 beetle families, most of which are non-symbiotic (and some really large non-symbiotic families are missing, e.g. Staphylinidae, Tenebrionidae, as well as the phytophagous Buprestidae). Furthermore, the data presented in Figure 4a cannot be correct. There are an estimated 1 million described insect species to date (10^6), but some individual families in the figure are indicated to have 10^9 species.

The families in our database were included based on several methods. Specifically, searching the published literature using the key words [order name] OR [family name] and “symbio*”, by searching prominent reviews (e.g. Buchner 1965 Schneider 1939, Bourtzis and miller 2003, 2006, 2009), and forward and backward searches from the paper. Families included in the literature search were those in published phylogenies investigating insect biodiversity: e.g. Hedges et al. 201553, Misof et al 201454, and Rainford et al 201412. See methods on ‘Data Collection’. Many beetle families were included in our search, but unfortunately, they are understudied for symbionts. However, Staphylinidae and Tenebrionidae were investigated. Tenebrionids had sufficient evidence to show they are asymbiotic, whereas Staphylinids, which harbour defensive symbionts, were excluded because there was insufficient evidence to classify the symbiosis as obligate or not (see response to Reviewer 1). To the author’s knowledge, there are no published studies investigating beneficial endosymbioses in Buprestid beetles.

With respect to species numbers, the y-axis in figure 4a and 4b are on the log scale ranging from 3 to 9. This equates to a range of 20 to 8103 average species per category on the raw

15

25data scale (range per family was 1 to 50615; Supplementary Table 1). We have now clarified in the figure legend that axes are on a natural logarithmic scale.

I. 1: The title seems somewhat overstated, since the manuscript does not demonstrate niche expansions (rather: shifts), nor does it establish a causal link between symbiosis and adaptive radiations.

We have now changed the title to ‘Symbioses shape feeding niches and diversification across insects’.

I. 80: It is important to somewhere explicitly list the 16 reconstructed evolutionary origins of symbioses and the eight losses, as they are difficult to impossible to extract from Fig. 1 and Table S5.

We agree and have edited Supplementary Table 5 to make this information more accessible. Specifically, we have ordered the table so that the origins of symbiosis appear at the top and are highlighted in blue, followed by the losses of symbiosis that are highlighted in grey, and then the nodes where there was no change in symbiosis.

I. 159: Here and elsewhere, the authors put too much weight on the B vitamins as the nutritional components driving the evolution of symbiosis, while it has been shown that other factors can be the ones driving this (e.g. eAAs, digestive enzymes). In particular in II. 307-308: This is correlational, so causality cannot be inferred! Stating that, based on this analysis of inferred ancestral diets, B vitamins are the only nutrients found to significantly correlate with the presence of symbionts is fine, but claiming that this was the driving force behind the evolution of symbiosis is not.

We have tempered the wording around B vitamins and the evolution of obligate symbiosis on lines 307-308 and elsewhere (e.g. lines 210-212, 491-199). Just to clarify, our intention was not to propose that B vitamins are the only nutritional component driving the evolution of obligate symbiosis. Instead, it was to highlight that B vitamins are associated with obligate symbiosis across diverse insect families with different feeding niches, whereas other nutrients provided by symbionts appear to be restricted to certain feeding niches and/or insect families.

I. 610-611: This seems like a strange approach. Given that there are missing values for some dietary components in some diets, this can have a strong impact on the mean, especially if a number of micronutrients is missing, pushing the mean towards much lower values. I understand the rationale for avoiding the problem of compositional data (although I am not sure whether using this standardization by dividing by the mean instead of the sum actually solves this, since the results will still depend on the other dietary components), but the approach the authors are using created a substantial problem when comparing nutrient composition across diets.

Dividing by the mean is standard practice when quantifying the relative amount of an individual entity (e.g. nutrient) within a sampling unit (e.g. food type). Perhaps the most widespread use of standardising by the mean in evolutionary biology is the measurement of relative fitness of individuals within a population (e.g. Lande, R., and S. J. Arnold. 1983. The Measurement of Selection on Correlated Characters. *Evolution* 37:1210–1226). Furthermore, mean standardization provides a simple way to compare across sampling units in a comparable way (food types, populations ect..). Standardising nutrient values across food

16

26types was necessary because of the different ways that nutrient values were presented (g per wet weight versus per dry weight).

With respect to missing data, mean standardization is more robust to missing values than dividing by the sum. This is because the sum is always directly influenced by missing values, whereas the mean is only influenced if there is systematic bias in the missing values with respect to the measured values. For example, if missing values are larger than measured values then the mean will be underestimated and the relative amounts of measured nutrients will be overestimated. If missing values are random, then the mean is unlikely to change and standardized values will not be affected.

Perhaps the most important point regarding missing data is that it is not systematically biased towards families with or without symbionts. Rates of missing data were generally low and similar for all nutrients between families with and without symbiont (See Table below and Supplementary Table 1).

Table 1: Proportion of missing data across nutrients for families with and without obligate symbionts

Nutrient	Non-obligate proportion missing	Obligate proportion missing
Carbs	0.00	0.00
Protein	0.00	0.00
Fat	0.01	0.07
Essential amino acids	0.00	0.00
Non-essential amino acids	0.00	0.00
Vitamin A	0.15	0.43
Vitamin B	0.01	0.07
Vitamin E	0.25	0.43

Reviewer #5 (Remarks to the Author):

This manuscript seeks to study the macroevolution of symbiosis in insects. The paper is well written; however, I felt a little dazzled by results and p-mcmc values. So much so that I became slightly confused about which ones are the most important and which are linked to which hypothesis. I think in any revision I would recommend the author go through the manuscript with the reader in mind a really link the results to the specific hypotheses and expectations.

We are grateful for the constructive feedback and have edited the manuscript throughout to improve the links between analyses and the key hypotheses.

My expertise is in phylogenetic comparative methods so I'm reading the manuscript from that perspective. I think the analyses on the whole look well done. The tree itself is very difficult and I think that needs to be made more clear in any revision. The dating and topological uncertainty of the tree is not really considered. Whilst I appreciate it would be far outside the scope of this study to produce a better foundation tree – I think this need to be discussed more throughout.

17

27The phylogenetic tree was produced in a separate publication by other authors (Rainford, J. L., M. Hofreiter, D. B. Nicholson, and P. J. Mayhew. 2014. Phylogenetic Distribution of Extant Richness Suggests Metamorphosis Is a Key Innovation Driving Diversification in Insects. *PLoS ONE* 9:e109085). While the dating procedures and topological uncertainty of the tree are discussed in their publication, our manuscript and several other papers, including Wiens *et al* 2015 *Nat Commun* 6, 8370 and Condamine *et al* 2016 *Scientific Reports* 6: 19208, use the tree for comparative analyses because it has the broadest sampling of insects (874 terminal taxa) of trees constructed using multiple genes.

We have now included a statement in the methods highlighting where information on dating and topological uncertainty of the tree can be found (lines 1192-1193).

I have a few important concerns that I think the authors should consider:

1. Number of species in a family. The number of species estimated to be in each family is a critical estimate to many of the analyses in this study. However, I think it will be quite likely that the number of species estimated in each family will be highly correlated with research effort, which will in turn be associated with occurrences of symbiosis and other covariates important here. So given this, I think the author need to control for research effort (associated with each family) in their analyses.

This is an important point. The estimates of species richness were taken from previously published studies that collated information from encyclopaedias (See Table S1 of Rainford, J. L., M. Hofreiter, D. B. Nicholson, and P. J. Mayhew. 2014. Phylogenetic Distribution of Extant Richness Suggests Metamorphosis Is a Key Innovation Driving Diversification in Insects. *PLoS ONE* 9:e109085). This past research effort used to estimate species richness was therefore completely independent from our current work quantifying rates of obligate symbiosis across families. Consequently, there was a very weak relationship between the number of species examined for obligate symbiosis and the species richness of families (Figure 1).

Figure 1: Correlation between species richness and the number of species examined per family for obligate symbiosis (Pearson's correlation coefficient: $R = 0.15$).

In addition, we controlled for research effort in analyses of obligate symbiosis by analysing the number of species where obligate symbionts have been found versus the number of species where obligate symbionts have not been found. In doing so, we control for differences in the number of species examined for obligate symbiosis across families (e.g. lines 972-974 of previous manuscript, lines 1571-1572 of new version).

2. Stochastic mapping and Discrete. I am slightly confused about why the authors need to use both stochastic mapping and Discrete. I would suggest they use the best model (dependent/independent) and then use that to determine the ancestral states of the nodes of interest. This part of the methods was very unclear – why do you need Stochastic mapping at all here?

Both stochastic character mapping (SCM) and Discrete were used as they have different strengths tailored to our different aims.

First, we wanted to reconstruct feeding niche states (categorical variable with 8 states). Discrete is designed to test if two binary traits are correlated, and therefore is not designed for reconstructing the ancestral states of a single trait with multiple categories. Stochastic character mapping is designed for this purpose (Bollback, J. P. 2006. SIMMAP: Stochastic character mapping of discrete traits on phylogenies. BMC Bioinformatics 7:88).

Second, we wanted to test if obligate symbiosis was phylogenetically correlated with levels of B vitamins, and if so, whether changes in B vitamins preceded or followed the evolution of obligate symbiosis. Discrete allows us to do this by testing if B vitamins and obligate symbiosis were correlated, and by estimating transition rates between different states. SCM models can technically be used to test if two binary characters are correlated by specifying custom Q transition rate matrices. However, this is not as widely used as Discrete and Discrete has the advantage of being able to specify hyperpriors that is not available in make.simmmap (the R function used to perform SCM). Given this extra functionality and that Discrete is more widely used for testing for correlated evolution, we decided to use it over SCM.

More broadly, we have taken the policy of using two different types of analyses for addressing the same question (e.g. linear models versus continuous-time Markov models in sections 4.3.1 and 4.3.2), rather than using a single technique to identify a 'best' model. This is in order to verify that our conclusions were robust to the specifics of individual analytical techniques.

3. The significance and difference of rates in the Discrete mode. The authors say they use the posterior distribution of rates to determine whether the rates are significantly different to each other. However, there is a reversible-jump procedure available in BayesTraits that is specifically designed to reduce dimensionality and determine if there are differences among rates – I think the authors should use that.

We are aware of the reversible-jump procedure and that it reduces dimensionality by identifying transition rates that are unlikely to occur (e.g. by setting them to 0). This information can also be used to test hypotheses about whether specific transitions are likely to occur or not: the more posterior samples where transitions are set to 0 the less likely transitions are to occur.

However, estimating the likelihood of transitions occurring does not directly test if transition rates are significantly different from each other. To do this requires estimating the difference between the posterior distributions of transition rates. This was crucial for our analyses as we wanted to test if: (1) transitions to obligate symbiosis were significantly higher from a background of low versus high B vitamins; and (2) transitions to low B vitamins were higher from a background of obligate versus non-obligate symbiosis. We tested these hypotheses using standard Bayesian techniques for examining whether posterior distributions are different from each other (see Gelman, A., J. B. Carlin, H. S. Stern, D. B. Dunson, A. Vehtari, and D. B. Rubin. 2013. *Bayesian Data Analysis*. 3rd edition. Chapman and Hall).Decision Letter, first revision:

12th December 2022

Dear Charlie,

Your revise manuscript entitled "Symbioses shape feeding niches and diversification across insects" has now been seen by the same five reviewers, whose comments are attached. You will see that the first four reviewers are largely satisfied with the revisions but our phylogenetics expert still has major concerns which will need to be addressed before we can offer publication in Nature Ecology & Evolution. We will therefore need to see your responses to the criticisms raised and to some editorial concerns, along with a revised manuscript, before we can reach a final decision regarding publication.

We therefore invite you to revise your manuscript taking into account all reviewer and editor comments. Please highlight all changes in the manuscript text file in Microsoft Word format.

* If you have not done so already please begin to revise your manuscript so that it conforms to our Article format instructions at <http://www.nature.com/natecolevol/info/final-submission>. Refer also to any guidelines provided in this letter.

[REDACTED]

Note: This URL links to your confidential home page and associated information about manuscripts you may have submitted, or that you are reviewing for us. If you wish to forward

31this email to co-authors, please delete the link to your homepage.

Nature Ecology & Evolution is committed to improving transparency in authorship. As part of our efforts in this direction, we are now requesting that all authors identified as 'corresponding author' on published papers create and link their Open Researcher and Contributor Identifier (ORCID) with their account on the Manuscript Tracking System (MTS), prior to acceptance. ORCID helps the scientific community achieve unambiguous attribution of all scholarly contributions. You can create and link your ORCID from the home page of the MTS by clicking on 'Modify my Springer Nature account'. For more information please visit www.springernature.com/orcid.

[REDACTED]

Reviewers' comments:

Reviewer #1 (Remarks to the Author):

This is an excellent revision of the paper. The authors have taken on board comments in this new version, improving on what was already a compelling manuscript. Congratulations on a job well done!

Reviewer #2 recommends publication with no further comments to the authors.

Reviewer #3 (Remarks to the Author):

I thank the authors for their very thoughtful revision. After carefully reading this new version, I think that all of my criticism has been adequately addressed. In fact, I think that this new version is much clearer and easier to understand than the previous one. Especially the revised figures and descriptions of the data shown make a huge difference. Together, I think that this manuscript makes a substantial contribution to the field and fills an important gap. Thus, I recommend accepting it with minor revisions.

Below, I have one more suggestion that I think would even further enhance the manuscript:

32One of the key findings of the paper is that B vitamins was the only nutrient limitation that significantly correlated with the presence of obligate symbionts. This is a very interesting observation. While the authors acknowledge that also other deficiencies might play a role alone or in combination with vitamins, they do not address the question why the vitamins seem to stand out. I think the paper would significantly benefit from an analysis or at least a more detailed discussion of why this might be the case. For example, is it that a) vitamin auxotrophies evolve first and only then other metabolic genes are lost or b) vitamin auxotrophies are simply more common. The second point could be addressed in this way: Rather than showing the relative nutrient deficiencies (e.g. Fig. 2), the authors might want to consider to also display/ analyze the overall distribution of absolute deficiencies in their data set.

Moreover, this issue receives a more in-depth treatment in the discussion section.

Minor suggestions:

Line 53: Please reword „recruit“. This sounds like an active searching process.

Line 201: Consider changing the order of “negatively” and “phylogenetically” to read “are phylogenetically negatively correlated”.

Reviewer #4 (Remarks to the Author):

The authors have done a thorough job in addressing the reviewers’ comments. They put together an enormous database and analyze it in a phylogenetic framework. While some of the conclusions are not entirely surprising, an analysis of insect symbioses at this scale is very valuable for the community as a whole and will present an important basis for the further broad-scale studies. There are a few minor comments remaining from my side, but they are mostly issue regarding wording or small errors. However, one important aspect that I would like to point out is that the comment about the consistent results of the sensitivity analyses (l. 464-465) is not true and should be carefully reworded, as amino acids repeatedly come up as significant, but in an unexpected direction. This needs to be explained. Why should symbiotic lineages be more likely to feed on eAA-rich diet, if I understand the results presented in Supplementary Tables 20, 21, and 24 correctly? As this seems to be consistent across all three “Sensitivity Analyses”, it needs a thorough explanation.

Specific comments:

Comment on the rebuttal: In Suppl. Table 5, there are also nonOb to nonOb and Ob to Ob transitions highlighted, which they shouldn’t be.

l. 80-82: Just to be clear, please specify that co-speciation and negative fitness consequences of symbiont removal were used as obligate criteria. Just a small comment here (that I know will not be relevant to the results, since you carefully selected the symbioses): Strictly speaking, your two criteria would include some associations with Wolbachia as a reproductive manipulator, because it can be fixed in certain insect species, and removal would lead to severe fitness defects under CI (sometimes

33even extremely so, like in *Asobara* wasps). Not sure how to avoid this issue, but I wanted to at least point it out, but I assume the authors are anyways aware of it...

l. 139: please correct typo in "drosophilids"

l. 167: Please correct typo in "lagriid"

l. 181: the addition of "digestive enzymes" makes sense, but please revise, as these are not essential nutrients

Fig. 3 C+D: Could you please provide statistics for the differences between groups?

Fig.- 3 in general: I appreciate the authors' efforts to clarify what is depicted in this figure. However, I urge them to consider whether there is a better way to display this, or at least relabel. "Non-obligate maintenance" is misleading, and it really took me a long time to grasp the figure in its full extent. In part, this was due to the use of the term "vs" in C and D, which I interpreted as divided by, which is exactly what it apparently is not supposed to mean here (rather, it's the reverse)?

l. 352: would be good to mention that this is also true for essential amino acid supplementing symbionts.

l. 464-465: This is not exactly true. In most of the analyses, the eAAs (and sometimes non-eAAs) also come up as significant, which does change the interpretation of the results quite a bit. I strongly recommend to add a comment on this and carefully discuss it. And it seems that they strongly go into the opposite direction than the B vitamins, indicating that symbiotic lineages are more likely to feed on amino acid-rich diet, if I interpret this correctly? This would not only be contradicting most of the accumulated symbiosis knowledge, but also be highly unlikely. This needs to be checked.

l. 490: With the caveat indicated above... Don't get me wrong, I don't insist on eAAs being a general driving factor underlying the evolution of nutritional symbioses in insects, but considering the gaps in current understanding and the necessarily limited phylogenetic resolution, as well as the results of the sensitivity analyses, they may be similarly important as the B vitamins, so I think it would be wise to phrase carefully here.

Reviewer #5 (Remarks to the Author):

see attachment owing to the inclusion of a critical plot.

*****END*****

Author Rebuttal, first revision:

Response to reviewers

34We are very grateful to the referees for their continued efforts in helping us improve our manuscript. Below are the details of the specific changes we have made to address the referees' comments. Briefly, we have revised the text in our paper, added the extra analyses suggested by referee 5, and have added supplementary tables to present the results of the extra analyses.

Line numbers in word documents with track changes are not always reproducible across computers. Therefore, we refer to lines numbers in the pdf version of the revised manuscript.

Reviewer #1 (Remarks to the Author):

This is an excellent revision of the paper. The authors have taken on board comments in this new version, improving on what was already a compelling manuscript. Congratulations on a job well done!

Many thanks for the encouragement and help with our paper.

Reviewer #2 recommends publication with no further comments to the authors.

Thank you!

Reviewer #3 (Remarks to the Author):

I thank the authors for their very thoughtful revision. After carefully reading this new version, I think that all of my criticism has been adequately addressed. In fact, I think that this new version is much clearer and easier to understand than the previous one. Especially the revised figures and descriptions of the data shown make a huge difference. Together, I think that this manuscript makes a substantial contribution to the field and fills an important gap. Thus, I recommend accepting it with minor revisions.

Thank you for the positive feedback and continued effort in helping us improve the final details of our paper.

Below, I have one more suggestion that I think would even further enhance the manuscript:

35One of the key findings of the paper is that B vitamins was the only nutrient limitation that significantly correlated with the presence of obligate symbionts. This is a very interesting observation. While the authors acknowledge that also other deficiencies might play a role alone or in combination with vitamins, they do not address the question why the vitamins seem to stand out. I think the paper would significantly benefit from an analysis or at least a more detailed discussion of why this might be the case. For example, is it that a) vitamin auxotrophies evolve first and only then other metabolic genes are lost or b) vitamin auxotrophies are simply more common. The second point could be addressed in this way: Rather than showing the relative nutrient deficiencies (e.g. Fig. 2), the authors might want to consider to also display/ analyze the overall distribution of absolute deficiencies in their data set.

Moreover, this issue receives a more in-depth treatment in the discussion section.

This is an interesting point, and we appreciate the suggestion of adding analyses of absolute nutrient concentrations. The interpretation of such analyses is nevertheless not easy. Information on the nutrient contents of foods is given as concentrations. Therefore, to examine absolute amounts of nutrients requires data on intake rates, but such information is not available for many insect species. Furthermore, it is not clear that absolute nutrient concentrations accurately reflect the ability of species to synthesise compounds.

One possible way to address whether B-vitamin auxotrophies are more prevalent than other nutrient auxotrophies would be to conduct comparative genomic analyses. This touches upon a similar comment from referee 1 in the previous round of comments about including genomic analyses of vitamin synthesis pathways. For example, the completeness of pathways used to synthesis different nutrients could be analysed across species. However, this is a major undertaking, well beyond the current paper, but we hope our work stimulates more research in that direction. We have therefore taken the alternative suggestion made by the referee and extended the discussion of this point in the manuscript (lines 420-423).

Minor suggestions:

Line 53: Please reword „recruit“. This sounds like an active searching process.

Addressed.

Line 201: Consider changing the order of “negatively” and “phylogenetically” to read “are

36

phylogenetically negatively correlated”.

Many thanks for the suggestion. We have changed it.

Reviewer #4 (Remarks to the Author):

The authors have done a thorough job in addressing the reviewers’ comments. They put together an enormous database and analyze it in a phylogenetic framework. While some of the conclusions are not entirely surprising, an analysis of insect symbioses at this scale is very valuable for the community as a whole and will present an important basis for the further broad-scale studies. There are a few minor comments remaining from my side, but they are mostly issue regarding wording or small errors. However, one important aspect that I would like to point out is that the comment about the consistent results of the sensitivity analyses (lines 464-465) is not true and should be carefully reworded, as amino acids repeatedly come up as significant, but in an unexpected direction. This needs to be explained. Why should symbiotic lineages be more likely to feed on eAA-rich diet, if I understand the results presented in Supplementary Tables 20, 21, and 24 correctly? As this seems to be consistent across all three “Sensitivity Analyses”, it needs a thorough explanation.

We are very grateful for the supportive and insightful comments.

The sensitivity analyses presented in Supplementary Tables 20, 21 and 24 were all on datasets where the data exclusion criteria removed either insects with obligate symbionts and low amino acid diets (herbivores and phloem feeders - analyses 4.6.1 and 4.6.2), or insects without obligate symbionts and high amino acid diets (omnivores and predators - analysis 4.6.4). Removing these insect families therefore results in feeding niches, such as blood feeders (high rates of symbiosis and amino acids in their diets), inflating the correlation between symbiosis and amino acids. Therefore, we believe the positive correlation between symbiosis and amino acids that emerges when restricting data illustrates the importance of broad taxonomic sampling rather than anything biologically meaningful.

We have addressed this issue by editing lines 349-368 and adding a more detailed interpretation of the sensitivity analysis section (lines 1341-1354).

Specific comments:

Comment on the rebuttal: In Suppl. Table 5, there are also nonOb to nonOb and Ob to Ob transitions highlighted, which they shouldn't be.

Addressed

I. 80-82: Just to be clear, please specify that co-speciation and negative fitness consequences of symbiont removal were used as obligate criteria. Just a small comment here (that I know will not be relevant to the results, since you carefully selected the symbioses): Strictly speaking, your two criteria would include some associations with Wolbachia as a reproductive manipulator, because it can be fixed in certain insect species, and removal would lead to severe fitness defects under CI (sometimes even extremely so, like in Asobara wasps). Not sure how to avoid this issue, but I wanted to at least point it out, but I assume the authors are anyways aware of it...

Thank you for highlighting that this was unclear. The use of co-speciation and negative fitness consequences as secondary criteria is mentioned in our introduction (lines 73-76). In our data exclusion section (lines 704-713), we indicated that studies of known parasitic symbionts, such as those that manipulate host reproduction and have not evolved beneficial functions (e.g. *Spiroplasma*, *Cardinium*, *Wolbachia*), were excluded from our dataset. To make this clearer from the outset, we have added that these symbionts were excluded at the end of the introduction (lines 76-78).

I. 139: please correct typo in “drosophilids”

Addressed.

I. 167: Please correct typo in “lagriid”

Addressed.

I. 181: the addition of “digestive enzymes” makes sense, but please revise, as these are not essential nutrients

Addressed.

Fig. 3 C+D: Could you please provide statistics for the differences between groups?

Addressed.

Fig.- 3 in general: I appreciate the authors’ efforts to clarify what is depicted in this figure. However, I urge them to consider whether there is a better way to display this, or at least relabel. “Non-obligate maintenance” is misleading, and it really took me a long time to grasp the figure in its full extent. In part, this was due to the use of the term “vs” in C and D, which I interpreted as divided by, which is exactly what it apparently is not supposed to mean here (rather, it’s the reverse)?

We have clarified in the figure legend to make this clearer.

I. 352: would be good to mention that this is also true for essential amino acid supplementing symbionts.

This is a good point. Given this paragraph is specifically focused on B-vitamins, we have instead included this when discussing the role of amino acids in the evolution of symbiosis (lines 399-423).

I. 464-465: This is not exactly true. In most of the analyses, the eAAs (and sometimes non-eAAs) also come up as significant, which does change the interpretation of the results quite a bit. I strongly recommend to add a comment on this and carefully discuss it. And it seems that they strongly go into the opposite direction than the B vitamins, indicating that symbiotic lineages are more likely to feed on amino acid-rich diet, if I interpret this correctly? This would not only be contradicting most of the accumulated symbiosis knowledge, but also be highly unlikely. This needs to be checked.

39Agreed. Please see response to the comment above. We have added a section to the manuscript explaining that we think this is most likely due to sampling biases introduced by removing species from specific feeding niches rather than a real effect (lines 1341-1354).

I. 490: With the caveat indicated above... Don't get me wrong, I don't insist on eAAs being a general driving factor underlying the evolution of nutritional symbioses in insects, but considering the gaps in current understanding and the necessarily limited phylogenetic resolution, as well as the results of the sensitivity analyses, they may be similarly important as the B vitamins, so I think it would be wise to phrase carefully here.

We have rephrased this sentence in line with the referee's suggestion.

Reviewer #5 (Remarks to the Author):

I appreciate the authors have considered my comments. However, I still have some considerable concerns around these issues. I will go through them here.

We are grateful to the referee for once again helping us clarify important issues in our manuscript.

With regard to the tree uncertainty the authors added the following sentence in the methods 'For information on the dating methods and topological uncertainties see Rainford et al. 201412". This is not sufficient and certainly does not describe the dating and uncertainty in the tree. I think the authors need to be honest that the results could be dependent on this tree of insects and that there is lost uncertainty and polytomies in the tree itself that is not considered. I know the paper has been used before – but I think it is the authors duty to be honest that it is not clear what would happen in the case that these results were integrated across the uncertainty. However, I don't think this, in itself, should preclude publication of this work. It just needs to be made clear in the main text.

Thank you for this suggestion. We have now extended our description of how the Rainford tree was constructed and how analyses that account for tree uncertainty are an important extension of this work (lines 837-845).

40The authors response about the fact that number of species estimated in each family will be highly correlated with research effort is not at all satisfactory – and rather strange. In the figure 1 of their rebuttal they plot raw species richness verses raw number of species examined. As expected, they would not find a strong corelation! It is well established and obvious that these data should be transformed (the are bounded at zero!) for this demonstration. If one plots the transformed data (below) it is clear the corelation is very strong! It is also evident that the data are very singleton inflated (an over preponderance of ones in the dataset). Thus, this need very careful attention in the reanalyses for the data for another submission – I suggest the authors work with a statistician on this issue as there is some considerable confusion evident here!

We realise that our response to the referee’s original comment was confusing and are grateful for the opportunity to clarify. With regards to the above issues, the plot we included of species richness and the number of symbiont species studied does not help resolve the issues raised by the referee: a positive correlation between number of species examined for symbionts and species richness is expected both with and without biases in research effort, making it rather redundant.

Going back to the referee’s original concern: *“I think it will be quite likely that the number of species estimated in each family will be highly correlated with research effort, which will in turn is likely to be associated with occurrences of symbiosis and other covariates important here. So given this, I think the author need to control for research effort (associated with each family) in their analyses.”*

The main objective of our species richness analyses was to test if obligate symbiosis is related to the number of species within families. The referee’s concern is that variation in research effort will generate a spurious correlation between the number of species in each family and occurrences of symbiosis. There are three main reasons why this is not the case:

First, there was no correlation between species richness and obligate symbiosis as expected if research effort confounds our results. Instead, we found obligate symbiosis was associated with both extreme highs and lows of species richness.

Second, we analysed the proportion of species within families rather than the number of species with symbionts. Increased research effort (number of species checked for symbionts) is expected to increase the accuracy of the estimated proportion of species with obligate symbionts per family, but not systematically bias the mean proportion. It is therefore difficult to see how variation in research effort will induce correlations between the mean proportion of species with obligate symbionts and species richness.

Third, our analyses explicitly control for variation in the effort used to measure obligate symbiosis (e.g. the number of species examined for obligate symbionts). Specifically, we model the binomial outcome of the number of species with obligate symbionts out of the total number of species examined for symbionts (binomial BPMM presented in analysis 4.5.1). This approach accounts for variation in the number of species studied (research effort) across insect families. We have consulted with three experts about the approaches we use that are at the forefront of phylogenetic comparative analyses (Jarrod Hadfield, Shinichi Nakagawa and Dan Noble).

We agree with the referee that it would be ideal to account for the effort used to estimate species richness, but this is extremely difficult. The estimates of species richness come from Rainford *et al* 2014, who compiled information from field guides, encyclopaedias, museum collections, phylogenetic studies and online databases. Species richness was typically estimated by summing up numbers of species from these sources, which do not have quantitative measures of research effort that could be included in analyses (we do not know of any previous studies on diversification across insect families that have managed to do this). It is also difficult to see how variation in estimates of species richness would systematically bias our results. Estimates of species richness may be higher for some clades because they have received more attention (e.g. the attractiveness of butterflies means they are well studied), but symbionts are invisible to observers and so are unlikely to suffer from the same biases.

In line with the referee's suggestion, we have now added a discussion of the issues of sampling effort to the section on measuring species richness (lines 827-833).

The authors comments about Stochastic mapping and Discrete are not correct. The discrete model is for correlated models but there is another element of the program BayesTraits called Multistate that is far more appropriate for ancestral reconstruction – Stochastic mapping should only really be used for visualisation. The Bayesian implementation of Multistate was introduced in 2004 (Systematic Biology, Volume 53, Issue 5, October 2004, Pages 673–684) and has been cited over 1000 times.

The referee previously asked about the use of stochastic character mapping (SCM) and Discrete, not Multistate. As a result, our reply was focused on justifying why we used Discrete and Stochastic character mapping. To recap, we used Discrete to test for correlated evolution between obligate symbiosis and B-vitamins, which we think the referee agrees is appropriate here. SCM was used to reconstruct ancestral states of feeding niches. The new issue raised in this comment is that a different module of BayesTraits – Multistate - is more appropriate for this purpose than SCM.

SCM and Multistate can be used for similar analytical purposes – ancestral state reconstruction of multi-state categorical traits. SCM was designed for ancestral state reconstruction not just visualisation (details are given in^{1,2}, and as Bollback 2006 describes “SIMMAP can be used to calculate the posterior distribution of ancestral states”). SCM has been used to infer the ancestral states of a variety of categorical traits, including feeding niches, across a variety of organisms (e.g.³⁻⁶), which is why we choose to use this technique. However, we have added extra analyses that reconstruct the ancestral states of feeding niches using Multistate that show very similar results to our SCM analysis: 94% of nodes are predicted to have the same ancestral feeding niche by SCM and BayesTraits, which are presented in new supplementary table 20 and lines 1037-1054.

The authors suggest that they are talking the point of view that they use many techniques it makes there results more robust – but this is not true. As all these techniques have different assumption which need to be considered. And the most appropriate should be used. Again, I think some advice from a statistician might help here.

We agree with the referee that some analytical techniques are more appropriate for some purposes than others. However, some methods have been designed to address similar types of questions, but make different assumptions. For example, SCM and Multistate have been used to address similar problems, and in some cases have even be used on the same data to check the robustness of results to the analytical approach used^{5,7}. Therefore there are situations where different techniques with different assumptions are equally appropriate. We believe, as has been argued by others⁷, that in such cases conducting analyses with multiple techniques helps verify that results are robust to different analytical assumptions.

Unfortunately, the authors interpretation of the reversible-jump procedure available in

BayesTraits is not correct. This procedure can reduce the models in various ways not just on and off. This procedure is far more suitable for testing the phylogenetic based hypotheses than the comparison of posterior estimates – which might be fine in lots of contexts but has some pitfalls as outlined in the paper describing the procedure (Am Nat. 2006;167(6):808-25) which again has been cited almost 1000 times.

As suggested, we have now added BayesTraits analyses using the reversible-jump procedure that provide additional tests of differences between transition rates (Supplementary Tables 11 and 28).

Cited literature

1. Bollback, J. P. SIMMAP: Stochastic character mapping of discrete traits on phylogenies. *BMC Bioinformatics* **7**, 88 (2006).
2. Huelsenbeck, J. P., Nielsen, R. & Bollback, J. P. Stochastic Mapping of Morphological Characters. *Syst. Biol.* **52**, 131–158 (2003).
3. Song, H., Foquet, B., Mariño-Pérez, R. & Woller, D. A. Phylogeny of locusts and grasshoppers reveals complex evolution of density-dependent phenotypic plasticity. *Sci. Rep.* **7**, 6606 (2017).
4. Gajdzik, L., Aguilar-Medrano, R. & Frédérix, B. Diversification and functional evolution of reef fish feeding guilds. *Ecol. Lett.* **22**, 572–582 (2019).
5. Leschen, R. A. B. & Buckley, T. R. Multistate Characters and Diet Shifts: Evolution of Erotylidae (Coleoptera). *Syst. Biol.* **56**, 97–112 (2007).
6. Collar, D. C., O'Meara, B. C., Wainwright, P. C. & Near, T. J. Piscivory Limits Diversification of Feeding Morphology in Centrarchid Fishes. *Evolution* **63**, 1557–1573 (2009).
7. Ekman, S., Andersen, H. L. & Wedin, M. The Limitations of Ancestral State Reconstruction and the Evolution of the Ascus in the Lecanorales (Lichenized Ascomycota). *Syst. Biol.* **57**, 141–156 (2008).

Decision Letter, second revision:

24th January 2023

Dear Charlie,

Thank you for submitting your revised manuscript "Symbioses shape feeding niches and diversification across insects" (NATECOLEVOL-220616867B). It has now been seen again by Reviewer #5 and their

44comments are below. The reviewers find that the paper has improved in revision, and therefore we'll be happy in principle to publish it in Nature Ecology & Evolution, pending minor revisions to comply with our editorial and formatting guidelines.

[REDACTED]

Reviewer #5 (Remarks to the Author):

I thank the authors for their consideration of my comments. I think it is clear that I and they do not see eye-to-eye on some of the technical issues associated with the analyses. However, I don't think there is much value in prolonging this discussion here. The authors have been reasonable in their responses and largely considered my comments in the text. I enjoyed the manuscript from the outset, and congratulate the authors on an interesting piece of work.

Our ref: NATECOLEVOL-220616867B

27th January 2023

Dear Dr. Cornwallis,

Thank you for your patience as we've prepared the guidelines for final submission of your Nature Ecology & Evolution manuscript, "Symbioses shape feeding niches and diversification across insects" (NATECOLEVOL-220616867B). Please carefully follow the step-by-step instructions provided in the attached file, and add a response in each row of the table to indicate the changes that you have made. Please also check and comment on any additional marked-up edits we have proposed within

45the text. Ensuring that each point is addressed will help to ensure that your revised manuscript can be swiftly handed over to our production team.

****We would like to start working on your revised paper, with all of the requested files and forms, as soon as possible (preferably within two weeks). Please get in contact with us immediately if you anticipate it taking more than two weeks to submit these revised files.****

In recognition of the time and expertise our reviewers provide to Nature Ecology & Evolution's editorial process, we would like to formally acknowledge their contribution to the external peer review of your manuscript entitled "Symbioses shape feeding niches and diversification across insects". For those reviewers who give their assent, we will be publishing their names alongside the published article.

Nature Ecology & Evolution offers a Transparent Peer Review option for new original research manuscripts submitted after December 1st, 2019. As part of this initiative, we encourage our authors to support increased transparency into the peer review process by agreeing to have the reviewer comments, author rebuttal letters, and editorial decision letters published as a Supplementary item. When you submit your final files please clearly state in your cover letter whether or not you would like to participate in this initiative. Please note that failure to state your preference will result in delays in accepting your manuscript for publication.

Cover suggestions

As you prepare your final files we encourage you to consider whether you have any images or illustrations that may be appropriate for use on the cover of Nature Ecology & Evolution.

46Nature Ecology & Evolution has now transitioned to a unified Rights Collection system which will allow our Author Services team to quickly and easily collect the rights and permissions required to publish your work. Approximately 10 days after your paper is formally accepted, you will receive an email in providing you with a link to complete the grant of rights. If your paper is eligible for Open Access, our Author Services team will also be in touch regarding any additional information that may be required to arrange payment for your article.

Please note that *Nature Ecology & Evolution* is a Transformative Journal (TJ). Authors may publish their research with us through the traditional subscription access route or make their paper immediately open access through payment of an article-processing charge (APC). Authors will not be required to make a final decision about access to their article until it has been accepted. [Find out more about Transformative Journals](https://www.springernature.com/gp/open-research/transformative-journals)

Authors may need to take specific actions to achieve [compliance with funder and institutional open access mandates](https://www.springernature.com/gp/open-research/funding/policy-compliance-faqs). If your research is supported by a funder that requires immediate open access (e.g. according to [Plan S principles](https://www.springernature.com/gp/open-research/plan-s-compliance)) then you should select the gold OA route, and we will direct you to the compliant route where possible. For authors selecting the subscription publication route, the journal's standard licensing terms will need to be accepted, including [the journal's standard licensing terms](https://www.nature.com/nature-portfolio/editorial-policies/self-archiving-and-license-to-publish). Those licensing terms will supersede any other terms that the author or any third party may assert apply to any version of the manuscript.

[REDACTED]

[REDACTED]

Reviewer #5:

Remarks to the Author:

I thank the authors for their consideration of my comments. I think it is clear that I and they do not see eye-to-eye on some of the technical issues associated with the analyses. However, I don't think there is much value in prolonging this discussion here. The authors have been reasonable in their responses and largely considered my comments in the text. I enjoyed the manuscript from the outset, and congratulate the authors on an interesting piece of work.

Final Decision Letter:

15th March 2023

Dear Charlie,

We are pleased to inform you that your Article entitled "Symbioses shape feeding niches and diversification across insects", has now been accepted for publication in Nature Ecology & Evolution.

Over the next few weeks, your paper will be copyedited to ensure that it conforms to Nature Ecology and Evolution style. Once your paper is typeset, you will receive an email with a link to choose the appropriate publishing options for your paper and our Author Services team will be in touch regarding any additional information that may be required

You will not receive your proofs until the publishing agreement has been received through our system

Due to the importance of these deadlines, we ask you please us know now whether you will be difficult to contact over the next month. If this is the case, we ask you provide us with the contact information (email, phone and fax) of someone who will be able to check the proofs on your behalf, and who will be available to address any last-minute problems . Once your paper has been scheduled for online publication, the Nature press office will be in touch to confirm the details.

Acceptance of your manuscript is conditional on all authors' agreement with our publication policies (see www.nature.com/authors/policies/index.html). In particular your manuscript must not be published elsewhere and there must be no announcement of the work to any media outlet until the publication date (the day on which it is uploaded onto our web site).

Please note that *Nature Ecology & Evolution* is a Transformative Journal (TJ). Authors may publish their research with us through the traditional subscription access route or make their paper immediately open access through payment of an article-processing charge (APC). Authors will not be required to make a final decision about access to their article until it has been accepted. ](https://www.springernature.com/gp/open-research/transformative-journals) Find out more about Transformative Journals

Authors may need to take specific actions to achieve compliance with funder and institutional open access mandates. If your research is supported by a funder that requires immediate open access (e.g. according to Plan S principles) then you should select the gold OA route, and we will direct you to the compliant route where possible. For authors selecting the subscription publication route, the journal's standard licensing terms will need to be accepted, including https://www.nature.com/nature-portfolio/editorial-policies/self-archiving-and-license-to-publish. Those licensing terms will supersede any other terms that the author or any third party may assert apply to any version of the manuscript.

An online order form for reprints of your paper is available at https://www.nature.com/reprints/author-reprints.html. All co-authors, authors' institutions and authors' funding agencies can order reprints using the form appropriate to their geographical region.

We welcome the submission of potential cover material (including a short caption of around 40 words) related to your manuscript; suggestions should be sent to Nature Ecology & Evolution as electronic files (the image should be 300 dpi at 210 x 297 mm in either TIFF or JPEG format). Please note that such pictures should be selected more for their aesthetic appeal than for their scientific content, and that colour images work better than black and white or grayscale images. Please do not try to design a cover with the Nature Ecology & Evolution logo etc., and please do not submit composites of images related to your work. I am sure you will understand that we cannot make any promise as to whether any of your suggestions might be selected for the cover of the journal.

To assist our authors in disseminating their research to the broader community, our SharedIt initiative provides you with a unique shareable link that will allow anyone (with or without a subscription) to read the published article. Recipients of the link with a subscription will also be able to download and

49print the PDF.

You can generate the link yourself when you receive your article DOI by entering it here: http://authors.springernature.com/share.

[REDACTED]

P.S. Click on the following link if you would like to recommend Nature Ecology & Evolution to your librarian <http://www.nature.com/subscriptions/recommend.html#forms>

** Visit the Springer Nature Editorial and Publishing website at www.springernature.com/editorial-and-publishing-jobs for more information about our career opportunities. If you have any questions please click here. **